# Genotype-specific differences in infertile men due to loss-of-function variants in *M1AP* or ZZS genes

Nadja Rotte [1], Jessica E M Dunleavy [2], Michelle D Runkel[1], Lina Bosse [1], Daniela Fietz[3], Adrian Pilatz [4], Johanna Kuss[1], Ann-Kristin Dicke [1], Sofia B Winge[5], Sara Di Persio [6], Christian Ruckert[7], Verena Nordhoff [6], Hans-Christian Schuppe [4], Kristian Almstrup [5,8], Sabine Kliesch [6], Nina Neuhaus [6], Birgit Stallmeyer [1], Moira K O'Bryan [2], Frank Tüttelmann [1] & Corinna Friedrich [1]✉

## Abstract

**Male infertility has been linked to M1AP. In mice, M1AP interacts with the ZZS proteins SHOC1/TEX11/SPO16, promoting DNA class I crossover formation during meiosis. To determine whether M1AP and ZZS proteins are involved in human male infertility by recombination failure, we screened for biallelic/hemizygous loss-of-function (LoF) variants in the human genes to select men with presumed protein deficiency ($N = 24$). After in-depth characterisation of testicular phenotypes, we identified gene-specific meiotic impairments: men with ZZS deficiency shared an early meiotic arrest. Men with LoF variants in *M1AP* exhibited a predominant metaphase I arrest with rare haploid round or even elongated spermatids. These differences were explained by different recombination failures: deficient ZZS function led to incorrect synapsis of homologous chromosomes, unrepaired DNA double-strand breaks, and incomplete recombination. Abolished M1AP led to a reduced number of recombination intermediates and class I crossover. Medically assisted reproduction resulted in the birth of a healthy child, offering the possibility of fatherhood to men with LoF variants in *M1AP*. Our study establishes M1AP as an important, but non-essential, functional enhancer in meiotic recombination.**

**Keywords** M1AP; Meiosis; Crossover; Infertility; Recombination
**Subject Categories** Genetics, Gene Therapy & Genetic Disease; Urogenital System

## Introduction

Worldwide, around one in six adults is infertile (World Health Organization, 2023), and the underlying causes are equally distributed between both sexes (Vander Borght and Wyns, 2018). In men, the most severe forms of infertility are non-obstructive azoospermia (NOA) and cryptozoospermia, meaning that due to spermatogenic failure, no or only few spermatozoa are detected in the ejaculate (Nieschlag et al, 2023). For most of these cases, the only chance of fathering a child is through testicular sperm extraction (TESE) with subsequent medically assisted reproduction (MAR) using intracytoplasmic sperm injection (ICSI).

In many of these men, the absence of spermatozoa is caused by an arrest of spermatogenesis at meiosis (termed meiotic arrest or spermatocyte arrest; Wyrwoll et al, 2023b). It has been repeatedly described that this phenotype often arises via monogenic traits (Wyrwoll et al, 2023b; Krausz et al, 2020). One established disease gene for meiotic arrest is *M1AP*, which encodes meiosis 1-associated protein (Wyrwoll et al, 2020). In mice, M1AP has recently been shown to interact with three well-characterised meiosis-related proteins: SHOC1, TEX11, and SPO16 (Li et al, 2023). These form a highly conserved complex, called ZZS (from yeast Zip2/Zip4/Spo16), that in many species is crucial during prophase I of meiosis (De Muyt et al, 2018). However, it remains unknown whether the same applies for human ZZS and M1AP.

Meiosis is the crucial process during spermatogenesis that leads to the formation of haploid germ cells. A key step during meiosis is homologous recombination, which is required for accurate chromosome segregation and the formation of haploid gametes. Homologous recombination, which also enables genetic diversity, occurs via crossovers between homologous chromosomes (chiasmata). Briefly, the process is initiated by programmed DNA double-strand breaks (DSB) in early prophase I (Fig. 1A). At the end of this phase, the DSBs are repaired and resolved as either non-crossovers or crossovers. When at least one crossover per homologous chromosome is formed, correct segregation can ensue; this is the *obligatory crossover* principle (reviewed in De Massy, 2013; Xie et al, 2022; Bolcun-Filas and Handel, 2018; Gray and Cohen, 2016).

In humans, there are approximately 50 crossovers per spermatocyte, which translates to one to five crossovers per pair

[1]Centre of Medical Genetics, Institute of Reproductive Genetics, University of Münster, 48149 Münster, Germany. [2]School of BioSciences and Bio21 Molecular Sciences and Biotechnology Institute, Faculty of Science, University of Melbourne, Parkville, VIC 3010, Australia. [3]Institute of Veterinary Anatomy, Histology and Embryology, University of Gießen, 35392 Gießen, Germany. [4]Clinic and Polyclinic for Urology, Paediatric Urology and Andrology, University Hospital Gießen, 35392 Gießen, Germany. [5]Department of Growth and Reproduction, University Hospital Copenhagen, 2100 Copenhagen, Denmark. [6]Centre of Reproductive Medicine and Andrology, University of Münster, 48149 Münster, Germany. [7]Department of Medical Genetics, University Hospital Münster, 48149 Münster, Germany. [8]Department of Cellular and Molecular Medicine, Faculty of Health and Medical Sciences, University of Copenhagen, 2100 Copenhagen, Denmark. ✉E-mail: corinna.friedrich@ukmuenster.de

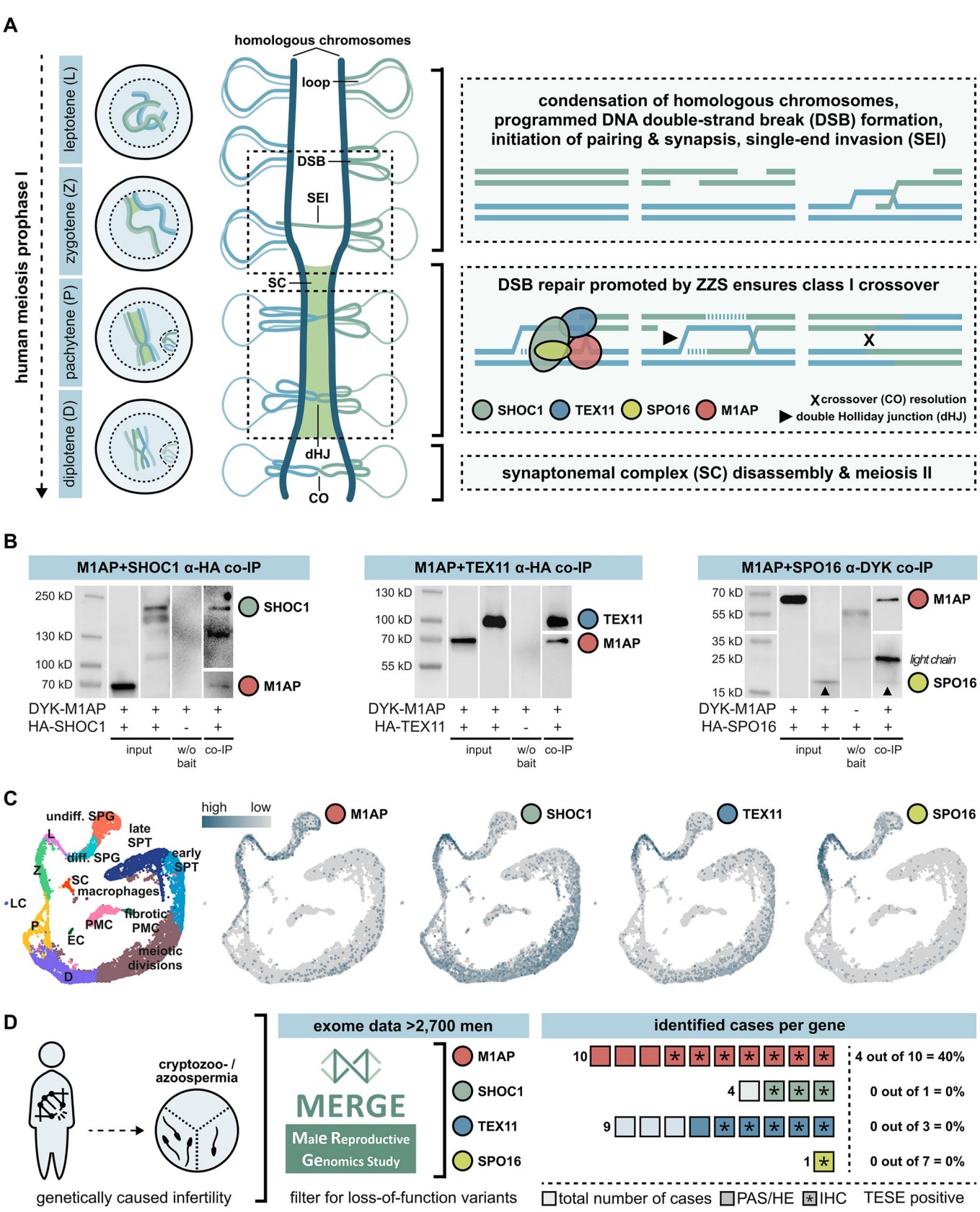

**Figure 1. Human meiotic recombination in men depends on ZZS function—and M1AP interacts with all three complex components.**

(A) Schematic representation of human prophase I and meiotic recombination. To simplify, the global arrangement of just one pair of homologous chromosomes is depicted and homologues are differentiated with two colours (blue, green). The molecular mechanisms of class I crossover resolution are represented in a simplified manner within the dotted boxes. In leptotene (L), homologous chromosomes duplicate, condense, and align. DNA double-strand breaks (DSBs) are initiated. During zygotene (Z), homologous chromosomes pair up and the initiation of synapsis is supported by the dynamic assembly of a ladder-like structure—the synaptonemal complex (SC). This complex provides a structural basis for meiotic recombination and in pachytene (P), the chromosomes are fully synapsed. DSB repair results in at least one crossover (CO) per chromosome pair (= obligatory crossover principle) and highly depends on SHOC1, TEX11, and SPO16 (= ZZS complex) activity. An integration of M1AP in this process was shown in mice (Li et al, 2023). Cells have completed DSB repair in diplotene (D), the SC disassembles, and homologues are physically connected by chiasmata. (B) Co-immunoprecipitation (IP) proved the interaction of human M1AP (detected by N-terminal DYK-tag) with each of the ZZS complex proteins (detected by C-terminal HA tag) co-transfected in HEK293T cells. Input and prey proteins served as positive and negative controls. Respective co-IP Western blot panels can be read from left to right as following: 1st lane = marker, 2nd = co-transfection of both plasmids, detection of M1AP in input sample, 3rd = co-transfection of both plasmids, detection of ZZS in input sample, 4th = transfection of pure prey protein and detection after immunoprecipitation (negative control), 5th = co-transfection of both plasmids, detection of both proteins from co-IP sample. In the last panel, the upturned arrowhead indicates the faint bands of SPO16 for better visualisation. Due to the low detection signal of SPO16, antibody chains of the co-IP specific anti-DYKDDDDK antibody, detected with anti-mouse HRP secondary antibody, are visible too. Experiments were conducted in three independent biological replicates. (C) Uniform manifold approximation and projection (UMAP) plot of obstructive azoospermic controls ($N = 3$) adapted from Di Persio et al, 2021. Through mRNA expression profiling, individual stages of human spermatogenesis were clustered and visualised. Expression data of human M1AP, SHOC1, TEX11, and SPO16 mRNA was compiled by querying the dataset. (D) For each gene, variants leading to dysfunctional proteins were searched for in the MERGE study dataset to associate a deficient genotype with a distinct phenotype. Overall, ten men with LoF variants in M1AP, four men with LoF variants in SHOC1 ($N = 3$ from MERGE and $N = 1$ from GEMINI), nine men with LoF variants in TEX11, and one man with a LoF variant in SPO16 were selected. If possible, material aiming for testicular sperm extraction (TESE) was used for subsequent analyses including periodic acid-Schiff (PAS) or haematoxylin and eosin (H&E) staining (both highlighted in bolder colour), immunohistochemical staining (IHC = *), and spermatocyte spreads (one sample per gene of interest). A positive TESE result was only seen in men with LoF variants in M1AP ($N = 4$). Source data are available online for this figure.

of homologous chromosomes (Sun et al, 2005). During meiosis, meiotic recombination is mediated by a subset of highly conserved proteins of the ZMM family (an acronym for the yeast proteins Zip1/Zip2/Zip3/Zip4, Msh4/Msh5, Mer3, and Spo16; reviewed in Pyatnitskaya et al, 2019), including ZZS proteins. In particular, these proteins assemble within a meiosis-specific structure, the synaptonemal complex (SC), support the synapsis of homologous chromosomes and stabilise recombination intermediates (reviewed in Zickler and Kleckner, 2015). Only when the intermediates are stabilised by ZZM, the DSBs can be resolved as class I crossovers (Börner et al, 2004).

Even minor errors in this tightly coordinated interplay can lead to meiotic arrest and infertility (Xie et al, 2022). Accordingly, genetic variants in each of the ZZS genes have already been linked to meiotic arrest in humans, with the typical phenotypes being NOA in men and primary/premature ovarian insufficiency (POI) in women. The ZZS gene TEX11 is a well-established X-linked gene for clinical diagnostics in male infertility (Wyrwoll et al, 2023b; Yatsenko et al, 2015); biallelic pathogenic variants in SHOC1 lead to infertility in men and women (Krausz et al, 2020; Ke et al, 2023); and one homozygous splice region variant in SPO16 has recently been associated with POI (Qi et al, 2023). While a strong association between M1AP and male infertility has been demonstrated (Wyrwoll et al, 2020; Li et al, 2023), the protein's molecular function remained unexplored in humans.

In this study, we identified that human M1AP interacts in vitro with each of the ZZS proteins. Additionally, we present the first man homozygous for a loss-of-function (LoF) variant in SPO16. The testicular phenotype of this man and men with LoF variants in the other ZZS genes SHOC1 and TEX11 shared an early prophase I arrest, including asynapsis and unrepaired DSBs. In contrast, LoF variants in M1AP led to a metaphase I arrest, where early meiotic recombination was completed while the total number of recombination intermediates and the final recombination products, the meiotic class I crossover events, was reduced. Ultimately, this leads to the loss of qualitatively intact haploid germ cell progression, but fertilisation-competent spermatozoa have on rare occasions been

retrieved by testicular sperm extraction. This demonstrates that M1AP is an important but not essential functional enhancer in the complex network of meiotic recombination. Collectively, these genotype-specific differences have important clinical implications, as they can be used to guide evidence-based treatment decisions or counselling of couples with male-factor infertility.

## Results

### Human M1AP interacts with the ZZS proteins SHOC1, TEX11, and SPO16 in vitro and shares a similar mRNA expression profile

To assess the interaction between the ZZS proteins and M1AP in humans, we performed co-immunoprecipitation (IP). Human DYK-tagged M1AP was co-expressed with human HA-tagged SHOC1, TEX11, or SPO16 in HEK293T cells. Proteins were immunoprecipitated from cell lysates by tag-specific antibodies. A subsequent Western blot showed that human M1AP binds specifically to each of the three human ZZS complex components (Fig. 1B), showing that M1AP interacts with the ZZS complex also in humans. To specify the interaction between M1AP and the ZZS complex, we immunoprecipitated different truncated M1AP proteins (Fig. EV1A) in conjunction with full-length TEX11. However, in subsequent Western blot analysis no evidence of interaction between any of the truncated M1AP proteins and full-length TEX11 was observed (Fig. EV1B,C) and thus the determination of a specific TEX11 binding domain in M1AP was not possible. Remarkably, the protein-protein interaction of M1AP and TEX11 is already abolished by the loss of the last 97 amino acids.

Assuming a closely related function of all four proteins, ZZS and M1AP, in human meiosis, we aimed to investigate whether all four genes share a similar testicular expression profile. Thus, previously published single-cell RNA sequencing data of human testicular tissue (Di Persio et al, 2021) were queried (Fig. 1C). Overall, the analysis showed that all four genes are similarly expressed in all

Table 1. Genetic and testicular characteristics of infertile men with loss-of-function variants in *M1AP* or the ZZS genes

| Case | Genetic variant (MAF gnomAD v4.1.0) | Semen analysis | Arrest type—most advanced germ cell type | TESE/MAR outcome | Reference |
|---|---|---|---|---|---|
| **M1AP—autosomal-recessive inheritance, NM_138804.4** | | | | | |
| M330 | c.[676dup];[676dup] p.[Trp226Leufs*4];[Trp226Leufs*4] (MAF = 0.003327) | Azoospermia | Metaphase I arrest — elongated spermatids | Negative | Wyrwoll et al, 2020 |
| M864 | | Azoospermia | Metaphase I arrest — round spermatids | Negative | Wyrwoll et al, 2020 |
| M1792 | | Azoospermia | NA | Negative | Wyrwoll et al, 2020, Nagirnaja et al, 2022 |
| M2062 | | Cryptozoospermia | Metaphase I arrest — elongated spermatids | Positive, ICSI failed | Wyrwoll et al, 2020, 2023 |
| M2525 | | Azoospermia | Metaphase I arrest — round spermatids | Negative | |
| M2746 | | Cryptozoospermia | Metaphase I arrest — elongated spermatids | Positive, ICSI successful | |
| M2747 | | Cryptozoospermia | Metaphase I arrest — elongated spermatids | Negative | |
| M3402 | | Azoospermia | Metaphase I arrest — elongated spermatids | Positive, ICSI failed | |
| M3511 | | Azoospermia | Metaphase I arrest — elongated spermatids | Positive, ICSI failed | |
| M3609 | c.[1073_1074+10del];[1073_1074+10del] p.[(Leu358Glnfs*58)];[(Leu358Glnfs*58)] (MAF = 0.00006815) | Azoospermia | Metaphase I arrest — round spermatids | Negative | |
| **SHOC1—autosomal-recessive inheritance, NM_173521.5** | | | | | |
| M2012 | c.[1085_1086del];[1085_1086del] p.[(Glu362Valfs*25)];[(Glu362Valfs*25)] (MAF: absent from gnomAD) | Azoospermia | NA | NA | Krausz et al, 2020 |
| G-377 | c.[1085_1086del];[1085_1086del] p.[(Glu362Valfs*25)];[(Glu362Valfs*25)] (MAF: absent from gnomAD) | Azoospermia | Meiotic arrest— spermatocytes | Negative | Nagirnaja et al, 2022 |
| M2046 | c.[1351del;1347 T > A];[945_948del] p.([Ser451Leufs*23;Cys449*];[Glu315Aspfs*6]) (MAF: absent; absent; 0.00001860) | Azoospermia | Meiotic arrest— spermatocytes | Negative | Krausz et al, 2020 |
| M3260 | c.[1939 + 2 T > C];[1939 + 2 T > C] p.? (MAF: 0.0000006309) | Azoospermia | Meiotic arrest— round spermatids | Negative | |
| **TEX11—X-linked inheritance, NM_001003811.2** | | | | | |
| M205 | c.[1837 + 1 G > C];[0] p.? (MAF: absent from gnomAD) | Azoospermia | Meiotic arrest— spermatocytes | Negative | Yatsenko et al, 2015 |
| M246 | c.[22del];[0] p.(Ser8Profs*31) (MAF: absent from gnomAD) | Azoospermia | Meiotic arrest— spermatocytes | Negative | |
| M281 | c.[792 + 1 G > A];[0] p.? (MAF: absent from gnomAD) | Azoospermia | Meiotic arrest— spermatocytes | Negative | Yatsenko et al, 2015 |
| M1390 | c.[(204 + 1_205-1)_(737 + 1_738-1)del];[0] p.? (MAF: absent from gnomAD SVs v2.1) | Azoospermia | Meiotic arrest— spermatocytes | Negative | Wyrwoll et al, 2023 |
| M2739 | c.[1052dup];[0] p.(Ser352Valfs*14) (MAF: absent from gnomAD) | Azoospermia | Meiotic arrest— NA | Negative | |
| M2820 | c.[(-157_-99 + 1)_(738-1_792 + 1)del];[0] p.? (MAF: absent from gnomAD SVs v2.1) | Azoospermia | Meiotic arrest— NA | Negative | |

**Table 1.** (continued)

| Case | Genetic variant (MAF gnomAD v4.1.0) | Semen analysis | Arrest type—most advanced germ cell type | TESE/MAR outcome | Reference |
|------|------|------|------|------|------|
| M2942 | c.[1245 G > A];[0]<br>p.(Trp415*)<br>(MAF: absent from gnomAD) | Azoospermia | NA | NA | |
| M3152 | c.[(652-1_737 + 1)_(738-1_792 + 1)];[0]<br>p.?<br>(MAF: absent from gnomAD SVs v2.1) | Azoospermia | NA | NA | |
| M3409 | c.[731 G > A];[0]<br>p.(Trp244*)<br>(MAF: absent from gnomAD) | Azoospermia | Meiotic arrest—spermatocytes | Negative | |
| SPO16—autosomal-recessive inheritance, NM_001012425.2 | | | | | |
| M3863 | c.[266del];[266del]<br>p.[(Leu89Trpfs*18)];[(Leu89Trpfs*18)]<br>(MAF = 0.000003724) | Azoospermia | Meiotic arrest—spermatocytes | Negative | |

*fs* frameshift, * premature stop codon, *NA* not available, *TESE* testicular sperm extraction, *MAR* medically assisted reproduction.

stages of meiosis with strong mRNA abundance during early prophase I (leptotene, zygotene). *SHOC1* and *TEX11* showed an additional increase of expression during later meiotic divisions.

## Composition of the study cohort with men carrying LoF variants in *M1AP* or ZZS

To compare the protein-related impact of each of the four proteins on human meiosis, we selected cases carrying LoF variants in either *M1AP* (NM_138804.4) or one of the ZZS components encoding genes, *SHOC1* (NM_173521.5), *TEX11* (NM_001003811.2), and *SPO16* (NM_001012425.2) from our Male Reproductive Genomics (MERGE) study cohort. In addition, we included one published case with a variant in *SHOC1* from the GEMINI cohort (Nagirnaja et al, 2022; G-377). Both previously published (N = 10) and novel (N = 14) cases were considered for functional in-depth analysis.

We compared the phenotypes of ten unrelated men with homozygous LoF variants in *M1AP*, four with biallelic LoF variants in *SHOC1*, nine with hemizygous LoF variants in *TEX11*, and one man with a homozygous LoF variant in *SPO16* (Fig. 1D; Table 1). Nine of the men with a LoF variant in *M1AP* carried the recurrent pathogenic frameshift variant c.676dup, of which four had already been published and the variant has been functionally analysed (Wyrwoll et al, 2020) while one man (M3609) carried a homozygous *M1AP* splice site variant (c.1073_1074+10del). This variant was predicted to undergo nonsense-mediated decay. The functional significance was analysed in vitro by a minigene assay which demonstrated aberrant splicing, which presumably results in a premature stop codon (Fig. EV2).

For *SHOC1*, we selected two men (M2012, G-377) carrying the same homozygous frameshift variant (c.1085_1086del) and one case (M2046) with confirmed compound heterozygous variants (c.[1351del;1347 T > A];[945_948del]), who were previously described (Table 1; Krausz et al, 2020; Nagirnaja et al, 2022). In addition, a novel splice site variant (c.1939+2 T > C) was identified in one man with predominant meiotic arrest (M3260) (Fig. EV3). The in vitro minigene assay showed aberrant splicing resulting in an in-frame exon skipping event involving only ~4% of the total protein (Fig. EV3A–C). This may not lead to a complete loss of but only to a reduced protein function. This deletion affects the distant

helicase hits region but not the highly conserved 'SHOC1 homology region', which is important for the interaction with the other ZZS proteins (Macaisne et al, 2008).

The cohort of nine men with hemizygous LoF variants in *TEX11* comprised three men with different partial gene deletions, of which two were inherited from the mother (Appendix Figs. S1 and S2), two men with splice site variants, and four men with variants encoding premature stop codons (Table 1); three of these cases have already been described but not in this detail (Yatsenko et al, 2015; Wyrwoll et al, 2023b).

One man (M3863) carried a novel homozygous frameshift variant (c.266del p.(Leu89Trpfs*15)) in the highly conserved ZZS gene *SPO16* (Table 1). This variant was predicted to induce a premature stop codon in exon 4 and to truncate >40% of the complete protein. Most of the protein's central domain and the entire functional helix-hairpin-helix (HhH) domain would be affected by the truncation (Fig. EV4). According to the prediction, it was more likely that the mRNA was degraded by nonsense-mediated decay. In both cases, a loss of protein function can be expected.

Semen analysis of all 24 men included in this study revealed a high proportion of azoospermia (N = 21), with the remaining three men displaying cryptozoospermia. Interestingly, all three crypto-zoospermic men were homozygous for the recurrent LoF variant in *M1AP* (c.676dup). Clinical features were similar between men, including normal testicular volumes, normal luteinising hormone, follicle-stimulating hormone in the high range, and normal testosterone (Table EV1; Appendix Fig. S3).

## Men with LoF variants in *M1AP* or ZZS showed gene-specific spermatogenic defects

Testicular biopsies originating from testicular sperm retrieval (TESE) procedures were available in 20 of the selected cases and used for histological evaluation of spermatogenic differentiation. Analysis using periodic acid-Schiff (PAS, N = 14) or haematoxylin and eosin (H&E, N = 6) staining determined a common testicular phenotype of predominant spermatocyte arrest in all cases (Human Phenotype Ontology [HPO] term: HP:0031039, Fig. 2A; Appendix Figs. S4 and S5). In addition, we observed cells in diakinesis or

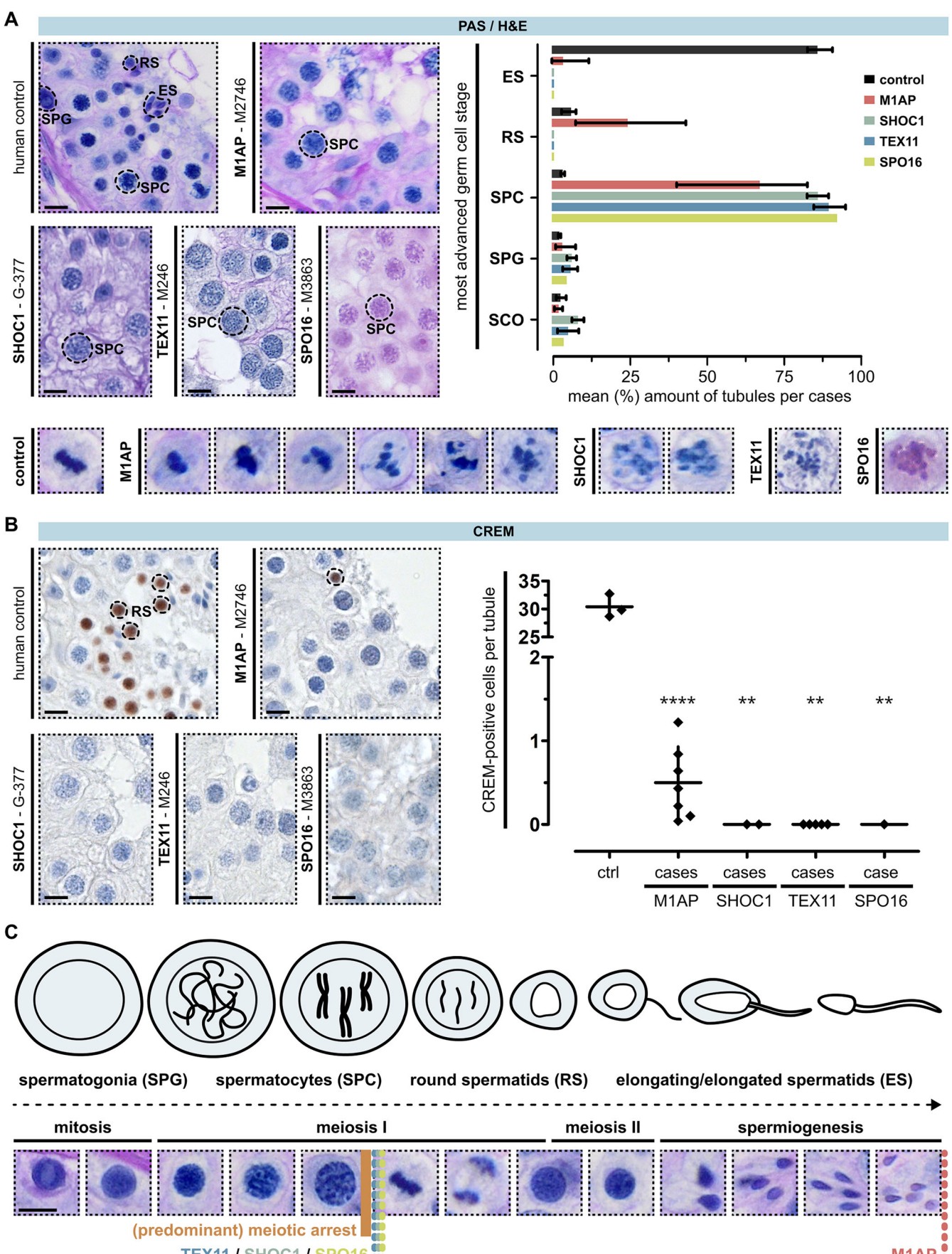

**Figure 2.   Testicular phenotyping of men with loss-of-function variants in *M1AP*, *SHOC1*, *TEX11*, and *SPO16*.**

(A) PAS staining of testicular tissue with full spermatogenesis and infertile men with LoF variants in genes encoding the ZZS proteins and *M1AP*. One representative case per gene is depicted (M1AP: M2746 (also shown in Appendix Fig. S6), SHOC1: G-377, TEX11: M246, SPO16: M3863). For all men, the most advanced germ cell type per tubule was assessed and is shown in the representative image and the bar graph. Data are presented as mean (%) and range, quantifying one biopsy per case (CTRL = 3 cases, M1AP = 7, SHOC1 = 2, TEX11 = 5, SPO16 = 1). In addition, intact (control, M1AP) metaphase and aberrant (M1AP, SHOC1, TEX11, SPO16) metaphase-like cells were observed (detail view). (B) Haploid germ cells were analysed by CREM localisation. Only men with LoF variants in *M1AP* had CREM-positive round spermatids. Compared to the controls, the total amount of these cells was significantly reduced. *P* values were determined between the control and M1AP group by an unpaired two-tailed *t* test (****$P$ = 1.9765*10$^{-10}$) or between the control and SHOC1, TEX11 or SPO16 group per one sample two-tailed *t* test (**$P$ = 0.0016). Data is presented as the mean and standard deviation, quantifying one biopsy per case (CTRL = 3, M1AP = 7, SHOC1 = 2, TEX11 = 5, SPO16 = 1). (C) Schematic illustration and corresponding PAS staining of human spermatogenesis. Coloured dotted lines represent the identified gene-specific germ cell arrest. SPG spermatogonia, SPC spermatocytes, RS round spermatids, ES elongated spermatids. Figure EV3E also shows part of the panel. The scale bar represents 10 μm. Source data are available online for this figure.

metaphase I stage characterised by scattered and misaligned chromosomes (Fig. 2A, detail view). Of note, we only saw metaphase cells with correctly aligned chromosomes in controls and in men with LoF variants in *M1AP*.

To analyse the presence of post-meiotic germ cells, we performed immunohistochemical staining for cAMP-responsive element modulator (CREM), a marker protein for round spermatids (Weinbauer et al, 1998). Tissue of 16 men was available for subsequent analysis (Fig. 2B; Appendix Fig. S6). In fertile control samples ($N$ = 3), CREM-positive spermatids were observed in almost all tubules with an average of 30.41 ± 2.08 spermatids per tubule. In all analysed men with LoF variants in *M1AP* ($N$ = 7), CREM-positive spermatids were observed however in significantly lower numbers than controls with an average of 0.50 ± 0.43 cell/tubule. Indeed, some but not all seminiferous tubules were found to contain round spermatids, and elongated spermatids were only observed in six of these cases. In contrast, CREM-positive round spermatids were only observed in one man with a splice site variant in *SHOC1* (M3260). This variant does presumably not cause a complete loss of protein function but rather appears to compromise SHOC1 function, leading to a notable reduction in the progression of spermatogenesis. Consequently, round spermatids were present in a limited number of tubules (Fig. EV3D). In total, we detected an average of 0.16 CREM-positive spermatids per tubule in M3260, which is still less than in most cases with LoF variants in *M1AP*. The specific type of arrest is described as metaphase I arrest with rare round spermatids (Fig. EV3E). In addition, we detected no CREM-positive round spermatids in cases with LoF variants in *SHOC1* or *TEX11*. The single case with a homozygous LoF in *SPO16* (M3863) exhibited complete meiotic arrest without CREM-positive round spermatids (Fig. 2B; Appendix Fig. S6). These results demonstrate a different testicular phenotype between men with LoF variants in *M1AP* and men affected by LoF variants in key components of the ZZS complex (Fig. 2C).

## Proper chromosome synapsis and XY body formation despite M1AP deficiency

To assess whether the LoF variants in *M1AP* and in ZZS genes led to the impaired production of post-meiotic cells because of alterations in meiotic recombination, we stained patients' testicular tissue for the two key meiosis markers γH2AX and H3S10p. Phosphorylated histone variant H2AX (γH2AX) is a marker for DSBs and was used to analyse DSB repair and correct synapsis of homologous chromosomes. In autosomes, DSBs are repaired once the homologues have successfully

aligned and synapsed, and, consequently, γH2AX staining disappears. In parallel, when meiotic sex chromosome inactivation (MSCI) takes place during pachytene, a condensed chromatin domain is formed, termed the XY body. Here, γH2AX accumulates independent of meiotic recombination-associated DSBs (Fernandez-Capetillo et al, 2003). Together, the DSB repair in autosomes and the distinct formation of the XY body suggest that cells are passing through the pachytene checkpoint (Hamer et al, 2003).

The distinct γH2AX localisation during meiosis I prophase and metaphase is shown in a human control (Fig. 3A). Men with LoF variants in *M1AP* were characterised by a qualitatively equivalent staining pattern, with quantitatively reduced amounts of pachytene- and increased amounts of zygotene-like cells (Fig. 3A, detail view, bar graph, Appendix Fig. S7). Interestingly, the complete repair of DSBs post-zygotene and the clear restriction of γH2AX to the XY body was only observed in controls and in men with LoF variants in *M1AP*. In men with LoF variants in *SHOC1*, *TEX11*, or *SPO16*, most of the cells maintained a zygotene-to-pachytene-like stage with enlarged nuclei, accumulated DSBs, and aberrant γH2AX localisation (Fig. 3A, detail view, bar graph, Appendix Fig. S8). Thus, an arrest in these men occurs between zygotene- to early pachytene-like (Z*/P*), referring to a human type I meiotic (Jan et al, 2018) or pachytene arrest (Enguita-Marruedo et al, 2019). Remarkably, even though cells do not proceed properly beyond zygotene-pachytene-like stage in the majority of men with LoF variants in *SHOC1*, *TEX11*, or *SPO16*, γH2AX-positive metaphase-like or diakinesis cells (M-I*) were seen in most of the cases ($N$ = 8). These cells were characterised by scattered chromosomes, incorrectly aligned at the metaphase plate or maintained in a diakinesis stage. While similar cells were seen in men with LoF variants in *M1AP*, we also observed in parallel γH2AX-negative, intact metaphase cells with properly aligned chromosomes ($N$ = 7).

In addition, we used phosphorylated histone 3 (H3S10p) to mark cells reaching the metaphase stage (Song et al, 2011). H3 is phosphorylated when the DNA is condensed during mitosis and meiosis for subsequent segregation of the chromosomes. Localisation of H3S10p in all cases with LoF variants in *M1AP* showed that the cells with properly aligned chromosomes are meiotic metaphase cells (Appendix Fig. S9). This, together with the presence of post-meiotic cells, classified the arrest in men with *M1AP* LoF variants as a partial metaphase I arrest (Enguita-Marruedo et al, 2019). In accordance with the γH2AX staining, localisation of H3S10p in men with LoF variants in *SHOC1*, *TEX11*, or *SPO16* revealed only aberrant metaphase-like/diakinesis stage (M-I*) or mitotic spermatogonia (Appendix Fig. S9). To verify whether the observed

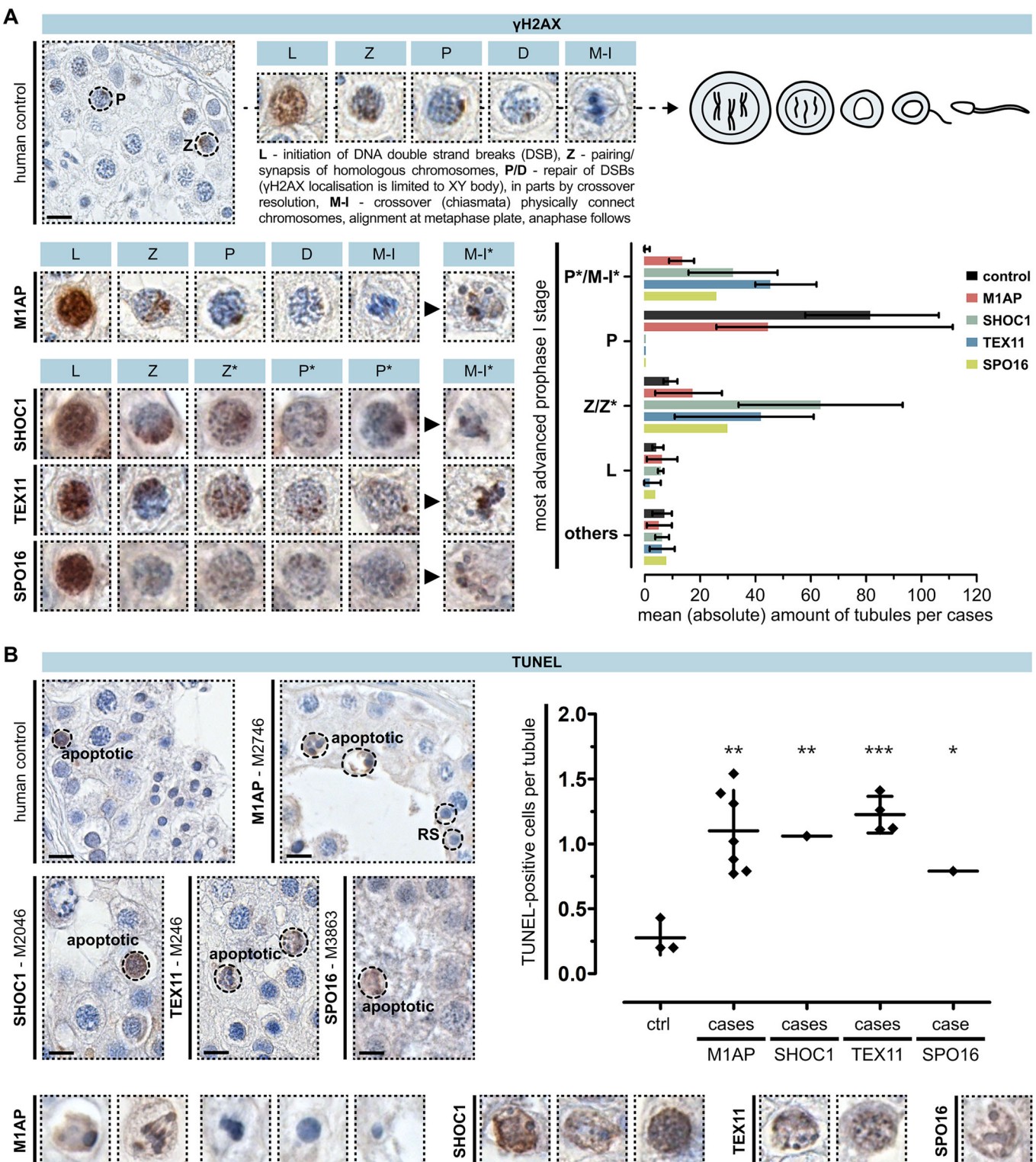

metaphase I-like cells were spermatocytes rather than spermatogonia, we conducted immunostaining for H3S10p, γH2AX, and MAGEA4 (the latter being a marker for spermatogonia) on sequential tissue sections from a human control sample and three representative cases with homozygous LoF variants in *M1AP*

(M864, M2062, M2746). Three sections were positioned on one slide with a spacing of 3 μm between each one. Our findings indicated that metaphase I (or metaphase I-like) cells were either positive for H3S10p or γH2AX, but not for MAGEA4, or vice versa (Appendix Fig. S10).

**Figure 3.    Investigation of meiotic recombination and arrest in men with loss-of-function variants in *M1AP*, *SHOC1*, *TEX11*, and *SPO16*.**

(A) Localisation of γH2AX in a control demonstrates the marker's specific staining pattern during each substage of meiosis prophase I. Analysis of this marker in the infertile men revealed genotype-specific aberrations and meiotic arrest: M1AP = partial metaphase (M-I) arrest, SPO16/TEX11/SHOC1 = zygotene- (Z*) to early pachytene-like (P*) arrest, with occasional metaphase-like cells (M-I*). Data are presented as mean and range, quantifying one biopsy per case (CTRL = 3 cases, M1AP = 7, SHOC1 = 2, TEX11 = 5, SPO16 = 1. (B) TUNEL assay showed increased apoptosis in patients independent of the genetic background (dot plot). Most TUNEL-positive cells already showed hallmarks of apoptosis (detail view), and only men with LoF variants in M1AP showed TUNEL-negative metaphase cells with correctly aligned chromosomes, round, and elongated spermatids. *P* values were determined between the control and M1AP or TEX11 group by an unpaired two-tailed *t* test (**P* = 0.0026, ***P* = 0.0003, respectively) or between the control and SHOC1 or SPO16 group per one sample two-tailed *t* test (**P* = 0.0096, **P* = 0.0219, respectively). Data are presented as the mean and standard deviation, quantifying one biopsy per case (CTRL = 3 cases, M1AP = 7, SHOC1 = 1, TEX11 = 4, SPO16 = 1). The scale bar represents 10 μm. Source data are available online for this figure.

## Increased apoptosis leads to a loss of germ cells

Maintenance of γH2AX localisation in metaphase I cells indicates a failure of meiotic recombination and checkpoint-induced apoptotic events (Enguita-Marruedo et al, 2019). Thus, we quantified apoptotic DNA fragmentation via TUNEL assay. Compared to controls, we observed a significant increase in apoptosis for all patients analysed (Fig. 3B; Appendix Fig. S11). Most TUNEL-positive cells showed hallmarks of apoptosis, namely condensation, fragmentation, and apoptotic bodies (Saraste and Pulkki, 2000). In addition, aberrant metaphase I-like cells with misaligned chromosomes or diakinesis-like stage cells (M-I*) were also TUNEL-positive (Fig. 3B, detail view). In cases with LoF variants in *SHOC1*, *TEX11*, and *SPO16*, apoptotic DNA fragmentation was also found in rare pre-metaphase spermatocytes. One key difference between the cases was that only men with LoF variants in *M1AP* exhibited TUNEL-negative metaphase cells, round, and even elongated spermatids.

## Genotype-specific meiotic recombination failure due to ZZS or M1AP deficiency

To address whether meiotic recombination was successfully completed in spermatocytes from men with LoF variants in *M1AP*, allowing them to differentiate into post-meiotic haploid cells, we analysed spermatocyte spreads of one man (M864). In comparison, we investigated the recombination failure in one man each with a LoF variant in *SHOC1* (M2046), *TEX11* (M3409), or *SPO16* (M3863). To indicate whether the synaptonemal complex (SC) formed properly, we stained for SYCP1 and SYCP3-containing elements of the SC that assemble along unaligned chromosome axes (Yuan et al, 2000). For crossover formation to occur, homologous chromosomes must undergo full synapsis, which is fulfilled when the transverse filament protein of the SC, SYCP1, localises between the axes (Dunce et al, 2018). In addition, to determine the progression of prophase I, we stained for γH2AX. For a better orientation, we marked the centromeres using a human anti-centromere antibody (ACA).

Similar to the control, we observed spermatocytes of M864 (LoF variant in *M1AP*) in the pachytene-like stage, including properly formed chromosome axes (SYCP3), fully synapsed homologues (SYCP1) (Fig. 4), In contrast, in men with SHOC1, TEX11, or SPO16 deficiency, SYCP1 assembly was almost absent in M2046 (SHOC1) and only slightly visible in M3409 (TEX11) and M3863 (SPO16) while the chromosome axis (SYCP3) was regular (Fig. 5).

In control and M1AP-deficient spermatocytes resolved DSBs of autosomes and a defined XY body were observed (γH2AX, Fig. 6). In contrast, in all three cases with ZZS deficiency, γH2AX

accumulated around the SYCP3-positive axes (Fig. 6). Thus, in ZZS deficiency no pachytene-like spermatocytes were identified.

In addition, we monitored the impact of the LoF variants in M1AP and the ZZS genes on the formation of recombination intermediates by staining the recombinase RAD51 at different prophase I substages. Together with DMC1, this protein promotes single-strand invasion into homologous chromosomes (Shinohara et al, 1992). In the control, distinct RAD51 foci were observed in leptotene, peaked in zygotene, were less visible in early pachytene and largely absent in later pachytene stages (Fig. 4). Similarly, in men with M1AP or ZZS deficiency, RAD51 foci were observed in leptotene-like stages and peak in zygotene-like stages (M864 for M1AP, Fig. 4; M2046, M3409, M3863 for ZZS, Fig. 5). In contrast to the control, RAD51 foci were visible throughout all observed prophase I stages.

Aberrant RAD51 kinetics could impact downstream recombination steps. To analyse stabilisation of recombination intermediates and the correct recruitment of the ZZS complex, we labelled TEX11 and quantified the corresponding foci. In the control, TEX11 co-localises with the synaptonemal complex and is visible as distinct foci along fully synapsed chromosome axes in the pachytene stage. TEX11 was completely absent in M3409 (TEX11) and highly reduced and disorganised in M2046 (SHOC1) and M3863 (SPO16; Fig. 6A). In contrast, M864 (M1AP) showed significantly reduced but proper TEX11 recruitment to the chromosome axes of pachytene cells (Fig. 6A). Ultimately, the number of TEX11 foci was reduced from 199.88 ± 21.56 (control spermatocytes) to 66.00 ± 7.85 in M864 (M1AP) (Fig. 6B).

Recombination intermediates, which are designated for class I crossover formation, are marked by the heterodimer MSH4/MSH5 (Lynn et al, 2007). To explore if crossover formation is disrupted in M1AP and ZZS deficiency, we targeted MSH5 and quantified foci compared to the control (Fig. 7A). In control pachytene spermatocytes, an average of 53.70 ± 13.84 MSH5 foci was counted, which was significantly reduced to 28.60 ± 6.73 in pachytene spermatocytes of M864 (M1AP). Since no pachytene-like spermatocytes were detectable in men with ZZS deficiency, most progressed zygotene-like cells were analysed, where MSH5 foci were almost completely absent (Fig. 7B).

To validate if MSH5-marked recombination intermediates achieve resolution under M1AP deficiency and form class I crossovers, we visualised designated crossover sites by staining for MLH1, an endonuclease involved in the resolution of class I crossovers (Baker et al, 1996) (Fig. 8A). In accordance with the absence of pachytene cells in ZZS deficiency, no MLH1 foci, which localised to the chromosome axes, were observed. To quantify class I crossover, we counted MLH1 foci in pachytene cells in M1AP-

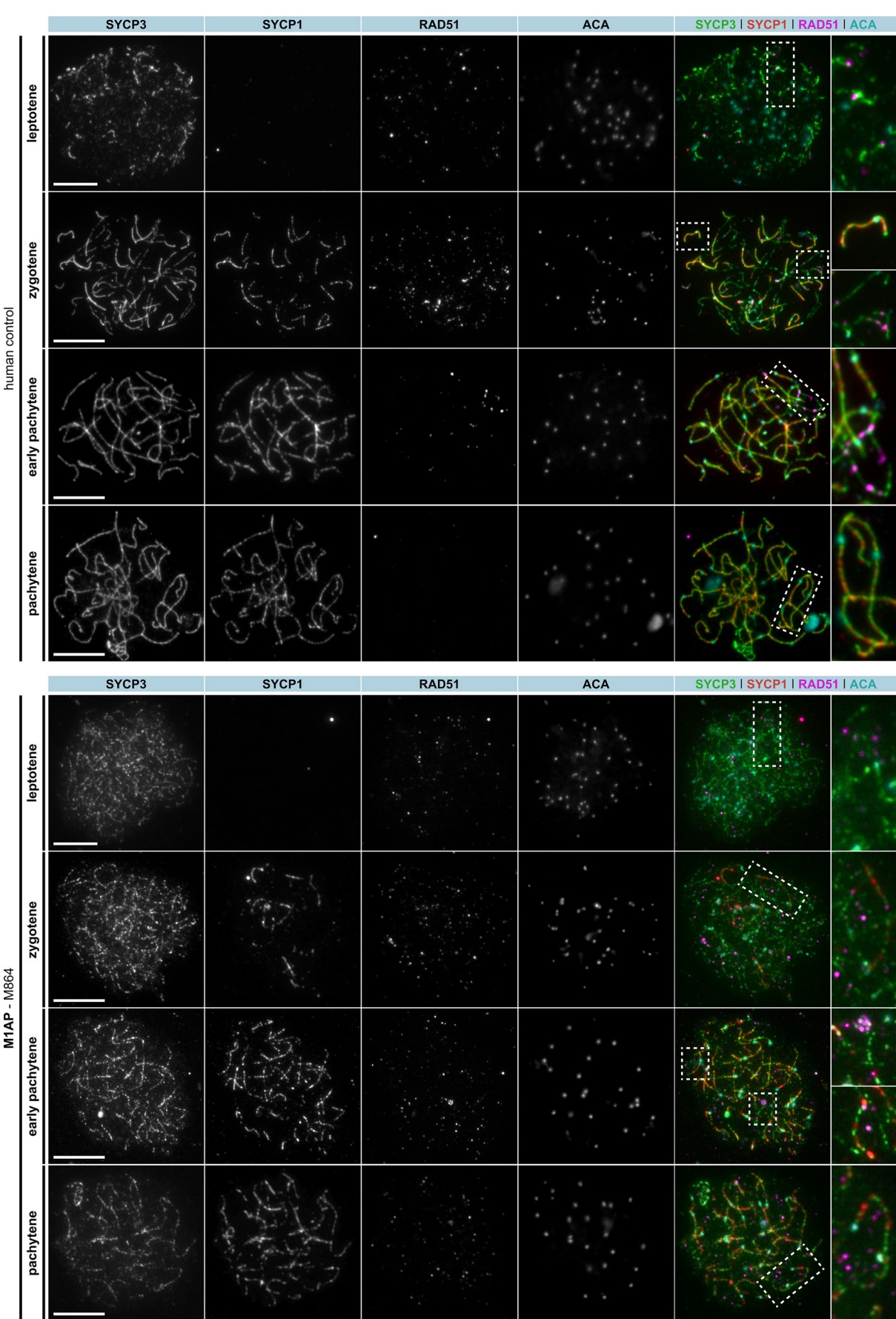

◄ **Figure 4.  Human spermatocyte spreads showed persistent RAD51 foci throughout prophase I in the man with loss-of-function variant in *M1AP*.**

Human spermatocyte spreads were stained for synaptonemal complex formation (= SYCP3, green + SYCP1, red), centromeric regions (= ACA, cyan), and single-strand invasion (RAD51 = magenta). The LoF variant in *M1AP* led to persistence of RAD51 foci during pachytene progression. The scale bar represents 10 μm. Source data are available online for this figure.

deficient spermatocytes, which were significantly reduced compared to the control. Specifically, M864 (M1AP) had an average of 35.00 ± 2.05 MLH1 foci per spermatocyte compared to 48.33 ± 2.55 foci per control spermatocyte (Fig. 8B).

### Sufficient class I crossover events occurred despite *M1AP* LoF variants

One crucial principle of meiotic recombination is the acquired number of crossovers per pair of homologous chromosomes, called the *obligatory crossover*. This means that each pair has to form at least one crossover, as it is crucial for the subsequent equilibrated segregation during meiosis I (reviewed in Wang et al, 2015). Taking advantage of super resolution structured illumination microscopy (SR-SIM), we investigated whether this principle was still fulfilled when M1AP was presumably dysfunctional (Fig. 9). From one representative spermatocyte of the man with a LoF variant in *M1AP* (Fig. 9A) MLH1 foci (dotted circles) were marked on each pair of homologous chromosomes to localise the sites of class I crossovers. In total, 36 MLH1 foci were seen in this example, and all paired chromosomes showed at least one MLH1 focus, ensuring the obligatory crossover principle in this set of designated chromosomes (Fig. 9B).

### Men with LoF variants in *M1AP* can father healthy children by MAR

Four of the ten men with LoF variants in *M1AP* tried to conceive via ICSI. Two men (M2062 and M2746) were counselled about their genetic diagnosis before trying MAR. So far, only one man, M2746, has successfully fathered a healthy child following ICSI with ejaculated sperm. In the first and second ICSI attempts, oocyte fertilisation was possible, but cells did not develop to the blastocyst stage or did not lead to a clinical pregnancy. In the third attempt, nine oocytes were fertilised, seven of which developed up to the two-pronuclei (2 N) stage. Two reached the blastocyst stage and were transferred to the female partner, after which one successfully implanted and developed normally, resulting in the birth of a healthy boy. Genome sequencing data of the child's genomic DNA compared to the father's demonstrated the paternity and the inheritance of the *M1AP* frameshift variant in a heterozygous state (Fig. 10A). In an ICSI-TESE attempt by M2062 and his female partner, three oocytes were fertilised, but none developed to the blastocyst stage and a transfer was not possible. For M3402 and his female partner, two ICSI attempts were not successful. For M3511, five spermatozoa were retrieved by TESE and used for ICSI; however, none of the five injected oocytes were fertilised.

To rule out any aneuploidies in the newborn boy, genome sequencing coverage data were analysed, and aberrations were excluded (Fig. EV5). Considering the exemplary case of M2746, ICSI with ejaculated spermatozoa or spermatozoa recovered from TESE offers the possibility of fatherhood for men with LoF variants in *M1AP* (Fig. 10B).

## Discussion

Indications that *M1AP* is required for mammalian spermatogenesis date back to 2006, when the mouse orthologue was demonstrated to be expressed in male germ cells during the late stages of spermatogenesis (Arango et al, 2006). This was strengthened by two subfertile mouse models with predominant meiotic arrest, hinting toward the protein's role during male meiosis (Arango et al, 2013; Li et al, 2023). In addition, several human cases of male infertility have been described in which azoo-, crypto-, or extreme oligozoospermia have been associated with biallelic variants in *M1AP* (Wyrwoll et al, 2020; Tu et al, 2020; Li et al, 2023; Khan et al, 2023; Appendix Table S1). Accordingly, *M1AP* has achieved a definitive clinical gene-disease association based on the criteria of the Clinical Genome Resource (ClinGen) Gene Curation Working Group, and, therefore, we have proposed that this gene be included in routine diagnostics of infertile men (Wyrwoll et al, 2023b; Stallmeyer et al, 2024). Recently, M1AP was proposed as a fourth component of the mouse ZZS complex, linking its role to three well-known and highly conserved meiosis-related proteins: SHOC1, TEX11, and SPO16 (Li et al, 2023). In our study, we analysed the interaction between M1AP and the human ZZS proteins and showed that each ZZS protein bound to M1AP independently. In addition, all four genes display a similar mRNA expression in early prophase I. These observations suggest that M1AP probably acts as a partner of the ZZS complex in human meiotic recombination similar to the mouse. Furthermore, no functional domains or binding sites for interacting proteins have been described for M1AP to date. We demonstrated that the deletion of the last 97 amino acids of M1AP (~18% of the protein) is already sufficient to completely abolish the binding of M1AP and TEX11 implying that for the protein-protein interaction the full-length protein of M1AP is mandatory.

The men with LoF variants in *M1AP* investigated in this study displayed a predominant meiotic arrest with occasional haploid germ cells. Our in-depth characterisation of the patients' testicular phenotype implied a partial metaphase I arrest, which was also seen in another reported man with a *M1AP* LoF variant (Li et al, 2023). In addition, we observed the activation of the spindle-assembly checkpoint, which is crucial for preventing premature chromosome separation and, thus, abnormal segregation and aneuploidies (reviewed in Lane and Kauppi, 2019). Accordingly, meiotic recombination was presumably intact in individual cells that progressed beyond this checkpoint. As such, an individual fertilisation-competent spermatozoon was allowed to develop, ultimately resulting in the birth of a healthy, euploid child.

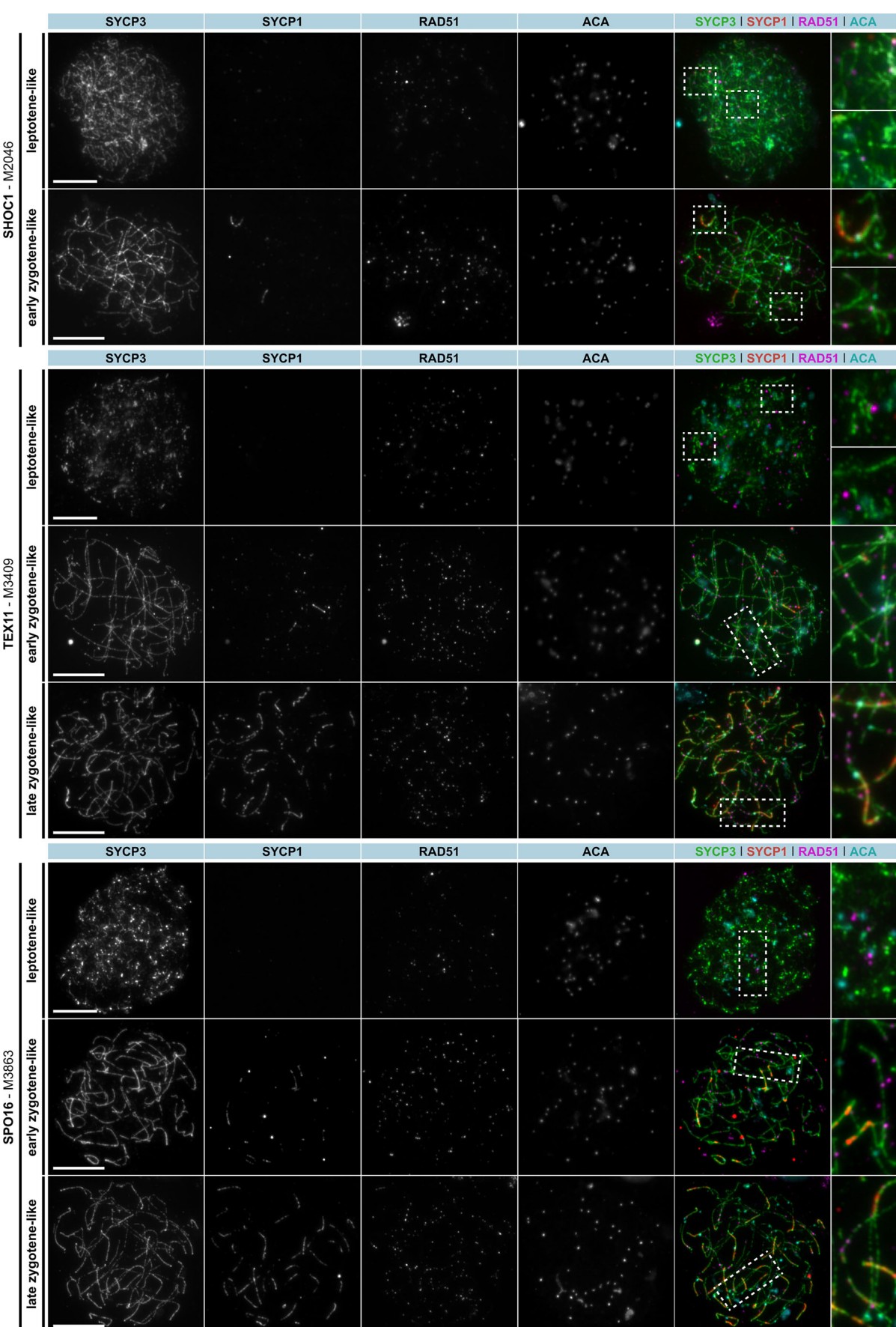

**Figure 5. Human spermatocyte spreads showed persistent RAD51 foci in men with loss-of-function variants in the ZZS genes throughout incomplete prophase I progression.**

Human spermatocyte spreads were stained for synaptonemal complex formation (= SYCP3, green + SYCP1, red), centromeric regions (= ACA, cyan), and single-strand invasion (RAD51 = magenta). LoF variants in *SHOC1*, TEX11, or *SPO16* led to persistence of RAD51 until late zygotene-like stage. The scale bar represents 10 μm. Source data are available online for this figure.

These findings in humans mimic the *M1ap* knockout mouse model, which was also characterised by male subfertility (Li et al, 2023) with lower sperm counts and an increased number of arrested metaphase I cells with unaligned chromosomes. Here, regular meiotic DSB formation and persistent single-strand invasion but fewer recombination intermediates and reduced class I crossover events led to metaphase I arrest and cellular apoptosis, and only a fraction of cells progressed to fertility-competent spermatozoa.

In humans, *SHOC1* and *TEX11*, which encode two of the M1AP binding partners in the meiotic ZZS complex, have also previously been associated with male infertility and meiotic arrest (Krausz et al, 2020; Yatsenko et al, 2015; Appendix Table S1), and both genes represent validated disease genes for male infertility (Wyrwoll et al, 2023b; Stallmeyer et al, 2024). In contrast to *M1AP*, hemizygous and biallelic LoF variants in these genes, respectively, have primarily been associated with a complete lack of haploid germ cells (Yatsenko et al, 2015; Krausz et al, 2020). However, the evaluation of the testicular phenotype in these initial reports was based on testicular overview staining, making a definitive conclusion difficult. In this study, we concordantly observed complete meiotic arrest and a lack of haploid, CREM-positive germ cells in all men we analysed with LoF variants in *TEX11*, pointing to an activation of the first, the pachytene, checkpoint (Yatsenko et al, 2015; Yu et al, 2021) and to an invariable genotype-phenotype correlation. This congruent phenotype was also observed in two men with LoF variants in *SHOC1*. Only in one man with a homozygous splice site variant in *SHOC1* did we detect CREM-positive round spermatids. This splice site variant induces the in-frame deletion of a single exon. The resulting protein may still have reduced function and maintained interaction to the other ZZS proteins, due to a preserved XPF-ERCC1-like complex formation and a partly maintained distant helicase hits region, as described similarly for yeast and mouse mutants (De Muyt et al, 2018; Guiraldelli et al, 2018).

In contrast to *TEX11* and *SHOC1*, biallelic variants in *SPO16* have so far only been linked to female infertility (Qi et al, 2023); thus, our study demonstrates the first case of an infertile man with a biallelic LoF variant in *SPO16*, highlighting it as a novel candidate gene also for male infertility. His testicular phenotype and type of spermatogenic arrest is comparable to that of *TEX11* and *SHOC1* variant carriers, where no haploid germ cells were detected. DSBs remain persistent and complete synapsis of all chromosomes was lacking. In line, recombination intermediate stabilisation was disrupted and class I crossover resolution was absent, resulting in a complete meiotic arrest. In conclusion, in-depth testicular characterisation of the large number of men with LoF variants in *TEX11* points to a genotype-specific phenotype that correlates with the phenotype of the other two key ZZS genes, *SHOC1* and *SPO16*. However, further identification of cases is needed to fully understand how the loss of SHOC1 and SPO16 function affects meiotic progression.

Our observations on human cases are substantiated by described mouse models targeting the orthologues of the respective genes, which displayed sterility and an early spermatocyte arrest (Yang et al, 2008; Zhang et al, 2018, 2019). *Tex11* knockout mice were associated with male sterility due to apoptosis of spermatocytes, asynapsis of homologues, delayed DSB repair, and a decreased number of crossover events (Yang et al, 2008). Both complete and germ cell-specific knockout of *Shoc1* resulted in sterility. Germ cell arrest varied between zygotene and mid-pachytene stages, and cells lacked distinct XY bodies, complete chromosomal synapsis, recombination intermediates and class I crossover events (Zhang et al, 2018). Homozygous knockout of *Spo16* in mice led to sterility, impaired chromosome pairing, and reduced class I crossover formation but regular meiotic DSB formation and persistent single-strand invasion (Zhang et al, 2019). Compared to *Shoc1*, the *Spo16* knockout mice displayed milder defects in DSB repair and synapsis, indicating that SHOC1 alone maintains partial functionality and enables reduced DSB repair.

The common basis of all four genes *M1AP, SHOC1, TEX11*, and *SPO16*, is male infertility and meiotic arrest in the case of a loss of protein function. However, this study identified striking differences in the testicular phenotype of men with LoF variants in *M1AP* in contrast to those patients affected by LoF variants in the main ZZS complex genes *SHOC1, TEX11*, and *SPO16*. To explain why M1AP deficiency still allows a fraction of germ cells in men and mice to progress through meiosis to the haploid germ cell stage, we argue that even in the absence of M1AP, a reduced SHOC1-TEX11-SPO16 interplay is maintained: SHOC1 forms a heterodimer with SPO16 that recognises DNA-joined molecules and binds to and stabilises early recombination intermediates (Guiraldelli et al, 2018; Zhang et al, 2019). TEX11 is recruited to these sites and, in turn, assists in recruiting meiosis-specific proteins such as SYCP2 and MutSy (MSH4-MSH5), which are needed to facilitate synaptonemal complex assembly and DSB repair, finally yielding to crossover formation (Yang et al, 2008; Zhang et al, 2018). We propose that M1AP is functionally linked to the regulation of SHOC1, TEX11, and SPO16, thereby affecting recombination stability, processing, and class I crossover resolution.

Thus, our findings indicate that M1AP is an important functional enhancer for promoting meiotic resolution in the ZZS network, but it is not mandatory. This is supported by the fact that although meiosis is an evolutionarily highly conserved process (reviewed in Börner et al, 2023), and, accordingly, the ZZS proteins Zip2 (SHOC1), Zip4 (TEX11), and Spo16 (SPO16) play an essential role in meiotic recombination in yeast, this lower eukaryote has no M1AP orthologue. While yeast is a well-established model organism for meiosis, M1AP is only conserved down to fish species, indicating that this protein's function in meiotic progression is evolutionarily younger than those of the ZZS complex.

In contrast to male meiosis in mice and humans, where deficiency in M1AP or ZZS complex components results in

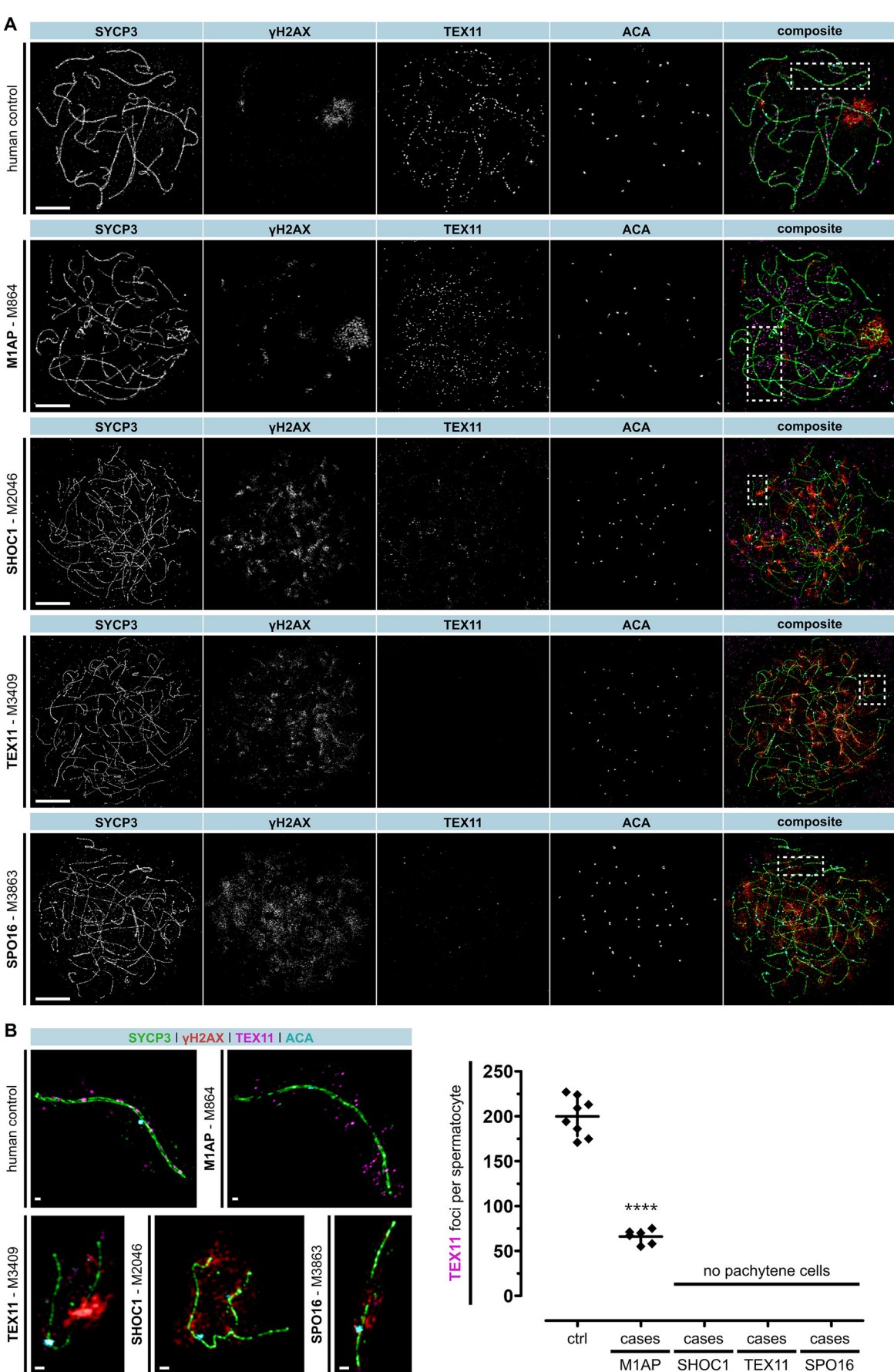

**Figure 6. Reduced TEX11 foci in a man with loss-of-function variant in M1AP (M864).**

(A) Human spermatocyte spreads were stained for synaptonemal complex formation (= SYCP3, green), DNA double-strand breaks (γH2AX = red), centromeric regions (= ACA, cyan), and ZZS recruitment (TEX11 = magenta). Pachytene-like cells with progressed DNA double-strand breaks and γH2AX-positive XY bodies of a control and M864, deficient for M1AP, were compared. LoF in SHOC1 (M2046), TEX11 (M3409), and SPO16 (M3863) was associated with an early meiosis I defect including incomplete DSB repair and disturbed assembly of early recombination intermediates leading to a lack of TEX11 foci on the chromosomal axes. (B) Along the chromosomal axis of wild-type spermatocytes, 199.88 ± 20.16 TEX11 foci were counted per cell, while M1AP deficiency significantly reduced the recruitment of TEX11 to the axes (66.00 ± 7.16 foci per cell). In contrast, men deficient for SHOC1 (M2046), TEX11 (M3409), or SPO16 (M8363) completely lacked pachytene cells, and no quantification was performed in these cases. $P$ value was determined between the control and M1AP group by an unpaired two-tailed $t$ test (****$P$ = 6.2408*10$^{-9}$). Data are presented as the mean and standard deviation, quantifying individual pachytene spermatocytes of one case each (CTRL = eight spermatocytes, M1AP = six spermatocytes). The scale bars represent 10 μm or 1 μm for magnification. Source data are available online for this figure.

subfertility or infertility, the role of these proteins in female fertility is less clear. In mice, female *M1ap* and *Tex11* knockouts are fertile, albeit with a reduced litter size in the case of *Tex11* knockout (Arango et al, 2013; Li et al, 2023; Yang et al, 2008). Murine *Shoc1* and *Spo16* knockouts display sterility in both sexes (Zhang et al, 2018, 2019), and in humans, biallelic LoF variants in *SHOC1* and *SPO16* have been described to lead to female infertility. For *TEX11*, only women with heterozygous variants have been found to retain their fertility (Wyrwoll et al, 2023b). For *M1AP*, however, because we and others have not identified biallelic variants in *M1AP* in fertile or infertile women, we cannot yet define the role of M1AP in human female meiosis.

*M1AP* is a clinically relevant gene for male fertility. The TESE success rate in our cohort was 40%, and we described one man who fathered a healthy, euploid child, thus showing that biallelic LoF variants in *M1AP* are compatible with fatherhood and, thereby, presenting a proof-of-principle. However, further studies are needed to fully elucidate the underlying molecular mechanisms and enable a risk estimation of checkpoint failure and aneuploidy in these men. Accordingly, it currently remains elusive whether all altered cells in men with LoF variants in *M1AP* indeed undergo checkpoint-induced apoptosis or whether rare exceptions develop independently of the spindle-assembly checkpoint, resulting in aneuploid spermatozoa.

The majority of aneuploid embryos are not viable, frequently leading to spontaneous abortion (Hassold and Hunt, 2001). While studies have suggested an equal fertilisation capability between aneuploid and euploid spermatozoa, they have also evidenced a correlation between high sperm aneuploidy and recurrent ICSI or MAR failure as well as lower rates of pregnancy and live birth (reviewed in Ioannou et al, 2019). ICSI attempts with spermatozoa from two men with LoF variants in *M1AP* (M2062 and M2746) succeeded in fertilising the oocytes. Most of these did, however, not develop to the blastocyst stage or did not lead to a clinical pregnancy. Aneuploid spermatozoa could be one potential explanation for this, especially concerning chromosomes 21, 22, X, and Y, as these typically have a single recombination event and are, thus, more prone to progress without the obligatory crossover (Ferguson et al, 2007; Sun et al, 2008). This hypothesis could not be substantiated by our data, because we could not perform a direct analysis of the very few spermatozoa from any of the men to assess the frequency of aneuploidies. Thus, subsequent studies are needed to answer this question. So far, couples with M1AP-related male-factor infertility should be counselled regarding the small but existed chance to conceive a chromosomally-balanced biological child. At the same time it is imperative to discuss the potential risks

of aneuploidies. These couples could benefit from preimplantation genetic testing for aneuploidy (PGT-A) to enhance their reproductive success. Whereas the benefits of PGT-A in unexplained recurrent pregnancy failure cases remain uncertain, some studies focusing on male-factor infertility have shown improved clinical MAR outcomes after PGT-A (Rodrigo et al, 2019; Xu et al, 2021)—and M1AP-associated infertility may be such a case.

The consideration of certain limitations of this study is imperative: first of all, we detected the protein-protein interaction of human M1AP with the human ZZS complex, which reflected very well the findings published for mice. However, the determination of the TEX11 binding sides of M1AP was not performed. Furthermore, the study is based on human material. This means that only limited testicular material from men with LoF variants was available for specific experiments such as immunofluorescence staining of homologous recombination markers on meiotic spreads. Accordingly, it was not feasible to provide data on the staging of the different prophase I substages analysed. The decision that had to be made was whether to describe a larger number of markers descriptively or only a small number with quantification. As the results obtained from the human data were corroborated by the findings published for the corresponding mouse models, it can be deduced that the broader perspective on human meiotic impairment due to M1AP or ZZS deficiency was appropriate.

In conclusion, both *M1AP* and *TEX11* represent two of the most frequently identified monogenic causes for NOA in men (Yang et al, 2015; Nagirnaja et al, 2022; Wyrwoll et al, 2023b). By identifying the first man with an underlying LoF variant in *SPO16*, we expand the clinical relevance of genetic causes to all three main ZZS genes and underline their overall importance. Here, we present the first detailed description of the testicular phenotypes of affected men, which is important for inferring the function of the proteins involved. At least for *M1AP* and *TEX11*, we provide for the first time a clear genotype-phenotype correlation, as men with LoF in the same gene show a concordant phenotype. As a result, it will now be possible to distinguish likely pathogenic variants in *TEX11* and *M1AP* from likely benign variants, i.e., those less likely to have a functional effect. Specifically, pathogenic missense variants in these genes leading to abolished protein function are expected to lead to the described, highly specific testicular phenotype. To the best of our knowledge, our study is also the first report on the birth of a healthy child (reviewed in Xie et al, 2022) having a father with a pathogenic gene variant in a clinically established disease gene for NOA. This not only distinguishes *M1AP* from other NOA genes, but it also shows that human M1AP, in contrast to the other ZZS proteins, is not required for class I crossover formation.

                     

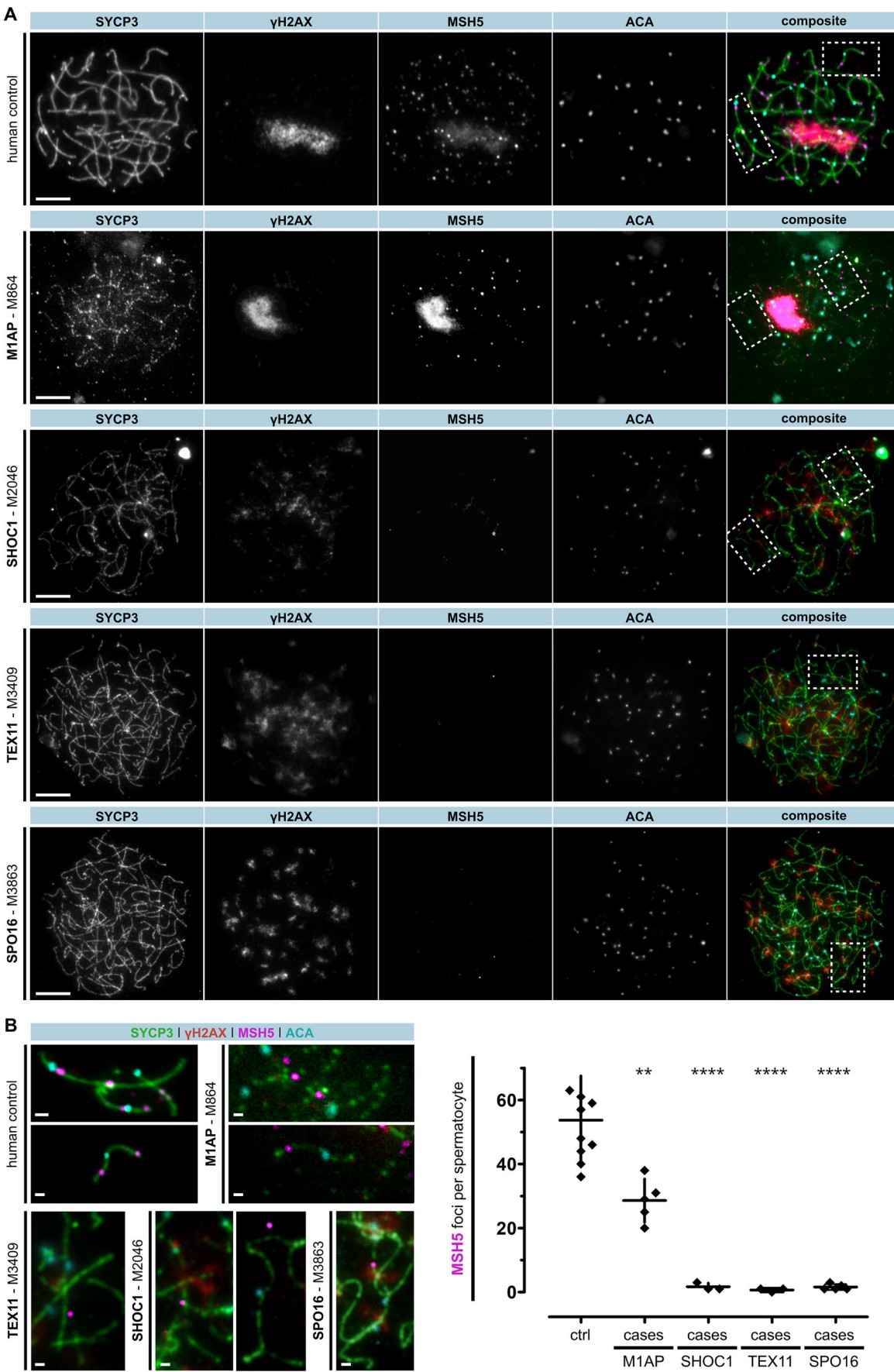

◄ **Figure 7. Human spermatocyte spreads showed reduced or absent MSH5 foci in men with loss-of-function variants in *M1AP* or the ZZS genes.**

(A) Human spermatocyte spreads were stained for synaptonemal complex formation (= SYCP3, green), DNA double-strand breaks (γH2AX = red), centromeric regions (= ACA, cyan), and ZMM recruitment (MSH5 = magenta). A control and men deficient for M1AP (M864), SHOC1 (M2046), TEX11 (M3409), or SPO16 (M8363) were compared. (B) Distinct MSH5 foci were seen on chromosome axes of pachytene cells in the control and M864. The LoF variant in *M1AP* led to a significant reduction of MSH5 foci, while LoF variants in *SHOC1, TEX11*, or *SPO16* led to a nearly complete absence of MSH5 foci. Bleed-through was observed between the red and magenta channels. Therefore, only MSH5 foci that were distinct from the XY body (γH2AX) were counted because the γH2AX signal was superimposed on the MSH5 foci. *P* value was determined between the control and each LoF group by an unpaired two-tailed *t* test (**$P = 0.0023$, ****$P = 5.7658 \times 10^{-5}$ (SHOC1), $4.8489 \times 10^{-5}$ (TEX11), $1.5895 \times 10^{-6}$ (SPO16)). Data are presented as the mean and standard deviation, quantifying individual pachytene (CTRL, M1AP) or most progressed zygotene-like spermatocytes (ZZS) of one case each (CTRL = nine spermatocytes, M1AP = five spermatocytes, SHOC1/TEX11 = three spermatocytes, SPO16 = five spermatocytes). The scale bars represent 10 μm and 1 μm for magnification. Source data are available online for this figure.

# Methods

### Reagents and tools table

| Reagent/resource | Reference or source | Identifier or catalogue number |
|---|---|---|
| **Experimental models** | | |
| HEK293T | DSMZ, Braunschweig, Germany | ACC 635 |
| **Recombinant DNA** | | |
| C1orf146_OHu16306C_pcDNA3.1(+)-C-HA | Genscript, Piscataway Township, USA | OHu16306C |
| M1AP_OHu01575C_pcDNA3.1(+)-N-DYK | Genscript, Piscataway Township, USA | OHu01575C |
| SHOC1_OHu76993C_pcDNA3.1(+)-C-HA | Genscript, Piscataway Township, USA | OHu76993C |
| TEX11_OHu29882C_pcDNA3.1(+)-C-HA | Genscript, Piscataway Township, USA | OHu29882C |
| pDESTsplice vector | Stefan Stamm, Addgene | plasmid #32484 |
| pENTR/D-TOPO® vector | Thermo Scientific, Waltham, USA | K240020 |
| **Antibodies** | | |
| Primary antibodies | | Appendix Table S3 |
| Secondary antibodies | | Appendix Table S3 |
| **Oligonucleotides and other sequence-based reagents** | | |
| ddPCR supermix | Bio-Rad, Hercules, USA | 1863024 |
| primer | This study | Appendix Table S2 |
| **Chemicals, enzymes, and other reagents** | | |
| 3,3'-diaminobenzidine tetrahydrochloride (DAB) | Sigma Aldrich, St. Louis, USA | D5905 |
| anti-HA tag beads | Thermo Scientific, Waltham, USA | 88838 |
| bovine serum albumin (BSA) | Merck, Darmstadt, Germany | A9647 |
| CaCl₂ | Roth, Karlsruhe, Germany | CN93.1 |
| chloroquine diphosphate salt | Sigma Aldrich, St. Louis, USA | C6628 |
| DTT | Merck, Darmstadt, Germany | 10197777001 |
| Dulbecco's phosphate-buffered saline (D-PBS) | Sigma Aldrich, St. Louis, USA | 14190 |
| EDTA | AppliChem, Darmstadt, Germany | A4892 |
| EDTA-free protease inhibitor cocktail | Roche, Basel, Schweiz | 11836170001 |
| foetal calf serum (FCS) | Gibco, Thermo Scientific, Waltham, USA | 10082-147 |
| GatewayTM LR clonaseTM enzyme mix | Thermo Scientific, Waltham, USA | 11791020 |
| glycerol | Sigma Aldrich, St. Louis, USA | G5516 |
| glycine | Sigma Aldrich, St. Louis, USA | G7126 |
| H₂O₂, 3% | Pharmacy of the University Hospital Münster | 1002187 |

| Reagent/resource | Reference or source | Identifier or catalogue number |
|---|---|---|
| K2® transfection system | Biontex, München, Germany | T060 |
| Laemmli sample buffer, 4x | Bio-Rad, Hercules, USA | 1610747 |
| Mayer's haematoxylin | Sigma Aldrich, St. Louis, USA | 1092491000 |
| MgCl2 | Merck, Darmstadt, Germany | 442611 |
| M-GLAS® liquid cover glass medium | Merck, Darmstadt, Germany | 103.973 |
| milk powder | AppliChem, Darmstadt, Germany | A0830 |
| NaCl | Sigma Aldrich, St. Louis, USA | S9888 |
| Neo-ClearTM | Sigma Aldrich, St. Louis, USA | 109843 |
| normal donkey serum | Merck, Darmstadt, Germany | S30 |
| normal goat serum | abcam, Cambridge, United Kingdom | Ab7481 |
| NP-40 | Thermo Scientific, Waltham, USA | 85124 |
| Opti-MEMTM reduced serum medium | Gibco, Thermo Scientific, Waltham, USA | 31985062 |
| PageRulerTM plus prestained protein ladder | Thermo Scientific, Waltham, USA | 26619 |
| Penicillin-Streptomycin | Thermo Scientific, Waltham, USA | 15070-063 |
| PFA | Merck, Darmstadt, Germany | 158127 |
| Photo-Flo | Kodak, Rochester, USA | K1464510 |
| PhusionTM high-fidelity DNA polymerase | Thermo Scientific, Waltham, USA | F530 |
| PierceTM 3x DYKDDDDK peptide | Thermo Scientific, Waltham, USA | A36806 |
| PierceTM elution buffer, pH2 | Thermo Scientific, Waltham, USA | 21028 |
| PierceTM magnetic anti-DYKDDDDK tag beads | Thermo Scientific, Waltham, USA | A36797 |
| ROTI® mount FluorCare medium | Roth, Karlsruhe, Germany | HP19.1 |
| sodium acid (NaN₃) | Sigma Aldrich, St. Louis, USA | S2002 |
| sodium borate | Roth, Karlsruhe, Germany | 8643 |
| sodium citrate buffer, pH6 | Thermo Scientific, Waltham, USA | 5000 |
| sodium DL-lactate solution | Thermo Scientific, Waltham, USA | 41529 |
| Sperm-FreezeTM | Ferti-Pro, Leveringen, Belgium | 3080 |
| sucrose | Thermo Scientific, Waltham, USA | 10134050 |
| Tris-EDTA buffer, pH9 | Zytomed Systems, Bargteheide, Germany | ZYT-ZUC029 |
| Tris-HCl | Sigma Aldrich, St. Louis, USA | T3253 |
| Triton X-100 | Sigma Aldrich, St. Louis, USA | T8787 |
| TWEEN® 20 Detergent | Merck Millipore, Darmstadt, Germany | 655205 |
| **Software** | | |
| AlphaFold2 from EMBL-EBI | https://alphafold.ebi.ac.uk/ | |
| BWA Mem v0.7.17 | https://github.com/lh3/bwa | |
| Cutadapt v1.15 | https://cutadapt.readthedocs.io/en/v1.15/ | |

| Reagent/resource | Reference or source | Identifier or catalogue number |
|---|---|---|
| Ensembl variant effect predictor | https://www.ensembl.org/info/docs/tools/vep/index.html | |
| Fiji by ImageJ (v2.3.0/1.54 h) | https://fiji.sc/ | |
| GATK toolkit v3.8 | https://gatk.broadinstitute.org/hc/en-us | |
| GraphPad Prism software (v10.1.2) | https://www.graphpad.com/ | |
| HGNC | https://www.genenames.org/ | |
| Illumina Dragen Bio-IT platform v4.2 | https://support-docs.illumina.com/SW/dragen_v42/Content/SW/FrontPages/DRAGEN.htm | |
| Leica Application Suite X (LASX) 3.10.0.28982 | https://www.leica-microsystems.com/products/microscope-software/p/leica-las-x-ls/ | |
| Mutation Taster | https://www.mutationtaster.org/ | |
| **Other** | | |
| 4–15% mini-PROTEAN® TGX Stain-FreeTM precast gels | Bio-Rad, Hercules, USA | 4568085 |
| Agilent's SureSelect human all exon kits V4, V5, and V6 | Agilent, Santa Clara, USA | |
| Agilent's SureSelectQXT target enrichment for Illumina multiplexed sequencing featuring transposase-based library prep technology | Agilent, Santa Clara, USA | |
| Bio-Rad image lab software and molecular weight analysis tool | Bio-Rad, Hercules, USA | 12012931 |
| ChemiDoc MP imaging system | Bio-Rad, Hercules, USA | 12003154 |
| ClarityTM Western ECL substrate kit | Bio-Rad, Hercules, USA | 1705060S |
| GatewayTM LR clonaseTM enzyme mix | Thermo Scientific, Waltham, USA | 11791020 |
| HiSeq 3000/4000 SBS Kit | Illumina, Cambridge, United Kingdom | |
| HiSeq X Ten Reagent Kit | Illumina, Cambridge, United Kingdom | |
| Illumina HiSeq4000 | Illumina, Cambridge, United Kingdom | |
| Illumina HiSeqX | Illumina, Cambridge, United Kingdom | |
| Illumina NextSeq500 | Illumina, Cambridge, United Kingdom | |
| Illumina NextSeq550 | Illumina, Cambridge, United Kingdom | |
| Illumina NovaSeq6000 | Illumina, Cambridge, United Kingdom | |
| K2® Transfection System | Biontex, Munich, Germany | T060-2.0 |
| Leica DM6 B microscope + Flexacam C3 camera | Leica, Wetzlar, Germany | |
| Leica DM750 microscope + Leica ICC50 HD camera | Leica, Wetzlar, Germany | |
| Leica Filtersystem A UV BP 360/40, Dichroid: 400, LP 425 | Leica, Wetzlar, Germany | |
| Leica Filtersystem D BP 355/425, Dichroid: 455, LP 470 | Leica, Wetzlar, Germany | |
| Leica Filtersystem I3 BP 450/490, Dichroid: 510, LP 515 | Leica, Wetzlar, Germany | |
| Leica Filtersystem N2.1 BP 515-560, Dichroid: 580, LP 590 | Leica, Wetzlar, Germany | |
| Leica HC PL APO 100x/1.40-0.70 Oil | Leica, Wetzlar, Germany | |
| NextSeq 500 V2 high-output Kit | Illumina, Cambridge, United Kingdom | |

| Reagent/resource | Reference or source | Identifier or catalogue number |
|---|---|---|
| NovaSeq 6000 S1 and S2 reagent kits v1.5 | Illumina, Cambridge, United Kingdom | |
| Olympus BX61VS microscope | Olympus, Shinjuku, Japan | |
| PhusionTM high-fidelity DNA polymerase | Thermo Scientific, Waltham, USA | F530 |
| PreciPoint O8 scanning microscope system | Precipoint, Garching, Germany | |
| ProtoScript® II first strand cDNA synthesis kit | New England Biolabs, Frankfurt, Germany | E6560 |
| QuikChange II XL Site-Directed Mutagenesis Kit | Agilent Technologies, Santa Clara, USA | 200516 |
| QX200 droplet digital PCR system | Bio-Rad, Hercules, USA | 1864001 |
| QX200 droplet reader | Bio-Rad, Hercules, USA | 1864003 |
| QX200TM droplet generator | Bio-Rad, Hercules, USA | 1864002 |
| RNeasy plus mini kit | Qiagen, Hilden, Germany | 74134 |
| TapeStation D1000 system | Agilent, Santa Clara, USA | 5067-5582 |
| Tecan Infinite 200Pro | Tecan, Männedorf, Switzerland | |
| Thermo Fisher Qubit | Thermo Scientific, Waltham, USA | |
| Thunder Imager 2D Tissue | Leica, Wetzlar, Germany | |
| Trans-Blot® TurboTM mini PVDF transfer packs | Bio-Rad, Hercules, USA | 1704156EDU |
| TUNEL Kit | Thermo Scientific, Waltham, USA | C10625 |
| Twist Bioscience's exome 2.0 plus comprehensive spike-in | Twist Bioscience, San Francisco, USA | |
| Twist Bioscience's human core exome plus RefSeq spike-in | Twist Bioscience, San Francisco, USA | |
| Zeiss Elyra 7 microscope for specialised 3D structured illumination (SIM2) | Zeiss, Jena, Germany | |

## Ethical approval

All human subjects gave written informed consent compliant with local requirements. The MERGE study was approved by the Ethics Committee of the Ärztekammer Westfalen-Lippe and the Medical Faculty Münster (Münster #2010-578-f-S, #2012-555-f-S; Giessen #26/11); the study of GEMINI-377 was approved by the ethics committee for the Capital Region in Denmark (#H-2-2014-103), all in accordance with the WMA Declaration of Helsinki and the Department of Health and Human Services Belmont Report.

## Study cohort

The Male Reproductive Genomics (MERGE) study comprised exome ($N = 2,629$) and genome sequencing ($N = 74$) datasets of overall 2703 probands. It includes men with various infertility phenotypes: 1622 men with azoospermia (no spermatozoa in the ejaculate, HP:0000027), 487 men with cryptozoospermia (spermatozoa only identified after centrifugation, HP:0030974), 380 men with varying oligozoospermia (total sperm count: >0 to <39 million spermatozoa, HP:0000798), 188 men with a total sperm count above 39 million, and 26 family members. Established causes for male infertility including previous radio- or chemotherapy, hypogonadotropic hypogonadism, Klinefelter syndrome, or microdeletions of the azoospermia factor (AZF) regions on the

Y-chromosome were exclusion criteria. All men underwent routine physical and hormonal analysis of luteinising hormone (LH), follicle-stimulating hormone (FSH), testosterone (T) as well as semen analysis according to the respective WHO guidelines.

For this study, we selected men with rare biallelic (minor allele frequency [MAF] in gnomAD database v2.1.1, ≤0.01) or hemizygous (MAF ≤ 0.001) loss-of-function (LoF) variants (stop-, frameshift-, splice site variants) and deletions in *M1AP*, *SHOC1* [*C9orf84*], *TEX11*, and *C1orf146* [*SPO16*]. For consistency in the manuscript, we used the HGNC (https://www.genenames.org/) approved gene symbols for *M1AP*, *SHOC1*, and *TEX11*, whereas we refer to *C1orf146* using its alias gene symbol, *SPO16*. As a reference, the longest transcript with the highest testicular expression was chosen (*M1AP*: NM_001321739.2, *SHOC1*: NM_173521.5, *TEX11*: NM_1003811.2, *SPO16*: NM_001012425.2). If possible, segregation analysis was conducted on the DNA of family members. Samples with qualitatively and quantitatively normal spermatogenesis were included as controls for IHC and the statistical analyses (M2132, M2211, M3254, Appendix Fig. S12). In addition, one previously published case from an external cohort (GEMINI-377, referred to as G-377) was included for subsequent analysis (Nagirnaja et al, 2022).

## Exome and genome sequencing

Genomic DNA was extracted from peripheral blood leucocytes by standard methods. For exome sequencing, samples were prepared and enrichment was performed according to the protocols of either Agilent's SureSelectQXT target enrichment for Illumina multiplexed sequencing featuring transposase-based library prep technology or Twist Bioscience's Twist Human Core Exome. For library capturing, Agilent's SureSelect human all exon kits V4, V5, and V6 or Twist Bioscience's human core exome plus RefSeq spike-in and exome 2.0 plus comprehensive spike-in were used. Sample multiplexing was achieved by tagging the libraries with appropriate index primer pairs. Quality and quantity was determined using the ThermoFisher Qubit, the Agilent TapeStation 2200, and the Tecan Infinite 200Pro microplate reader. Finally, sequencing itself was performed on the Illumina HiSeq 4000, the Illumina HiSeqX, the Illumina NextSeq 500, the Illumina NextSeq 550, or the NovaSeq 6000 system, using the HiSeq 3000/4000 SBS (300 cycles), the HiSeq X Ten Reagent (300 cycles), the NextSeq 500 V2 high-output (300 cycles), or the NovaSeq 6000 S1 and S2 reagent kits v1.5 (200 cycles), respectively. Exome sequencing and analysis of patient GEMINI-377 has been described previously (Nagirnaja et al, 2022).

Genome sequencing libraries were prepared with Illumina's DNA PCR-Free library kit. Index tagging for multiplexed sequencing was accomplished by employing suitable pairs of index primers. DNA and library quantity and quality were assessed using the ThermoFisher Qubit and the Tecan Infinite 200 Pro Microplate Reader, respectively. Sequencing was performed on the NovaSeq 6000 System, utilising NovaSeq 6000 S1, S2, and S4 Reagent kits v1.5 (300 cycles), respectively.

## Variant calling

Adaptor sequences and primers were trimmed using Cutadapt v1.15 (Martin, 2011) and reads were aligned against Genome Reference Consortium human build 37 (GRCh37.p13) with BWA Mem v0.7.17 (Li and Durbin, 2010). Recalibration of base quality and variant calling was performed with the GATK toolkit v3.8 (McKenna et al, 2010) or with the with Illumina Dragen Bio-IT platform v4.2, both with haplotype caller according to the best practice recommendations. Duplicate reads or reads mapping to multiple locations in the exome were excluded. Identified variants were annotated using the Ensembl variant effect predictor (McLaren et al, 2016).

## Cohort screening for high impact variants

The MERGE sequencing data was screened for loss-of-function (LoF) (start-loss, stop-gain and frameshift) and splice site variants (± 20 nucleotides) in *M1AP*, *SHOC1* [*C9orf84*], *TEX11*, and *C1orf146* [*SPO16*]. For consistency in the manuscript, we used the HGNC (https://www.genenames.org/) approved gene symbols for *M1AP*, *SHOC1*, and *TEX11*, whereas we refer to *C1orf146* using its alias symbol, *SPO16*. We considered only variants corresponding to the respective mode of inheritance for each gene (*M1AP*/*SHOC1*/*SPO16* = autosomal-recessive (AR), *TEX11* = X-linked recessive (XR)). Resulting variants were filtered for the general population frequency (gnomAD database v2.1.1; (Karczewski et al, 2020; minor allele frequency [MAF] ≤0.01 (AR) or ≤0.001 (XR)) and an occurrence ≤30 times in our in-house database containing 4,377 datasets from individuals with other genetic diseases. As reference, the longest transcript per gene with the highest testicular expression (according to GTEx; (Yu et al, 2011) was selected (*M1AP*: NM_001321739.2, *SHOC1*: NM_173521.5, *TEX11*: NM_1003811.2, and *SPO16*: NM_001012425.2). Nonsense-mediated decay prediction was performed by (https://www.mutationtaster.org/).

The sequencing data were analysed for second hits present in a list of 18 azoospermia genes with at least moderate clinical validity (Wyrwoll et al, 2023b) and 363 candidate genes with strong expression in human male germ cells and an associated Gene Ontology classification of *male infertility* in the Mouse Genome Informatics Database.

## Variant validation and segregation analyses

Identified SNVs were confirmed by Sanger sequencing. Similarly, DNA from family members was analysed for co-segregation attempts and to show if variants occur biallelic. Briefly, the region of interest was amplified using the respective primers listed in Appendix Table S2. Purification of PCR products and sequencing was performed according to standard protocols.

For validation of deletions, droplet digital PCR (ddPCR) was used as described previously (Dicke et al, 2023). In brief, ddPCR was carried out using the QX200 droplet digital PCR system (Bio-Rad, #1864001), the ddPCR supermix for probes (no dUTP) (Bio-Rad, #1863024), respective primers/probes (final concentrations: 200 nM, Appendix Table S3), and 100 ng template DNA in a final volume of 20 μl. For restriction digestion, HaeIII was used. Droplets were generated using the QX200™ droplet generator (Bio-Rad, #1864002) followed by a two-step cycling protocol, set at 95 °C for 10 min, with 40 cycles of 94 °C for 30 s and 58 °C for 1 min, with a final extension of 98 °C for 10 min and a 4 °C hold. The final analysis was performed with 6-FAM- and HEX-channels using the QX200 droplet reader (Bio-Rad, #1864003). Droplet digital PCR validated the identified deletions in M2820 and M3152 (Appendix Figs. S1 and S2).

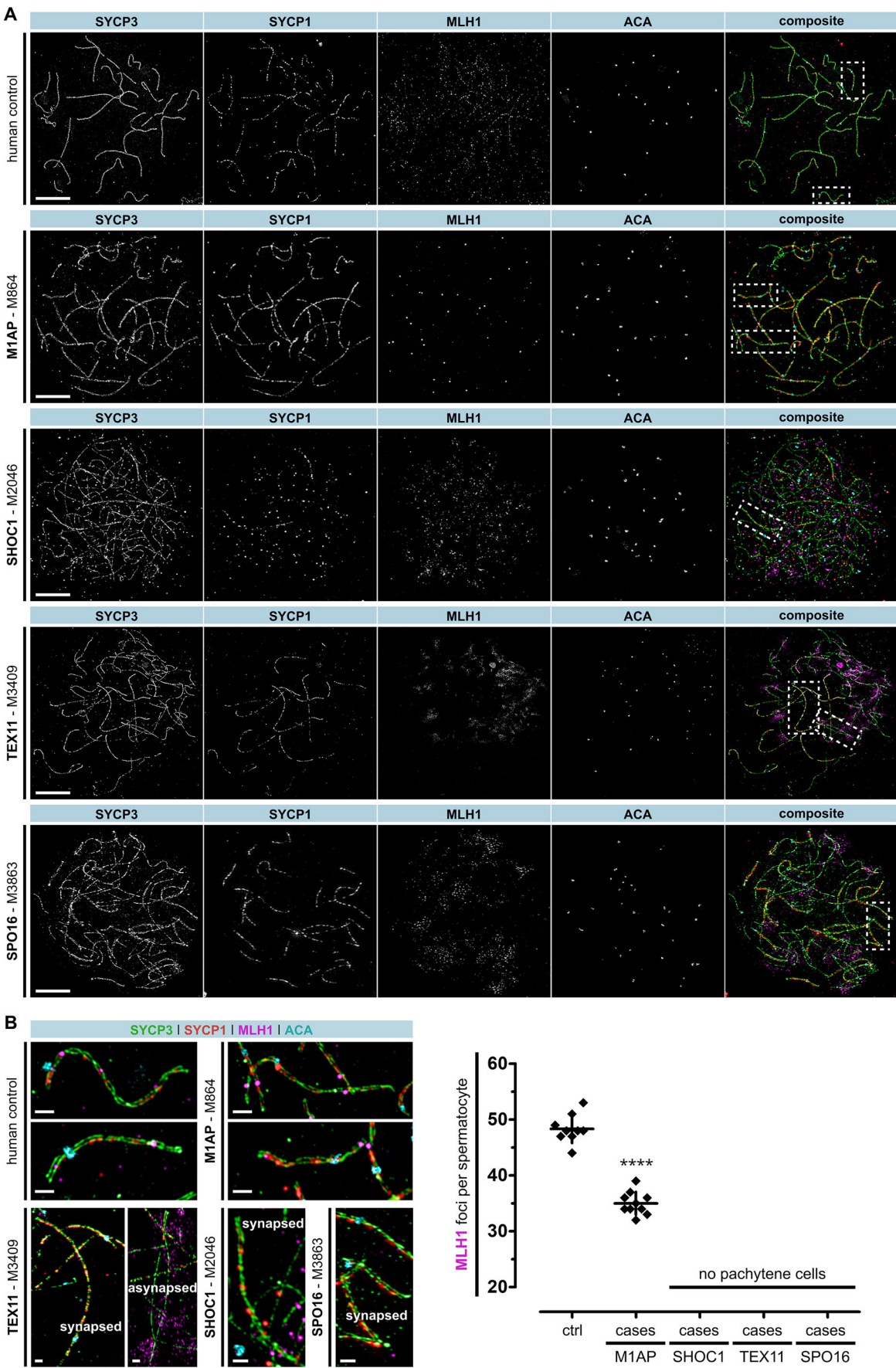

**Figure 8. A man with loss-of-function variant in *M1AP* showed reduced MLH1 foci.**

(A) Human spermatocyte spreads were stained for synaptonemal complex formation (= SYCP3, green + SYCP1, red), centromeric regions (= ACA, cyan), and designated crossover sites (= MLH1, magenta). A control and M864, deficient for M1AP (also seen in Fig. 9A), M2046 (SHOC1), M3409 (TEX11), and M3863 (SPO16) were compared. (B) MLH1 localisation was seen on synapsed chromosome axes of pachytene spermatocytes in the control and M864. The total number of MLH1 foci per spermatocyte was reduced in M864. In contrast, men deficient for SHOC1 (M2046), TEX11 (M3409), or SPO16 (M8363) completely lacked pachytene-like cells. Instead, spermatocytes showed asynapsed (SHOC1) or incomplete synapsed (TEX11 and SPO16) chromosomes where SYCP1 localisation was lacking or reduced. No MLH1 was seen because cells arrested in a zygotene-like stage, and no quantification was performed in these cases. $P$ value was determined between the control and M1AP group by an unpaired two-tailed $t$ test (****$P = 4.6836*10^{-10}$). Data are presented as the mean and standard deviation, quantifying individual pachytene spermatocytes of one case each (CTRL nine spermatocytes, M1AP ten spermatocytes). The scale bars represent 10 µm and 1 µm for magnification. Source data are available online for this figure.

## Ploidy analysis from genome sequencing data of M2746's offspring

Ploidy for autosomes and gonosomes was estimated using Illumina DragenBio-IT Platform v4.2. Read counts were normalised by dividing the median read count of each chromosome by the median read count of all autosomes. Data is depicted as log2.

## Cell culture and transient (co-)transfection experiments

Human embryonic kidney (HEK) 293 T cells were cultured in Dulbecco's modified Eagle medium (DMEM) with 10% foetal calf serum (FCS) and 1% penicillin/streptomycin (PS). The culture was maintained in T75 cell culture flasks at 37 °C, 5% $CO_2$, and 85% humidity. Test for mycoplasma contamination was negative. Cell authenticity was certified by the Leibniz Institute DSMZ-German Collection of Microorganisms and Cell Culture GmbH.

For overexpression experiments, 400,000 cells per well were seeded in 6-well plates, cultivated for two days in DMEM-FCS without PS, and transfected at 80-100% confluence using the K2® transfection system (Biontex, #T060) according to the manufacturer's instructions. Briefly, K2® maximiser reagent was added two hours before transfection. Directly before transfection, the medium was supplemented with 100 µm chloroquine diphosphate salt (Sigma Aldrich, #C6628). Subsequently, 4 µg cDNA, 260 µl Opti-MEM™ reduced serum medium (Gibco, #31985062), and 9 µl K2® transfection reagent per well were mixed, incubated for 15 min, and added to the cells. For co-transfection of DYK-tagged *M1AP* (NM_001321739.2) and either HA-tagged *SHOC1* (NM_173521.5), *TEX11* (NM_031276.3), or *SPO16* (NM_001012425.2), the total amount of DNA was maintained, and the cDNA ratio was adapted according to the respective size of each gene. Transfection and subsequent analysis were performed in three independent biological replicates minimum and three wells were pooled per replicate.

Cell lysis was performed 24 h post-transfection. Accordingly, cells were washed with 1x PBS and collected in a microcentrifuge tube on ice. After centrifugation (4 °C, 5 min, 200 rpm), cells were re-suspended with co-IP lysis buffer (25 mM Tris-HCl pH 7.4, 150 mM NaCl, 10 mM EDTA, 1% NP-40, 5% glycerol) and supplemented with 1x EDTA-free protease inhibitor cocktail (Roche, #11836170001). Samples were incubated for 15 min on ice and collected by centrifugation (4 °C, 15 min, 13000 rpm). The protein-containing supernatant was transferred to a fresh microcentrifuge tube and directly processed or stored at −20 °C until further usage. When co-immunoprecipitation was conducted, a small amount of lysate was kept for the 'input' protein validation.

## Co-immunoprecipitation

Co-immunoprecipitation protocols were optimised and, depending on the respective protein-protein interaction, two different types of magnetic beads were used for this study: Pierce™ magnetic anti-DYKDDDDK tag beads (Thermo Scientific, #A36797) for M1AP-SPO16 or anti-HA tag beads (Thermo Scientific, #88838) for M1AP-SHOC1 as well as M1AP-TEX11. Beads were equilibrated and pre-cleared according to the instructions and incubated with respective protein lysates for 30 min at RT under gentle rotation. Anti-DYKDDDDK beads were washed with 1x PBS three times; anti-HA beads were washed with 0.025% TBS-Tween six times, and both were washed with double-distilled water once. Gentle elution was achieved by incubation with a Pierce™ 3× DYKDDDDK peptide (Thermo Scientific, #A36806) at 1.5 mg/ml in 1× PBS for 5 min shaking or by using the Pierce™ elution buffer (pH 2, Thermo Scientific, #21028) for 8 min. Beads were collected with a magnetic stand and supernatants contained the eluted targets. Immunoprecipitation (IP) with pure prey lysates served as negative controls. Samples were either stored at −20 °C or processed directly.

## Western blotting

Protein samples of either co-transfected DYK-*M1AP*, HA-*SHOC1*, HA-*TEX11*, or HA-*SPO16*, 'input' samples of co-IP eluates, or singularly transfected "w/o bait" IP controls were pre-mixed 1:4 with 4× Laemmli sample buffer (Bio-Rad, #1610747) supplemented with DTT (Merck, #10197777001) and incubated at 95 °C, 10 min for denaturation. Protein separation was achieved using 4–15% mini-PROTEAN® TGX Stain-Free™ precast gels (Bio-Rad, #4568085) for SDS polyacrylamide gel electrophoresis (SDS-PAGE). Proteins were transferred to a PVDF membrane using the Trans-Blot® Turbo™ mini PVDF transfer packs (Bio-Rad, #1704156EDU) following the manufacturer's instructions. Subsequently, membranes were blocked with 5% milk powder solution in 0.025% TBS-Tween for 30 min at room temperature (RT). Incubation of primary antibody diluted in blocking solution was performed overnight at 4 °C. Antibody details are listed in Appendix Table S3. Between the incubation steps, washing with 0.1% TBS-Tween was included. Peroxidase-conjugated secondary antibody incubation followed. Visualisation was achieved by a chemiluminescence reaction using the Clarity™ Western ECL substrate kit (Bio-Rad, #1705060S) and the ChemiDoc MP imaging system (Bio-Rad, #12003154). The molecular weights of analysed proteins were calculated using a PageRuler™ plus prestained protein ladder (Thermo Scientific, #26619) and image processing was performed using the Bio-Rad image lab software and molecular

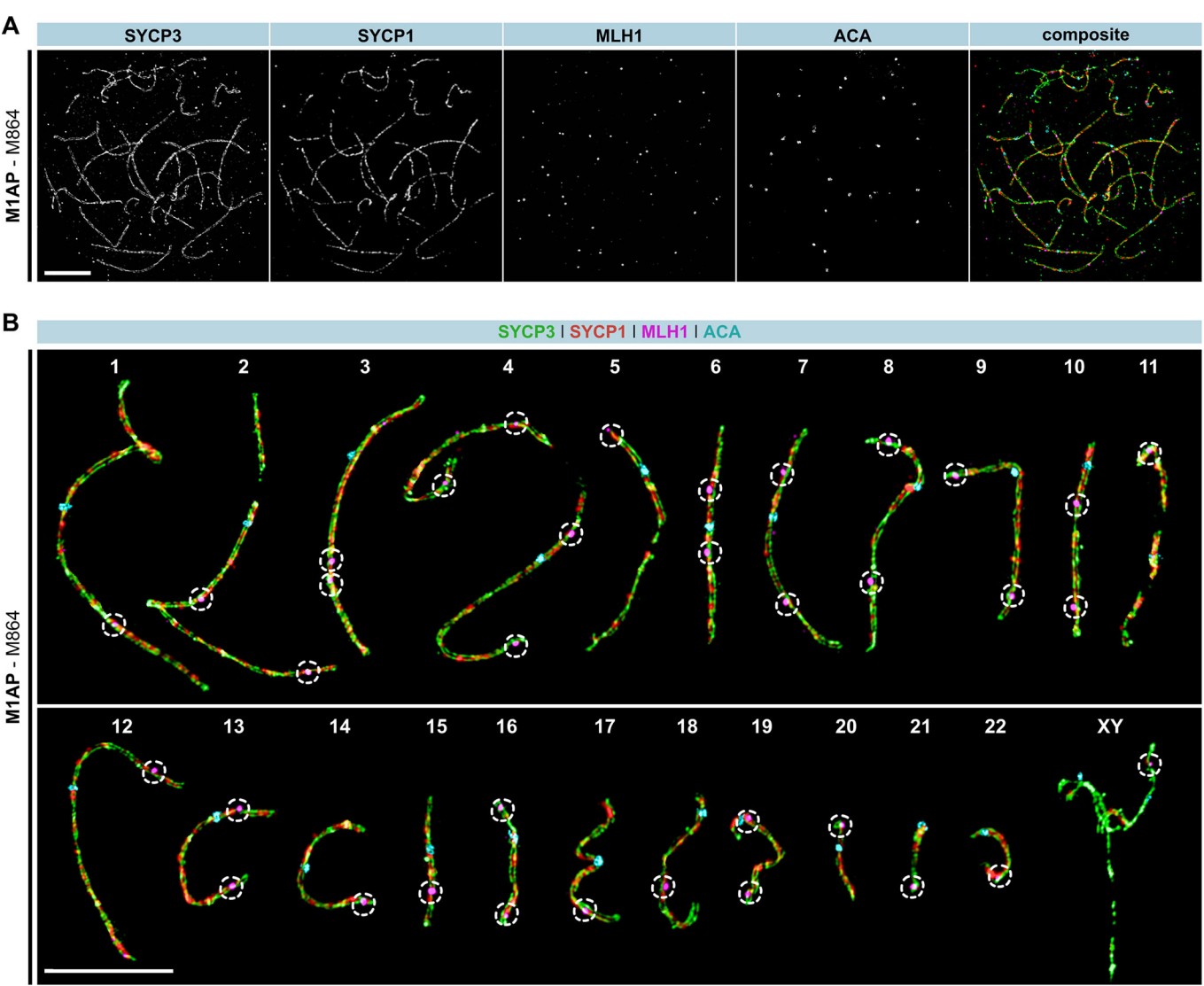

**Figure 9. A man with loss-of-function variant in M1AP showed at least one class I crossover per chromosome.**

(A) A pachytene spermatocyte of M864 deficient for M1AP was stained for SYCP3 (green), SYCP1 (red), ACA (centromere, cyan), and MLH1 (magenta). Panel is also seen in Fig. 5A. (B) Super resolution structured illumination microscopy (SR-SIM) of the spread spermatocyte shown in (A). Homologous chromosomes were digitally separated to mark the MLH1 foci (dotted circles), which represent the sites of class I crossover. In this spermatocyte, each pair of homologues has at least one MLH1 focus and, thus, the obligatory crossover principle is met. The scale bars represent 10 µm and 1 µm for magnification.

weight analysis tool (Bio-Rad, #12012931). Samples shown in one figure were derived from the same experiment.

### Site-directed mutagenesis

To specify the interaction between M1AP and the ZZS complex truncated constructs of M1AP were cloned using the QuikChange II XL Site-Directed Mutagenesis Kit (Agilent Technologies, USA) according to the manufacturer's instructions. Mutagenesis for point mutations inducing a stop codon was performed for the variants c.322 C > T  p.Gln108Ter  (Q108*),  c.1005 T > A  p.Cys335Ter (C335*) and c.1297 C > T, c.1298 T > A  p. Leu433Ter (L433*). Positions were selected in tolerant regions of the M1AP protein to prevent destruction of the protein's function. Sequences of

mutagenesis primers are found in Appendix Table S2. The construct of c.676dup p.Trp226Leufs*4 has been previously published (Wyrwoll et al, 2020). The truncated M1AP constructs were used for co-immunoprecipitation with full-length TEX11.

### Minigene splicing assay

The functional impact of the splice site variants *M1AP* c.1073_1074+10del and *SHOC1* c.1939+2 T > C was determined by in vitro splicing assays based on a case-specific minigene construct. Therefore, the affected region of interest was cloned with adjacent intronic sequences encompassing the variant of interest in a eukaryotic expression vector (pDESTsplice) suitable to analyse splicing events. The artificial minigene construct used in this study consists of two

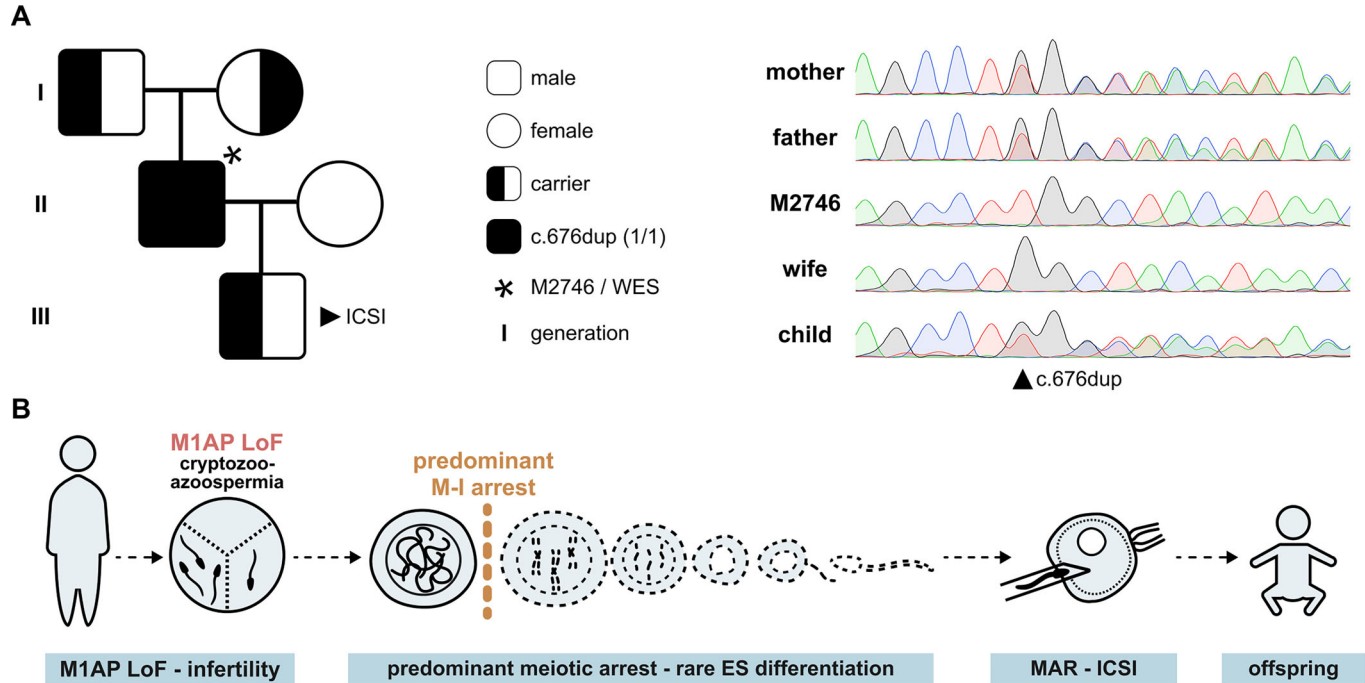

**Figure 10. Proof-of-principle: M1AP-associated infertility can be overcome by medically assisted reproduction (MAR).**

(A) One man (M2746) with a loss-of-function variant in *M1AP* was diagnosed with cryptozoospermia and predominant meiotic arrest. Three cycles of intracytoplasmic sperm injection (ICSI) with ejaculate-derived spermatozoa resulted in the birth of a healthy boy. Segregation analysis showed the autosomal-recessive inheritance pattern of the frameshift variant c.676dup. (B) Illustration of how predominant meiotic arrest caused by M1AP-associated infertility still enables fatherhood.

known exons of rat *Insulin 2*, exon 3 and 4, separated by intronic sequences. Using a two-step cloning technique results in the insertion of the region of interest into those intronic sequences separating the known exons. The region of interest was amplified from genomic DNA of the respective individual or a human male control sample using the standard PCR technique. Primers are listed in Appendix Table S2. 0.5 U/μL Phusion™ high-fidelity DNA polymerase (Thermo Scientific, # F530) was used for amplification according to the manufacturer's instructions. Cloning into a pENTR/D-TOPO® vector (Thermo Scientific, #K240020) was followed by LR recombinase reaction and gateway cloning using the Gateway™ LR clonase™ enzyme mix (Thermo Scientific, #11791020) and an ultimate pDESTsplice vector (a gift from Stefan Stamm (addgene plasmid #32484, Kishore et al, 2008). Subsequently, HEK293T cells were transiently transfected with 2 μg of wild-type or case-specific minigene DNA constructs using the K2 transfection reagent according to the manufacturer's instructions. After 24 h of transfection, total RNA was extracted using the RNeasy plus mini kit (Qiagen, #74134) and transcribed into cDNA using the ProtoScript® II first strand cDNA synthesis kit (New England Biolabs, #E6560). Amplification was conducted with primers annealing to rat insulin exon 3 and exon 4, respectively. Finally, RT-PCR products were separated on a 2% agarose gel, visualised with the TapeStation D1000 system (Agilent, #5067-5582), cut out and extracted, and confirmed by Sanger sequencing following standard protocols.

## Protein structure predictions

Presented protein structures (M1AP, SPO16) were predicted using AlphaFold2 from EMBL-EBI (Varadi et al, 2024; Jumper et al, 2021).

Images of protein structures were adapted in colour or variant consequences and exported from the EBI server.

## Histological evaluation of testicular biopsies

Testis biopsies of men with variants in *M1AP* or one of the ZZS genes and of control subjects were obtained from testicular sperm extraction (TESE) approaches or histological examinations at the Department of Clinical and Surgical Andrology of the CeRA, Münster, or at the Clinic for Urology, Gießen, and were included for in-depth histological phenotyping. Tissue samples were either snap-frozen, cryo-preserved (Sperm-Freeze™, Ferti-Pro, #3080), or fixed in Bouin's solution, paraformaldehyde (PFA), or GR fixative overnight. Fixed samples were washed with 70% ethanol and embedded in paraffin for routine histological examination. Tissues were sectioned at 5 μm and stained with Periodic acid-Schiff (PAS) or haematoxylin and eosin (H&E) according to standard protocols. If no differences between biopsies were observed, further analysis focused on one biopsy per case. In addition, the most advanced germ cell type was quantified, representing the percentage of elongated spermatids (ES), round spermatids (RS), spermatocytes (SPC), spermatogonia (SPG), Sertoli cell-only (SCO), and hyalinised tubules (tubular shadows, TS) per section.

## Immunohistochemical staining of human testicular tissue sections

For immunohistochemistry (IHC), biopsy samples were sectioned at 3 μm and incubated in Neo-Clear™ (Sigma Aldrich, #109843) for de-

paraffinisation. Rehydration was performed in a descending ethanol row (99%, 98%, 80%, and 70% EtOH, respectively). Between individual incubation steps, washing was performed using Tris-buffered saline (1× TBS). Incubation and washing steps were performed at RT, if not stated otherwise. Heat-induced antigen retrieval followed at 90 °C, using either sodium citrate buffer (pH 6, Thermo Scientific, #005000) or Tris-EDTA buffer (pH 9, Zytomed Systems, #ZYT-ZUC029), depending on the respective antibodies' specifications. Blocking of endogenous peroxidase activity and unspecific antibody binding was achieved by incubation in 3% hydrogen peroxidase ($H_2O_2$, Pharmacy of the University Hospital Münster, #1002187) for 15 min and in 25% normal goat serum (abcam, #ab7481) diluted in TBS containing 0.5% bovine serum albumin (BSA, Merck, #A9647) for 30 min. Primary antibodies were diluted in 5% BSA-TBS and incubated in a humid chamber at 4 °C overnight, if not stated otherwise (for antibody details, see Reagents and Tools Table). Round spermatid arrest was scrutinised by cAMP-responsive element modulator (CREM) expression. Apoptosis of germ cells was analysed by TUNEL assay, following the manufacturer's instructions (Thermo Scientific, #C10625). The progression of meiosis I was evaluated by phospho-histone H2A.X (γH2AX) staining. Metaphase cells were detected by phosphorylation of serine 10 on histone H3 (H3S10p). To discriminate spermatogonia, MAGE family member A4 (MAGEA4) localisation was performed in sequential slides with a distance of 3 μm. Incubation with unspecific immunoglobulin G (IgG) adapted to the primary antibody host system or omission of the first antibody (OC) served as negative controls (Appendix Fig. S12). A secondary, biotinylated goat anti-rabbit (abcam, #ab6012) or goat anti-mouse antibody (abcam, #ab5886) was incubated for one hour, followed by conjugation with streptavidin–horseradish peroxidase (HRP, Sigma Aldrich, #S5512) for 45 min. Peroxidase activity was visualised by 3,3'-diaminobenzidine tetrahydrochloride (DAB, Sigma Aldrich, #D5905) incubation for 1–20 min according to evaluation by microscope. The reaction was stopped in double-distilled water. Counterstaining was performed with Mayer's haematoxylin (Sigma Aldrich, #1092491000). Dehydration followed using increasing EtOH concentrations and finally, slides were cleared with Neo-Clear™ and mounted with glass coverslips using M-GLAS® liquid cover glass medium (Merck, #1.03973). Antibody details are provided in Appendix Table S3.

## Meiotic spermatocyte spreading

Analysis of the effects of LoF variants in *M1AP*, *SHOC1*, *TEX11*, and *SPO16* on meiosis was assessed on a single-cell level by meiotic spermatocyte spreading and subsequent immunofluorescence staining. For this, snap-frozen or cryo-preserved (Sperm-Freeze™, Ferti-Pro, #3080) human testis samples were used (*N* = 1 per gene [*M1AP*: M864, *SHOC1*: M2046, *TEX11*: M3409, *SPO16*: M3863]). The protocols described here are based on the established drying-down technique for mammalian meiocyte spreading by (Peters et al, 1997) and were further optimised with protocols described by (de Boer et al, 2009). Adjustments were made to adapt the protocols for the preparation of human testicular samples.

### Notes

1. Execute processing at room temperature (RT) if not stated otherwise.
2. To minimise cell loss, it is essential to perform all preparation steps fast yet precisely.

3. Whenever feasible, filter all solutions used during spermatocyte spreading through a 0.2-μm filter to minimise the presence of unspecific background dots.

### Specific equipment

1. Tweezers and forceps for dissecting testicular tissue, curved-ones are preferred.
2. Petri dish (100 mm, glass, with lid) for dissecting testicular tissue.
3. 50 ml syringes and 20-μm filter.
4. Humid chambers with flat base to dry and stain slides. Maintain a consistent humidity by allowing the chamber to equilibrate for at least one hour prior to use.
5. 100 ml beaker and aluminium foil.
6. Microscope slides (silane-coated adhesive slides): boil slides in ddH₂O for 20 min using a water bath; then allow them to air-dry in a dust-free environment. When properly stored, cleaned slides remain usable for several months. If contamination is suspected, reboil and freshly prepare slides as required.

### Solutions

- 1× Dulbecco's phosphate-buffered saline plus additives (D-PBS⁺): 50 ml D-PBS supplemented with 1.1 mM $Ca^{2+}$, 0.52 mM $Mg^{2+}$, and 0.05% sodium DL-lactate. Filter the solution through a 0.2-μm filter. Freshly prepare this solution for each experiment.
- 100 mM sucrose. Solution can be aliquoted and stored at −20 °C.
- 50 mM borate buffer; adjust to pH 9.2. Buffer can be aliquoted and stored at RT.
- 1 M NaOH.
- 0.08% (v/v) Photo-Flo solution, prepared with filtered ddH₂O.
- 1% (w/v) paraformaldehyde fixation buffer (PFA⁺): dissolve 1 g PFA in 90 ml ddH₂O. Add two drops of 1 M NaOH; the solution will appear turbid. Heat to 50 °C while stirring until the solution becomes clear. Cool on ice for 15 min. Add 2 ml borate buffer and adjust the pH to 9.2 using 1 M NaOH. Bring the final volume to 100 ml with ddH₂O. Filter the solution through a 0.2-μm filter. Add 150 μl Triton X-100 (0.15% (v/v)) and stir until fully dissolved. Transfer the entire buffer to a 100 ml beaker and cover it with aluminium foil until further use. Use within 8 h and prepare freshly each time.

### Protocol

1. Directly transfer the snap-frozen or cryo-preserved human testicular biopsies to a Petri dish with D-PBS⁺. Perform the following preparation steps at RT.
2. Mechanically mince seminiferous tubules in 100–500 μl D-PBS⁺ using two forceps to release the cells for about 5 min. Avoid excessive disruption to preserve nuclear integrity.
3. Transfer the cell suspension to a 15-ml tube and fill up with 5 ml D-PBS⁺. Invert gently and incubate for 5 min to allow tubular remnants and debris to sediment.
4. Collect the supernatant in a fresh 15-ml tube.
5. Centrifuge at 1000 rpm for 5 min at RT. Repeat once.
6. Discard the supernatant and resuspend the pellet in D-PBS⁺ to a final concentration of approximately $1.5 \times 10^7$ cells/ml. Typically, 10–500 μl of D-PBS⁺ is sufficient, depending on the size of the pellet. Avoid dense cell concentrations during spreading to ensure

well-separated meiocytes. A slightly opaque suspension indicates an adequate density. To minimise cell loss, omit cell counting at this stage.

7. Prepare individual droplets by mixing 10 µl of the cell suspension with 20 µl of 100 mM sucrose. A pre-cleaned slide offers an optimal surface for droplet preparation. Do not prepare more than three droplets simultaneously. If working at a slower pace, prepare each droplet freshly.

8. Prepare the microscope slides by dipping the whole slide in freshly prepared fixation buffer, ensuring that the bottom remains clean. A uniform and sufficient coating of PFA$^+$ on the slide surface is critical for optimal spreading.

9. Place one cell suspension droplet at one corner of the slide. Gently tilt the slide to disperse the solution—first horizontally, then vertically—and ensure even distribution. Do not move back and forth to avoid clumping of the cells/nuclei.

10. Incubate the slides in a humid chamber for 120 min with the lid closed. Then, partially open the lid and continue incubation for 45 min to initiate controlled drying. Finally, remove the lid completely and allow slides to dry fully for approximately 30 min at RT. Adhering to the recommended drying times is crucial to ensure optimal spermatocyte spreading and efficient antibody-antigen reactions.

11. Briefly wash the slides with 0.08% Photo-Flo for 10 s at RT. Transfer the slides to a slide rack and allow them to dry completely at RT.

12. Slides can be wrapped in aluminium foil and stored at −80 °C until further processing.

## Immunofluorescence staining on spermatocyte spreads

A sequential staining protocol was established to prevent cross-reactivity and to minimise background dots that could be misidentified for the characteristic foci of meiotic proteins.

Specific equipment

1. Humid chambers with flat base to dry and stain slides. Maintain a consistent humidity by allowing the chamber to equilibrate for at least one hour prior to use.
2. PAP liquid blocker pen to subdivide slides for different marker combinations.
3. Glass cover slips (60 × 24 mm, #1.5).
4. Nail polish to seal mounted slides for long-term storage.

Solutions

- 0.08% (v/v) Photo-Flo solution, prepared with filtered ddH$_2$O.
- 1x PBS, filtered through a 0.2-µm filter.
- Donkey blocking solution: 1× PBS supplemented with 5% (v/v) normal donkey serum, 5% (w/v) glycine, 0.3% (v/v) Triton-X, 0.01% (w/v) sodium acid (NaN$_3$), 0.05% (v/v) Tween. Adjust the pH to 7.4 using 1 M NaOH. Centrifuge at 15,000 × g for 15 min. Avoid using bovine serum albumin or milk powder when working with secondary antibodies originating from cloven hoofed animals.
- Goat blocking solution: 1× PBS supplemented with 5% (v/v) normal goat serum, 5% (w/v) glycine, 0.3% (v/v) Triton-X, 0.01% (w/v) sodium acid (NaN$_3$), 0.05% (v/v) Tween. Adjust the pH to 7.4 using 1 M NaOH. Centrifuge at 15,000 × g for 15 min.
- Human blocking solution: 1× PBS supplemented with 50 µg/ml anti-human Fab fragments.
- Antibody diluent: 1× PBS supplemented with 0.3% (v/v) Triton-X, 0.01% (w/v) sodium acid (NaN$_3$), 0.05% (v/v) Tween. Centrifuge at 15,000 × g for 15 min. This diluent was used to prepare all antibody solutions. Primary and secondary antibody details including applied concentrations are provided in Appendix Table S3.

Protocol

1. If slides were stored at −80 °C, thaw slides for 15 min at RT, leave in aluminium foil.
2. Use a hydrophobic barrier pen (e.g. PAP pen) to subdivide the slide for different antibody combinations.
3. Briefly wash the slides with 0.08% Photo-Flo for 10 s.
4. Wash the slides three times for 10 min in PBS under gentle shaking.
5. For minimising non-specific binding, cover the complete slide with 500 µl donkey blocking buffer and incubate for 30 min in a closed humid chamber at RT. Slides should never be allowed to dry out during the entire staining protocol.
6. Gently remove excess blocking solution and add 150 µl of primary antibody solution per slide. If the slide has been subdivided using a PAP pen, adjust the volume accordingly to ensure complete coverage of each section. A slide-sized sterile piece of parafilm can be used to cover slides during prolonged incubation steps. At this step, only rabbit, mouse, and goat primary antibodies should be included. Incubate the slides overnight at 4 °C, followed by 30 min at RT, and 15 min at 37 °C in a closed humid chamber.
7. Wash the slides five times for 10 min in PBS under gentle shaking.
8. Add 150 µl of the corresponding secondary antibody solution to each slide and incubate for 3 h at RT in a closed humid chamber.
9. Wash the slides three times for 10 min in PBS under gentle shaking.
10. Repeat blocking step with donkey blocking buffer for 15 min at RT.
11. Wash the slides three times for 5 min in PBS under gentle shaking.
12. Cover the slide with 500 µl goat blocking buffer and incubate for 15 min in a closed humid chamber at RT.
13. Gently remove excess blocking solution and add 150 µl human blocking buffer. Incubate for 60 min at RT in a closed humid chamber.
14. Gently remove excess blocking solution and add human anti-centromere antisera (ACA) solution. Incubate overnight at 4 °C, followed by 15 min at RT in a closed humid chamber.
15. Wash the slides three times for 5 min in PBS under gentle shaking.
16. Add 150 ml of anti-human fluorophore-conjugated secondary antibody solution per slide and incubate at 30 min at RT, followed by 60 min at 37 °C in a closed humid chamber.
17. Wash the slides five times for 10 min in PBS-T under gentle shaking.
18. Mount the slides using ROTI® FluorCare mounting medium and #1.5 coverslips. Once the mounting medium has dried, seal the coverslips with nail polish to ensure long-term preservation and compatibility with image acquisition. Optimal signal intensity is achieved when slides are imaged immediately after immunolabelling.

## Image acquisition, processing, and digital data generation

Immunohistochemical staining was captured with an Olympus BX61VS microscope and the corresponding scanner software VS-ASW-S6, the PreciPoint O8 scanning microscope system, or a Leica DM750 microscope and the Leica ICC50 HD camera. Immunofluorescence staining of meiotic spreads was complied with a Zeiss Elyra 7 microscope for specialised 3D structured illumination ($SIM^2$) and the Zeiss Zen black software (TEX11 and MLH1 labelled meiotic spreads) or a Leica DM6 B TL microscope and the LASX Software (RAD51 and MSH5 labelled meiotic spreads). Suitable filter sets (Zeiss Elyra 7: DAPI/GFP/TXR/Y5 or Leica DM6 B TL: Filtersystem A/D/I3/N2.1) were used to visualise fluorophore-based antibody staining.

Image processing was achieved with the open-source software Fiji by ImageJ (v2.3.0/1.54 h). For downstream processing of images, pictures were cropped to desired sizes to ensure a representative understanding of the testicular architecture or to allow focused visualisation of specific details. Meiotic progression was analysed by γH2AX localisation, and all tubules of one cross-section of each case were characterised by their most advanced prophase I stage. Quantification of staining was performed using the PreciPoint ViewPoint software and in-build measurement and counting tools (v1.0.0.9628). CREM- or TUNEL-positive cells were counted per round tubule per cross section and presented as average number of positive cells per tubule. Tubules were considered round when the ratio between the two diameters was in the range of 1–1.5. The minimum number of counted tubules per section was $N = 25$, the maximum number was $N = 177$. Class I crossover events and ZZS recruitment were quantified for pachytene spermatocytes spreads (only possible for control and M1AP), by counting the MLH1 or TEX11 foci, respectively, per one spermatocyte. MSH5 foci were quantified in pachytene (control and M1AP) or most progressed zygotene-like (SHOC1 and SPO16) spermatocytes. Proper chromosome preservation was confirmed by ACA staining. Prophase I stages were confirmed by SYCP1 or γH2AX staining, depending on the respective co-labelling.

### Statistical analysis

Data are presented as the mean with standard deviation for scatter plots or as mean with range for bar graphs. Given the limited sample sizes for each group, the Shapiro-Wilk test was used to assess the normality of the data. Results did not vary from normality ($P > 0.05$), supporting the assumption of a normal distribution. For subsequent analysis, an unpaired two-tailed Student's $t$ test was employed, appropriate for comparing independent samples between two groups using the GraphPad Prism software (v10.1.2). In cases where only one data point was available or values corresponded to zero, one sample $t$ test was applied instead, to assess whether the observed individual value significantly deviated from the control group's mean taking into account the control group's standard deviation and respective sample size. The difference was considered significant when the $P$ value was <0.05. The sample size selection was based on limited availability, our professional experience, and good laboratory practice. No statistical tests were used to predetermine the sample size. The researchers involved in the analysis and quantification of

**The paper explained**

**Problem**

Male infertility is a global health problem, yet the underlying causes often remain unexplained. If there are no sperm in the semen sample, men can still explore treatment options such as testicular sperm extraction (TESE) and medically assisted reproduction (MAR) to become fathers. However, if a defect in spermatogenesis, such as a block in meiosis known as meiotic arrest, leads to a lack of sperm, success rates are very limited. Understanding the causes of male infertility is therefore essential in order to better assess the potential benefits and risks of these treatments.

**Results**

Our study shows that pathogenic genetic variants in the human meiosis-associated gene *M1AP* result mainly in meiotic arrest at metaphase I, leading to male infertility. However, the homologous recombination process was still completed in certain cells. This included the formation and repair of DNA double-strand breaks, the establishment of early recombination nodules, full chromosome synapsis and reduced but intact resolution of class I crossovers. This allowed proper chromosome segregation and facilitated the occasional development of fertilisable sperm, as evidenced by the birth of a healthy euploid child. In contrast, the complete loss of one of the human ZZS proteins, SHOC1, TEX11 or SPO16, which interact with M1AP, resulted in a severe disruption of the recombination machinery, leading to early meiotic arrest and the complete absence of haploid cells. This is likely to exclude affected men from fatherhood.

**Impact**

Our findings highlight the clinical relevance of the *M1AP* gene and show that men with genetic variants in this particular gene may still have a chance of fatherhood. We also highlight the need for a comprehensive understanding of the pathophysiological mechanisms underlying male infertility. This knowledge is essential for evidence-based patient counselling and treatment options.

data were not blinded to the experimental conditions (Wyrwoll et al (2023a)).

## Data availability

Sequencing data of the MERGE study is available by contacting the Institute of Reproductive Genetics (https://reprogenetik.de). Access to this data is limited for each case and specific consent of the respective samples. All genetic variants from this study have been deposited in ClinVar, the corresponding accession numbers are provided in Appendix Table S4. Imaging datasets are available in the Open Microscopy Environment Remote Objects (OMERO) database under a public data group (https://doi.org/10.57860/min_prj_000011).

The source data of this paper are collected in the following database record: biostudies:S-SCDT-10_1038-S44321-025-00244-0.

## Peer review information

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

## Acknowledgements

We kindly thank all probands and their families for providing data, samples, and their contents which are the foundation of our interdisciplinary biomedical research. Thomas Zobel and the team from the Multiscale Imaging Centre, University of Münster is thanked for his excellent and professional support in SR-SIM microscopy (Elyra Zeiss Programmnummer INST 211/901-1 FUGB). The authors thank Celeste Brennecka for language editing. In addition, we thank the following people for their valuable and professional support: Pascal Hauser, Lena Schilling, Luisa Meier, Christina Burhöi, Alexandra Hax, Jochen Wistuba, Reinhild Sandhowe, Willy Baarends, Lieke Koordnneef, Esther Sleddens, Antoine Peters, Rita Exeler, Katja Poorthuis, Adelheid Kersebom, Elke Kößer, Sophie Koser, Claudia Krallmann, and Margot Julia Wyrwoll. This study was carried out within the frame of the German Research Foundation-funded Clinical Research Unit 'Male Germ Cells' (DFG CRU326, project number 329621271) to CF, FT, NN and the German Academic Exchange Service (DAAD) to FT and MOB (project ID 57511796).

## Author contributions

**Nadja Rotte**: Conceptualisation; Data curation; Formal analysis; Investigation; Visualisation; Methodology; Writing—original draft; Writing—review and editing. **Jessica E M Dunleavy**: Data curation; Formal analysis; Validation; Investigation; Methodology. **Michelle D Runkel**: Formal analysis; Investigation; Methodology. **Lina Bosse**: Investigation; Methodology. **Daniela Fietz**: Resources; Data curation. **Adrian Pilatz**: Resources; Data curation. **Johanna Kuss**: Investigation; Methodology. **Ann-Kristin Dicke**: Investigation; Methodology. **Sofia B Winge**: Investigation; Methodology. **Sara Di Persio**: Data curation; Investigation. **Christian**

Ruckert: Data curation; Software. **Verena Nordhoff**: Resources; Data curation. **Hans-Christian Schuppe**: Resources; Data curation. **Kristian Almstrup**: Resources; Data curation. **Sabine Kliesch**: Resources; Data curation. **Nina Neuhaus**: Resources; Data curation; Supervision; Funding acquisition. **Birgit Stallmeyer**: Data curation; Writing—review and editing. **Moira K O'Bryan**: Supervision; Funding acquisition; Writing—review and editing. **Frank Tüttelmann**: Conceptualisation; Supervision; Funding acquisition; Writing—review and editing. **Corinna Friedrich**: Conceptualisation; Data curation; Supervision; Funding acquisition; Validation; Writing—original draft; Project administration; Writing—review and editing.

Source data underlying figure panels in this paper may have individual authorship assigned. Where available, figure panel/source data authorship is listed in the following database record: biostudies:S-SCDT-10_1038-S44321-025-00244-0.

## Funding

## Disclosure and competing interest statement

The authors declare no competing interests.

# Expanded View Figures

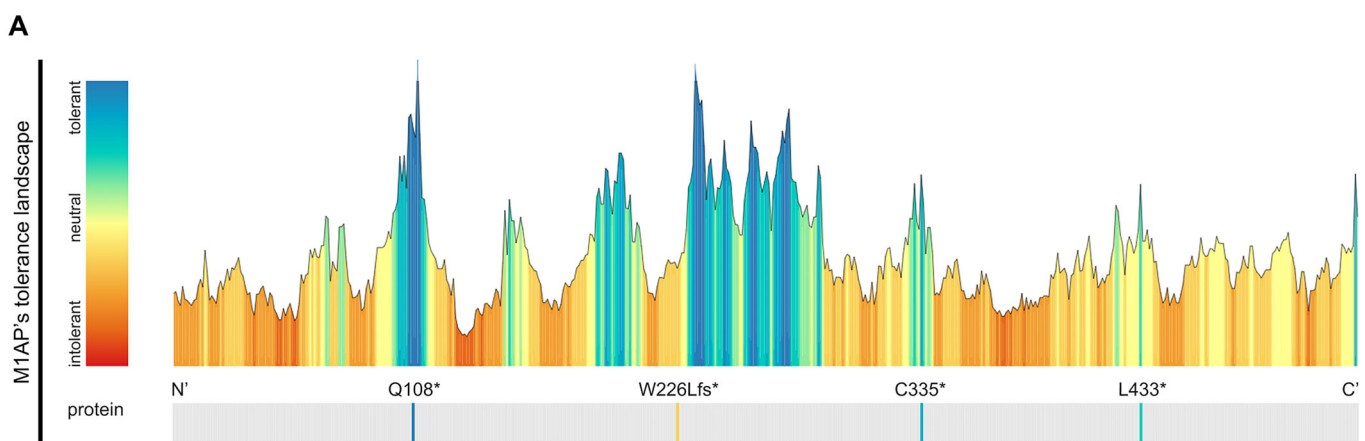

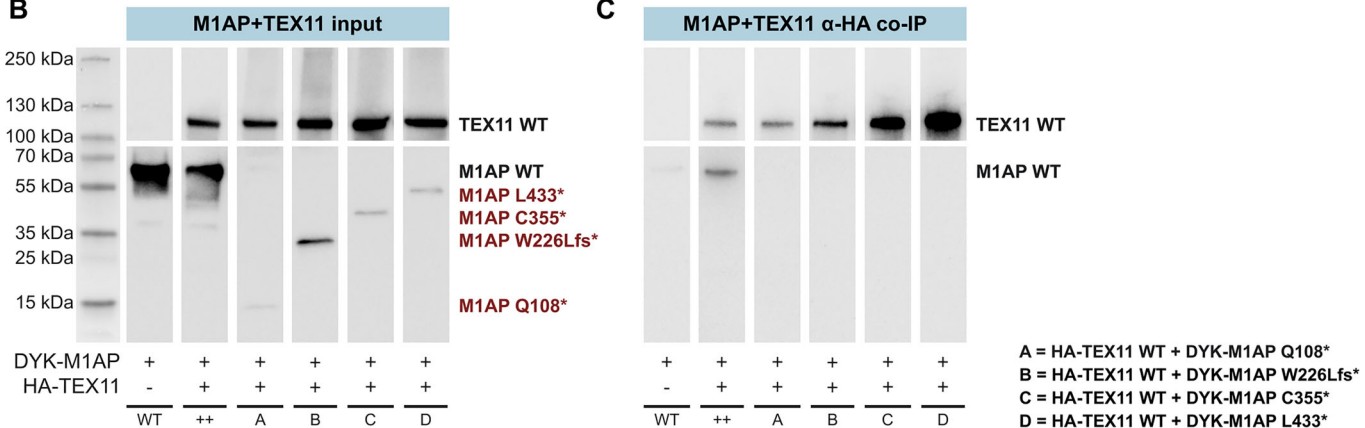

**Figure EV1. Full-length M1AP is mandatory for the protein-protein interaction with TEX11.**

(A) The protein tolerance landscape of M1AP illustrates the respective regions selected for mutagenesis for cloning truncated versions (Q108*: c.322 C > T p.Gln108Ter, C335*: c.1005 T > A p.Cys335Ter, L433*: c.1297 C > T, c.1298 T > A p. Leu433Ter). Positions were selected in tolerant regions (blue, green) of M1AP to prevent destruction of the protein's function. The construct c.676dup p.W226L*4 has been described in (Wyrwoll et al, 2020). (B) Western blot analysis of the input lysates of the co-transfection of full-length TEX11 (WT, detected by C-terminal HA tag) with full-length (WT) and truncated M1AP constructs (detected by N-terminal DYK-tag) confirm the expression in HEK293T cells. (C) Co-immunoprecipitation (IP) proved the interaction of human WT M1AP with WT TEX11. In contrast, no truncated M1AP was detected upon co-transfection with TEX11, pointing towards an absence of protein-protein interaction and thereby specifying the M1AP-TEX11 WT interaction. Experiments were replicated in three biological replicates. Source data are available online for this figure.

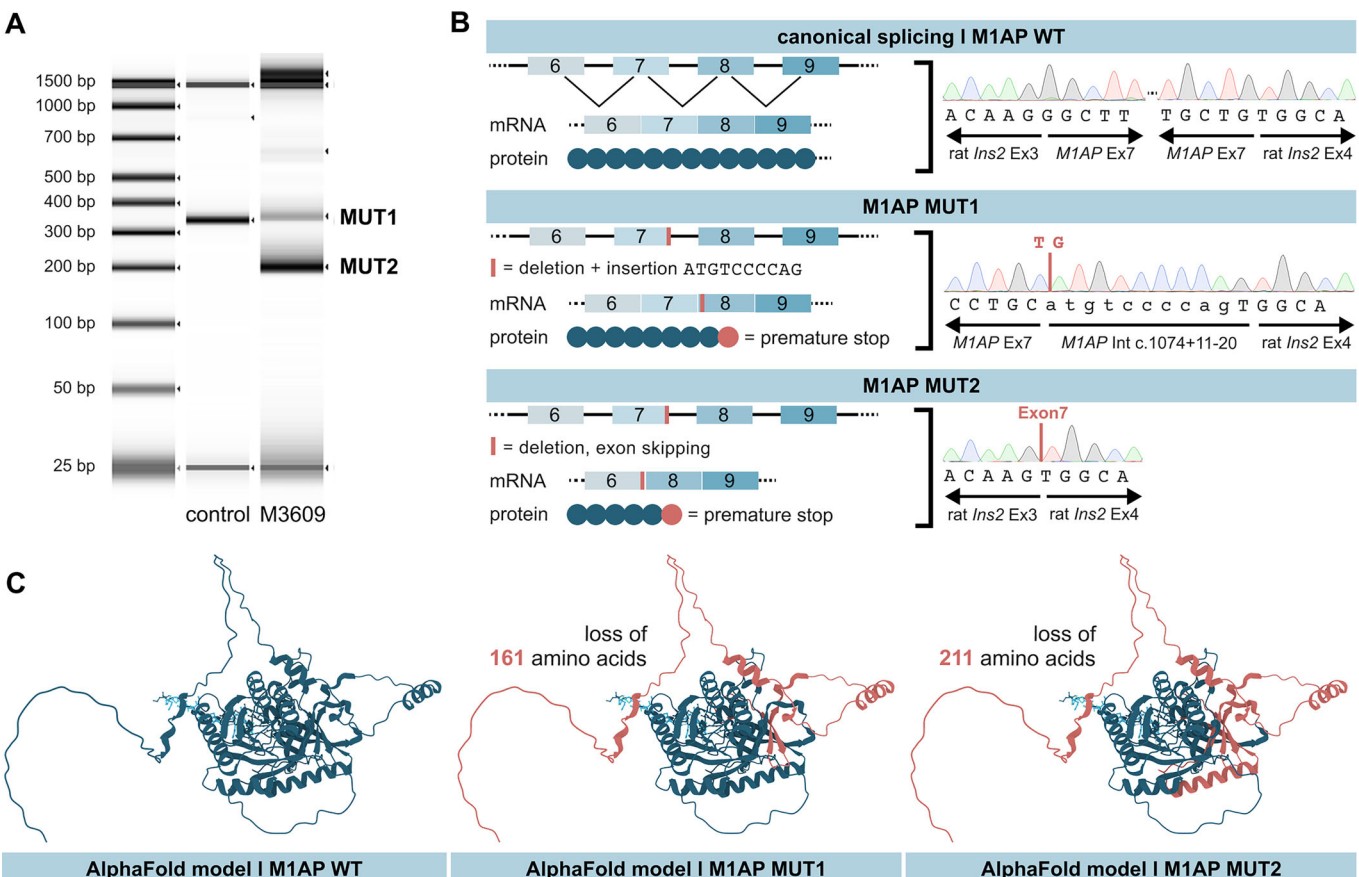

**Figure EV2. *M1AP* splice site variant identified in M3609.**

(A) Amplified minigene cDNA encompassing—c.1073_1074+10del or respective wild-type (control/WT) sequence. (B) Schematic illustration of variant effect on genomic, transcriptomic, and protein level combined with sequencing results for each minigene product reveals aberrant splicing. In the WT minigene construct, M1AP exon 7 (Ex7) is encompassed by two known exons of rat Insulin 2, exon 3 and exon 4 (rat Ins2 Ex3/Ex4). In M3609, the variant led to two splicing products: one (MUT1) showed the recognition of a cryptic splice site leading to a frameshift and premature stop codon in M1AP exon 8. The second (MUT2) resulted in skipping of exon 7 and a premature stop codon in exon 8. (C) Both splicing products lead to the loss of amino acids, presumably effecting M1AP's function and interaction.

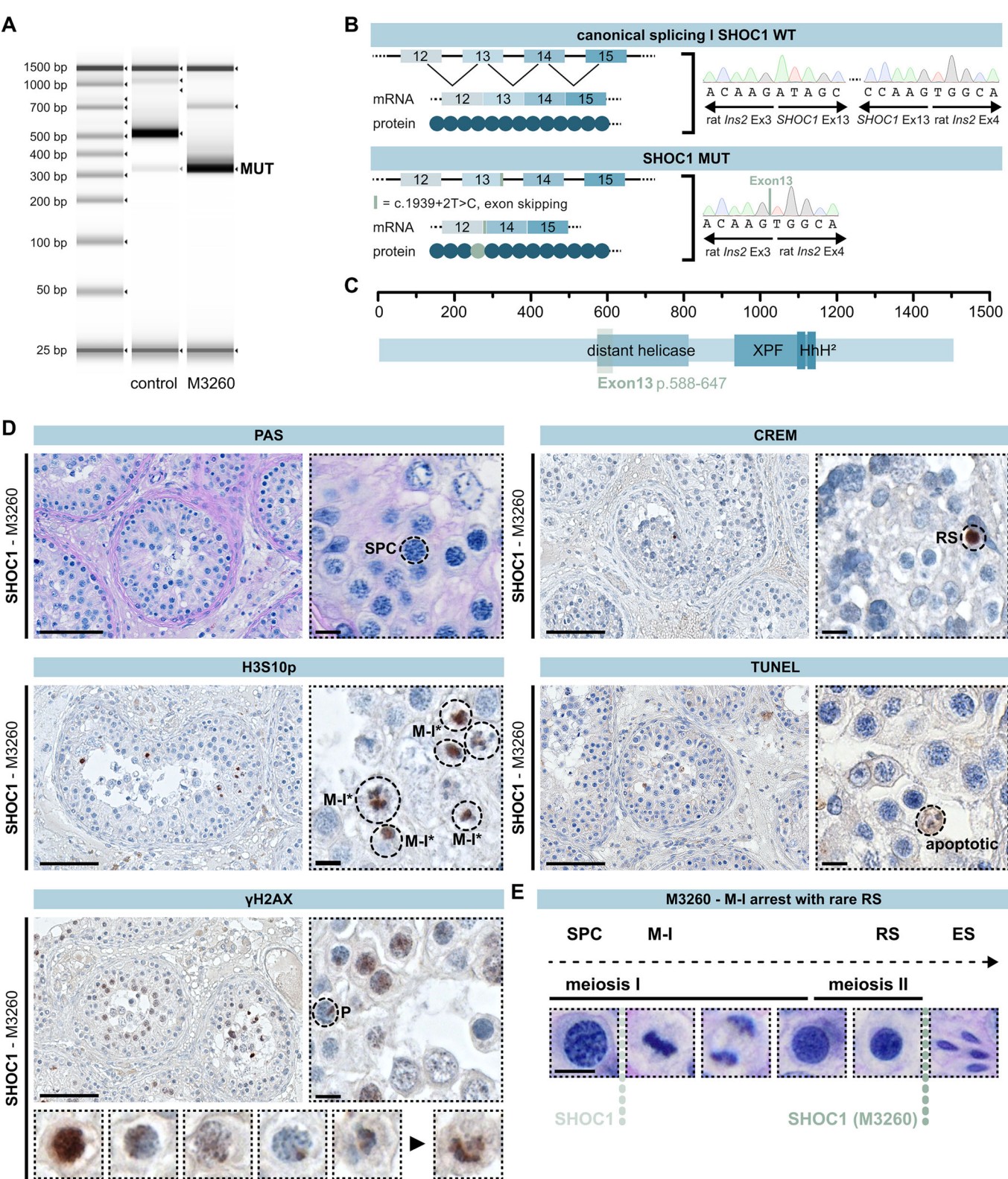

◀ **Figure EV3.** *SHOC1* splice site variant identified in M3260 with predominant meiotic arrest and rare round spermatids.

(**A**) *SHOC1* (NM_173521.5) has 26 exons and its corresponding protein comprises 1444 amino acids. Amplified minigene cDNA encompassing—c.1939+2 T > C (M3260) or respective wild-type (control/WT) sequence. (**B**) Schematic illustration of variant effect on genomic, transcriptomic, and protein level combined with sequencing results for each minigene product reveals aberrant splicing. In the WT minigene construct, *SHOC1* exon 13 (Ex13) is encompassed by two known exons of rat *Insulin 2*, exon 3 and exon 4 (rat *Ins2* Ex3/Ex4). In M3260, the variant resulted in in-frame skipping of exon 13 and a predictive loss of 59 amino acids representing 4% of the total protein. (**C**) This affects the distant helicase hits region but not the highly conserved 'SHOC1 homology region' (amino acids 937–1105, NP_775792; Macaisne et al, 2008). This region contains an XPF endonuclease-like central and a helix-hairpin-helix (HhH$^2$) domain and is important for the XPF-ERCC1-like complex formation between SHOC1 and SPO16 (De Muyt et al, 2018; Zhang et al, 2019). Yeast studies highlighted that the N-terminal part of Zip2 is linked to the chromosome axis and the other ZMM components through Zip4 interaction, while the XPF domain interacts exclusively with Spo16 (De Muyt et al, 2018). Given that M3260 expresses all exons of *SHOC1* except for exon 13, the interaction with SPO16 and in parts with the ZMM proteins, such as TEX11, remains intact. However, a changed protein conformation due to the loss of exon 13 could influence some of these interactions and explain the observed testicular phenotype (**D**) of predominant meiotic arrest with rare round spermatids that were positive for CREM-staining. H3S10 staining showed only aberrant metaphase I-like spermatocytes (M-I*). TUNEL staining showed an increased number of apoptotic spermatocytes similar to patients with complete LoF variants in *M1AP*, *SHOC1*, *TEX11*, or *SPO16*. In γH2AX staining, single tubules contained pachytene-like cells (P) with a clearly distinguishable XY body were observed, which is in line with the presence of round spermatids. In addition, also aberrant pachytene-like cells were present. (**E**) The specific type of arrest of M3260 is described as a metaphase I arrest (MM-I) with rare round spermatids (RS) (panel taken from Fig. 2C). The scale bar represents 100 μm and 10 μm, respectively.

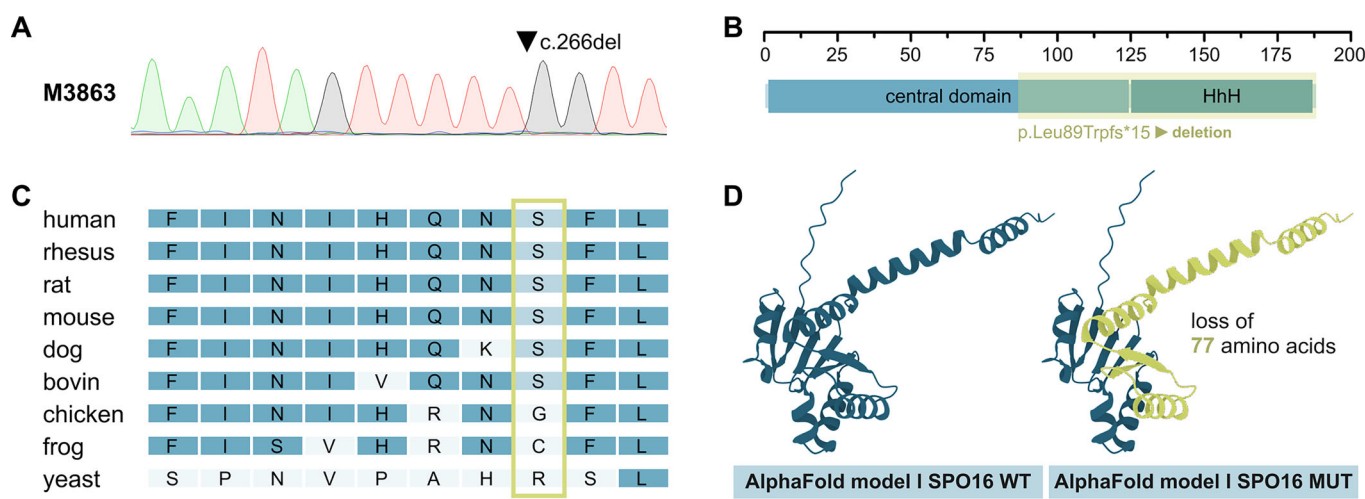

**Figure EV4.   *SPO16* loss-of-function variant identified in M3863.**

(**A**) Sanger sequencing of M3863 revealed the frameshift variant c.266del leading to premature stop codon (p.Leu89Trpfs*15). (**B**) Such a truncated protein would lack the highly conserved helix-hairpin-helix (HhH[2]) domain, which is important for the XPF-ERCC1-like complex formation between SHOC1 and SPO16 (De Muyt et al, 2018; Zhang et al, 2019). (**C**) Conversation analysis of the SPO16 variant. (**D**) The premature stop codon would truncate 42.5% of the complete protein.

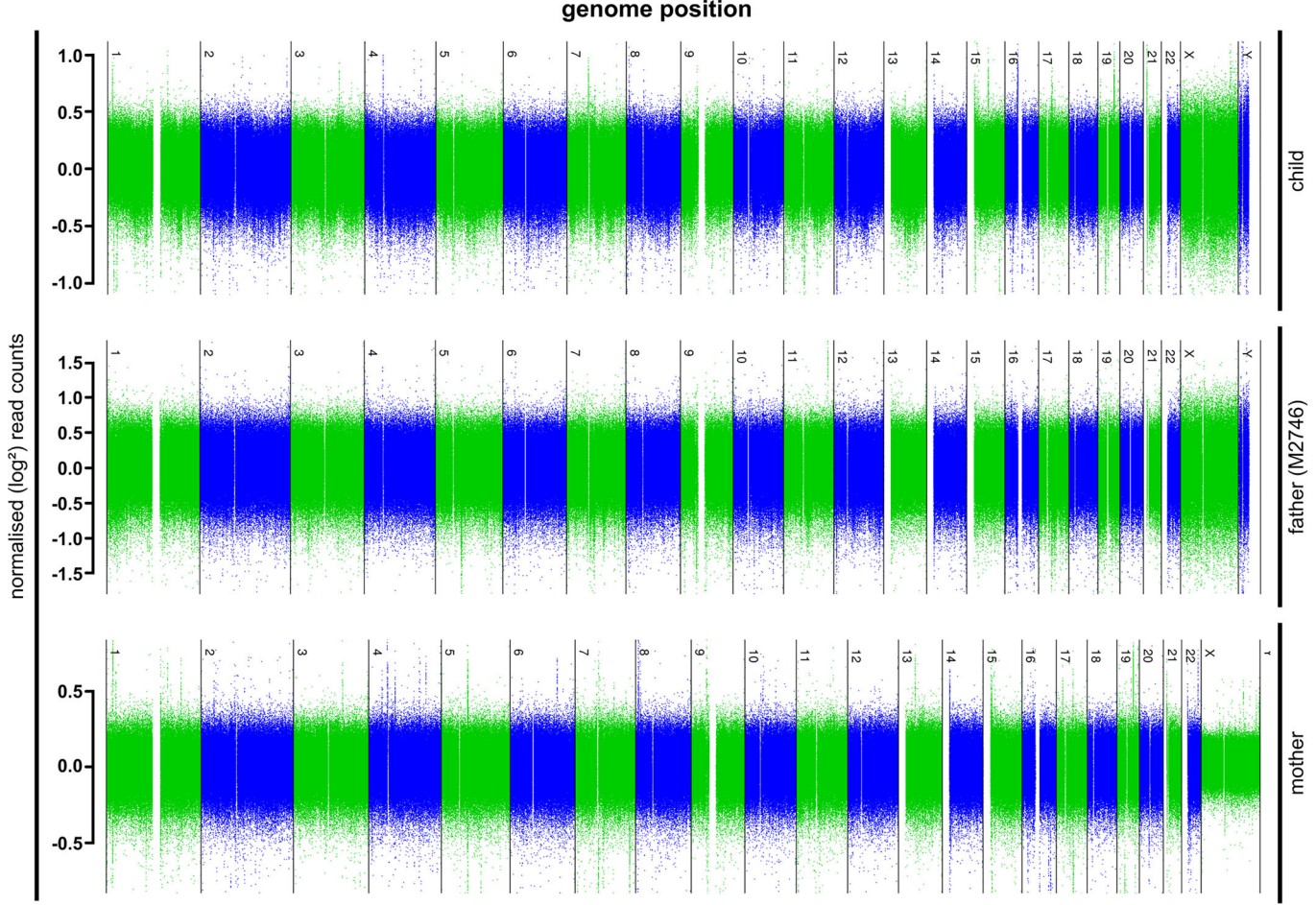

**Figure EV5. Euploidy analysis of M2746, his child, and the child's mother.**

Genome sequencing data was queried and read counts were normalised by dividing the median read count of each chromosome by the median read count of all autosomes. Normalised (log$^2$) read counts of autosomes (0.97 to 1.06) and of gonosomes (0.49 to 0.51) gave no evidence for chromosome aneuploidies.

