## [Peer Review File · EMBO Molecular Medicine]

Genotype-specific differences in infertile men due to loss-of-function variants in M1AP or ZZS genes

Nadja Rotte, Jessica Dunleavy, Michelle Runkel, Lina Bosse, Daniela Fietz, Adrian Pilatz, Johanna Kuss, Ann-Kristin Dicke, Sofia Winge, Sara Di Persio, Christian Ruckert, Verena Nordhoff, Hans-Christian Schuppe, Kristian Almstrup, Sabine Kliesch, Nina Neuhaus, Birgit Stallmeyer, Moira O'Bryan, Frank Tüttelmann, and Corinna Friedrich

Corresponding author: Corinna Friedrich (corinna.friedrich@ukmuenster.de)

Review Timeline:	Transfer from Review Commons:	11th Jul 24
	Editorial Decision:	15th Jul 24
	Revision Received:	15th Feb 25
	Editorial Decision:	13th Mar 25
	Revision Received:	18th Apr 25
	Accepted:	24th Apr 25

Editor: Zeljko Durdevic

Transaction Report:

This manuscript was transferred to EMBO Molecular Medicine following peer review at Review Commons.

Review #1**1. Evidence, reproducibility and clarity:****Evidence, reproducibility and clarity (Required)**

This manuscript compiles LoF variants of M1AP and ZZS proteins (i.e., SHOC1, TEX11 and SPO16) that almost certainly underlie infertility and reports the first case of an infertile man homozygous for a variant in SPO16. The authors validated interactions between human M1AP and ZZS that were found in mice. Analyzing testicular samples from infertile men revealed that those with deficiencies in SHOC1, TEX11 or SPO16 exhibited early meiotic arrest without haploid germ cells, whereas those with M1AP variants displayed a predominant metaphase I arrest with rare haploid germ cells. Further investigations showed that disrupted SHOC1, TEX11 or SPO16 led to defective synapsis and pairing of homologous chromosomes and unpaired DNA DSBs, while M1AP mutations reduced CO events. Importantly, men with LoF variants in M1AP can father healthy children by medically assisted reproduction. Overall, the results are clear and convincing in defining likely causative variants in infertility patients.

I have a few minor comments for improving the manuscript:

- No statistical analyses were performed. The meaning of error bars was not mentioned. It is essential to specify the minimum number of seminiferous tubules counted for each patient.
- Allele frequencies of variants are not provided.
- Figure 4 should clearly label the representations of each color channel.
- The authors should clearly label the bands of SPO16 in the right panel of Figure 1B.
- Appendix Figure S1B and S2B, what does "rat" mean in "rat Ins2 Ex3/4"?

2. Significance:**Significance (Required)**

Overall, this study significantly contributes to the understanding of some genetic causes of human infertility and offers a potential avenue to treat patients with M1AP variants/mutants. Since no knock-in animal model was applied to mimic the subtle phenotype variations observed in patients, the functionality of truncated proteins remains unexplored. For example, it is unclear why the germ cells in patient M3260 with the SHOC1 variant can progress to round spermatids (Fig. 2C), while those in Shoc1 KO mice (10.1093/molehr/gaac015) and other patients cannot. However, this is a minor concern.

Our field of expertise is gametogenesis and meiosis in mice.

3. How much time do you estimate the authors will need to complete the suggested revisions:

Estimated time to Complete Revisions (Required)

(Decision Recommendation)

Less than 1 month

4. Review Commons values the work of reviewers and encourages them to get credit for their work. Select 'Yes' below to register your reviewing activity at Web of Science Reviewer Recognition Service (formerly Publons); note that the content of your review will not be visible on Web of Science.

Yes

Review #2

1. Evidence, reproducibility and clarity:

Evidence, reproducibility and clarity (Required)

****Summary:****

This interesting manuscript provides evidence for the biological and clinical relevance in human males of mutations in genes encoding M1AP and other related proteins. In mice, M1AP, "meiosis 1 associated protein," is known to associate with several proteins (SHOC1, TEX11, and SPO16) in the ZZS complex that promotes DNA recombination and crossover formation during meiosis I prophase. Mutation of these proteins in model organisms disrupts the process of recombination and cause arrest of spermatocytes prior to the first meiotic division. Here the authors took advantage of their MERGE (Male Reproductive Genomics) cohort to screen for human loss-of-function (LoF) mutations in the relevant ZZS complex and M1AP genes and to associate these with human male reproductive phenotypes. They found that men with deficiency of ZZS proteins SHOC1, TEX11 or SPO16 genes were infertile, exhibiting arrest of germ cell development early in meiotic prophase, with aberrations of chromosome synapsis and failure to repair DNA double-strand breaks (DSBs). In interesting contrast, men with M1AP mutations exhibited metaphase arrest, and

indeed, in some cases, produced haploid spermatids, which in medically assisted reproduction (ICSI), led to the birth of offspring. Because they demonstrate that M1AP interacts with the other proteins, the authors conclude that M1AP is a "catalyzer," but not essential, for the processes of synapsis, recombination, and formation of haploid gametes.

****Major Comments:****

The work is clearly presented with detailed methods that should allow adaptation in other laboratories.

Overall, this study is a tour de force with what was no doubt difficult archival samples. The histology is generally of good quality, supporting the conclusions about progress of meiotic prophase in the mutant samples. The images of H&E-stained tissue are particularly striking, especially those in supplemental figures. That said, and with particular reference to Fig. 3A, it is difficult to sub-stage meiotic prophase by immunocytochemistry, even in optimal samples, with only one marker (in this case gH2AX). The staging here is also at odds with the statement in the subsequent section (and Fig. 4B) on absence of pachytene cells in men with mutation of SHOC1, TEX11, or SPO16. Because precise stages of arrest probably cannot be determined in these samples, the authors would be wiser to use phrases such as "zygotene-like." The authors should also clarify how it was confirmed that the metaphase-like cells were spermatocytes and not spermatogonia (given that gH2AX signal is weak or unclear in some such nuclei). Readers with a focus on the more regularly staged mouse or rat tubules would appreciate a few more guidelines to criteria for staging human tubules.

Evidence for the birth of a (healthy) child from one individual with M1AP mutation verges on the anecdotal (N=1). It is interesting but raises multiple questions and concerns about both the frequency of chromosomal abnormalities in such individuals and the transmission of the mutant alleles.

The authors conclude that the M1AP protein is an essential "catalyzer" in the meiotic recombination pathway. However, it is not clear from the data presented that M1AP in fact has enzymatic catalysis activity or exactly when and how it participates. Because the word "catalyzer" is not buttressed with hard or convincing evidence, the authors should consider other ways to describe the proposed role of M1AP, perhaps as a "putative component" and/or "modifier" of the recombination pathway.

****Minor comments:****

Fig. 1A - these are nice illustrations, but overly simplified with respect to timing (synapsis is not completed in zygonema)

Fig. 1B - greater clarity in legend would be appreciated

Figs. 2A & 3A - colors in bar graphs are difficult to discriminate

Fig. 4A - with full appreciation for the difficulty with this material, the images are of low contrast and require considerable enlargement

2. Significance:

Significance (Required)

This is a very interesting paper, which I evaluated from the perspective of a reproductive geneticist with expertise in meiosis and interest in infertility. I think this report will be of interest to clinicians because it identifies a gene possibly linked to marginal fertility and establishes human protein interactions similar to those previously identified in mice. It reinforces the importance of ZZS genes in humans. The contributions of this report to the field of meiosis confirm previous evidence on M1AP, including mutant phenotypes and protein interactions, extending them to humans. We can thus appreciate the conserved function of the mammalian M1AP protein, but as yet the molecular mechanisms of M1AP are not clarified.

3. How much time do you estimate the authors will need to complete the suggested revisions:

Estimated time to Complete Revisions (Required)

(Decision Recommendation)

Less than 1 month

Yes

Review #3

1. Evidence, reproducibility and clarity:

Evidence, reproducibility and clarity (Required)

In this manuscript, Rotte et al. investigate the meiotic molecular function in human of the M1AP protein and of the ZSS complex (SCHOC1, TEX11 and SPO16 proteins). The ZSS complex is a key player of meiotic recombination. It is a sub-complex of the conserved family of the ZMM proteins, essential for the formation of class I crossovers, a proper chromosomes segregation and fertility. Understanding its mode of action, regulation and conservation in human is thus a crucial issue in the fields of meiosis and human reproduction, with potential implications for patients. In that context, the recent identification of the protein M1AP as a partner of the ZSS proteins raise the question of its role, function and conservation. The aim of this study is thus of primary importance.

To perform this molecular characterization, the authors made a cohort (24 total) of men carrying LoF variants in M1AP and ZSS genes. They performed a molecular biology analysis to assess the physical interaction between the human M1AP protein and the three components of the ZSS complex. Their results confirm a previous work performed in mice, mentioned by the authors.

Then, they took advantage of available biopsies from different mutant men to perform a histological and cytological analysis of the impact of the different mutations on meiosis. The main conclusions are that in human, similarly to what is known in different organisms (ranging from yeast to mice), the ZSS complex is essential for crossover formation, synapsis and spermatogenesis, and that defect in the genes is associated with a premature prophase I arrest and no sperm formation. The authors also showed that M1AP protein plays a role in meiotic progression, but to a lesser extent compare to the ZSS proteins, with a metaphase I arrest, an undetectable recombination phenotype, apart of a reduced crossover number and, spermatozoa can form in its absence.

****Major points:****

The authors investigate the physical interaction between M1AP and the ZSS members through a single approach: Co-IP of tagged proteins after expression in human HEK293T cells. This approach is informative, but to reinforce the conclusions the authors should provide data from independent approaches: yeast two hybrid, expression of recombinant proteins followed by pull down, co-immunostaining (TEX11 antibodies were used in the study and M1AP antibody is present in the literature) are possible non-exclusive approaches to decipher, more in details, the interaction. Moreover, understanding the hierarchy of interactions appears important to understand its rational, regulation and

function. What is the meaning of a M1AP interaction with all the members of the complex? Remains an open question.

The ZZS mutants have a defect in γ H2AX pattern, a defect in synapsis and no MLH1 foci, associated to apoptosis and prophase I arrest. M1AP mutation has a minor impact. The characterization of the effect of the different mutations (in particular M1AP) on the recombination process should be addressed further, by cytological means. For example, effect on strand invasion and ssDNA production should be monitored using RPA, DMC1 and RAD51 antibodies. The impact on alternative resolution pathway (e.g. BLOOM dependent) should be tested as well as the effect on other ZMM proteins, in particular MSH4-5, should be investigated. These experiments are essential to characterize, at the molecular level, the function of the different proteins during recombination.

In the same line, TEX11 staining in M1AP mutant should be more documented and in particular the different stages shown, as well as the foci counting, to have a quantitative result, that can be compared to MLH1. Moreover, co-immunostaining of different markers with TEX11: RPA, DMC1, MSH and MLH1 are also important to understand how the pathway is perturbed and the recruitment delayed/affected.

The published M1AP antibody should be tested to investigate its perturbation in the absence of the ZZS proteins and the hierarchy of event.

OPTIONAL: the obligatory crossover was measured, a comment or calculation of interference would be very interesting, and it seems doable using the MLH1 counting, to test whether these mutants have an effect on this process.

****Minor comments****

As written, the title is misleading, the paper does not investigate the impact of M1AP in ZSS recombination. Such study implies to study genetic interactions or the genetic dependency between the different proteins, which is not the case here.

Labelled on histological images is not clear. The authors should clearly explain to what marker each staining correspond.

L67 to 72: the authors should update and use more accurate citations for meiotic recombination.

L76: the ZMM are specifically involved in the resolution of class I crossover. Please

rephrase.

L94: Strictly, the author identified an interaction, they didn't establish how the interaction takes place.

FigS13: TEX11 staining should be presented with foci counting as a main figure.

L255: MLH1 does not quantify homologous recombination but, class I crossovers.

L352: The sentence is hard to understand, rephrase please.

2. Significance:

Significance (Required)

In general, the paper is well written and easy to follow. However, in light of the importance of the questions for the field of meiosis, it currently seems a little superficial, in particular if the authors aim at addressing the molecular function of the different proteins. The role of the ZSS proteins and M1AP in the control of meiotic recombination, at the molecular level is very important to decipher and additional experiments might help to better address this question. In addition, the functional links between M1AP and ZSS remains unclear and to investigate further.

This study gives information for human process, and can be compared to more advanced work done with mice.

This study will be important for the community working on meiosis in mammals, but also for people interested in reproduction.

3. How much time do you estimate the authors will need to complete the suggested revisions:

Estimated time to Complete Revisions (Required)

(Decision Recommendation)

Between 3 and 6 months

4. Review Commons values the work of reviewers and encourages them to get credit for their work. Select 'Yes' below to register your reviewing activity at Web of Science

Reviewer Recognition Service (formerly Publons); note that the content of your review will not be visible on Web of Science.

Yes

Revision Plan

Manuscript number: RC-2024-02439

Corresponding author(s): Corinna Friedrich

[The "revision plan" should delineate the revisions that authors intend to carry out in response to the points raised by the referees. It also provides the authors with the opportunity to explain their view of the paper and of the referee reports.]

The document is important for the editors of affiliate journals when they make a first decision on the transferred manuscript. It will also be useful to readers of the reprint and help them to obtain a balanced view of the paper.

*If you wish to submit a full revision, please use our "Full Revision" template. **It is important to use the appropriate template to clearly inform the editors of your intentions.**]*

1. General Statements [optional]

This section is optional. Insert here any general statements you wish to make about the goal of the study or about the reviews.

We gratefully thank all three reviewers for the thorough and thoughtful evaluation of our work and appreciate their comments to improve our manuscript.

The goal of our study was to identify genotype specific characteristics of the testicular phenotypes of infertile men, who are affected by pathogenic loss-of-function variants in *M1AP* and the ZZS protein encoding genes *SHOC1*, *TEX11*, and *SPO16*. *M1AP* interacts with the ZZS proteins in mice and in human; the latter was shown in our study.

By an in depth characterisation of available patients' testicular material using (immuno-) histological and immunofluorescence approaches, we described genotype specific differences separating *M1AP* from the ZZS proteins: in contrast to the ZZS proteins, *M1AP* is dispensable for class I crossover formation during meiotic recombination but significantly enhances this process. Thus, production of sperm and biological fatherhood is solely compatible with a deficiency of *M1AP*, as shown in a proof-of-principle.

Based on our findings, patient counselling and risk assessment for offspring will need to be re-evaluated and adapted.

Peer review comments:

We are thankful to the reviewers for their overall positive evaluation of our work. We are convinced that our manuscript will benefit from the constructive peer-review of all three reviewers.

Based on the reviewers' suggestions, we already addressed a number of the concerns and additional experiments are ongoing. We incorporated all changes requested by Reviewer #1. In addition, we also considered the text and figure work to improve our manuscript as suggested by Reviewers #2 and #3.

Revision Plan

The full comments by the reviewers and our detailed responses, as set out in the revision letter, are presented in blue below a brief description of each item.

2. Description of the planned revisions

Insert here a point-by-point reply that explains what revisions, additional experimentations and analyses are planned to address the points raised by the referees.

To provide transparency of our work and beyond the Reviewers' suggestions we plan to upload all histological scans that were exemplary shown in the main and appendix figures in the microscopy image repository OMERO/Open Microscopy Environment (OME).

As requested by Reviewer #2 we will:

- Perform additional IHC staining of γ H2AX and MAGEA4 on sequential sections to confirm that the metaphase-like cells were indeed spermatocytes

Detailed response:

Reviewer #2 comment:

The authors should also clarify how it was confirmed that the metaphase-like cells were spermatocytes and not spermatogonia (given that γ H2AX signal is weak or unclear in some such nuclei).

Response: We thank the reviewer for raising this point. In order to confirm that the metaphase-like cells were indeed spermatocytes we will perform additional IHC staining for γ H2AX and MAGEA4 on sequential testis sections (distance 3 μ m) on representative samples of the patient cohort as well as controls as the hosts of both antibodies are mice.

As requested by Reviewer #3 we will:

- Verify the specificity of the co-immunoprecipitation by exploring the protein-protein-interaction of full-length and truncated constructs of M1AP with TEX11.
- Quantify TEX11 foci in spermatocyte spreads and compare the number to MLH1. We will present the TEX11 staining in spermatocyte spreads with foci counting in a new arranged Main Figure 4.
- Establish new immunofluorescence markers specific for meiotic proteins.

Revision Plan

Detailed response:

Reviewer #3 comment:

Major points:

The authors investigate the physical interaction between M1AP and the ZSS members through a single approach: Co-IP of tagged proteins after expression in human HEK293T cells. This approach is informative, but to reinforce the conclusions the authors should provide data from independent approaches: yeast two hybrid, expression of recombinant proteins followed by pull down, co-immunostaining (TEX11 antibodies were used in the study and M1AP antibody is present in the literature) are possible non-exclusive approaches to decipher, more in details, the interaction. Moreover, understanding the hierarchy of interactions appears important to understand its rational, regulation and function. What is the meaning of a M1AP interaction with all the members of the complex? Remains an open question.

Response: We thank Reviewer #3 for this comment. In an independent approach we aimed to specify the interaction of M1AP to the ZSS proteins. Thus, we already cloned truncated versions of *M1AP* to refine the binding site of M1AP to the ZSS proteins (Figure R1).

Figure R1 Tolerance landscape of *M1AP* NM_001321739.2 illustrating the respective regions selected for mutagenesis of truncated *M1AP* constructs. Adapted from MetaDome.

In a preliminary experiment, we co-transfected full-length as well as the truncated constructs of M1AP with TEX11 and showed via Co-IP that the interaction is only possible with full-length M1AP. Within the full-revision, we plan to finalise these experiments and thus validate the specificity of the interaction between M1AP and TEX11 and thereby gain more insight into the interaction/hierarchy of the interaction of M1AP with the ZSS complex.

Revision Plan

Reviewer #3 comment:

The ZZS mutants have a defect in γ H2AX pattern, a defect in synapsis and no MLH1 foci, associated to apoptosis and prophase I arrest. M1AP mutation has a minor impact. The characterization of the effect of the different mutations (in particular M1AP) on the recombination process should be addressed further, by cytological means. For example, effect on strand invasion and ssDNA production should be monitored using RPA, DMC1 and RAD51 antibodies. The impact on alternative resolution pathway (e.g. BLOOM dependent) should be tested as well as the effect on other ZMM proteins, in particular MSH4-5, should be investigated. These experiments are essential to characterize, at the molecular level, the function of the different proteins during recombination.

Response: We thank the reviewer for this suggestion and highly appreciate to investigate the different pathways in more depth. We plan to perform additional immunofluorescence staining of spermatocyte spreads of identified patients compared to the control for a better understanding of M1AP within human recombination. We already ordered the antibodies against meiotic marker proteins as suggested by the reviewer.

We would like to take the opportunity to refer to the extremely limited access to cryopreserved testicular material of the patients presented in this manuscript: for each gene (*M1AP*, *SHOC1*, *TEX11*, *SPO16*) we were lucky to get one testicular biopsy specimen from one man for only one preparation of spermatocyte spreads. We hope for the reviewer's understanding that we cannot address each requested staining albeit this would be of highest interest not only for ourselves but also for the scientific community. However, we are very confident that we will provide additional staining added to the yet shown to improve the understanding of M1AP's function on human male meiotic recombination.

Reviewer #3 comment:

In the same line, TEX11 staining in M1AP mutant should be more documented and in particular the different stages shown, as well as the foci counting, to have a quantitative result, that can be compared to MLH1. Moreover, co-immunostaining of different markers with TEX11: RPA, DMC1, MSH and MLH1 are also important to understand how the pathway is perturbed and the recruitment delayed/affected.

Response: In the planned revision, we will include the TEX11 foci counting using the acquired images that will be compared to MLH1 foci quantification. In addition, we plan additional co-immunostaining of TEX11 with different markers dependent on the availability of testicular material.

Reviewer #3 comment:

Minor comments:

FigS13: TEX11 staining should be presented with foci counting as a main figure.

Response: We plan to restructure Figure 4 along with the new meiosis specific markers and present TEX11 staining and foci counting as main figure.

3. Description of the revisions that have already been incorporated in the transferred manuscript

Please insert a point-by-point reply describing the revisions that were already carried out and included in the transferred manuscript. If no revisions have been carried out yet, please leave this section empty.

As requested by Reviewer #1 we already:

- Incorporated all minor comments
- Described in more detail the functionality of the truncated proteins M1AP and SHOC1

Detailed Response:

Reviewer #1 comment:

I have a few minor comments for improving the manuscript:

No statistical analyses were performed. The meaning of error bars was not mentioned. It is essential to specify the minimum number of seminiferous tubules counted for each patient.

Response: We added the statistical analysis. We described now in more clarity that all round tubules in a patient's testicular section were counted (l. 646-653).

Reviewer #1 comment:

Allele frequencies of variants are not provided.

Response: We added the allele frequencies from gnomAD v4.1.0 and gnomAD SVs v2.1 (CNVs) in Table 1.

Reviewer #1 comment:

The authors should clearly label the bands of SPO16 in the right panel of Figure 1B.

Response: We labelled the SPO16 band in Figure 1B more clearly.

Reviewer #1 comment:

Figure 4 should clearly label the representations of each color channel.

Response: Thanks for this suggestion. We labelled each color channel accordingly.

Reviewer #1 comment:

Appendix Figure S1B and S2B, what does "rat" mean in "rat Ins2 Ex3/4"?

Response: In the minigene assay, an artificial gene was constructed with exon 3 and 4 from the insulin 2 gene of the species rat (*Rattus norvegicus*). We described this in more detail in the Appendix methods section (l. 119) and in the Figure legend S1B and S2B.

Reviewer #1 comment:

Overall, this study significantly contributes to the understanding of some genetic causes of human infertility and offers a potential avenue to treat patients with M1AP variants/ mutants.

Since no knock-in animal model was applied to mimic the subtle phenotype variations observed in patients, the functionality of truncated proteins remains unexplored. For example, it is unclear

Revision Plan

why the germ cells in patient M3260 with the SHOC1 variant can progress to round spermatids (Fig. 2C), while those in Shoc1 KO mice (10.1093/molehr/gaac015) and other patients cannot. However, this is a minor concern.

Response: Thanks for this comment. *SHOC1* variant c.1939+2T>C present in M3260 is a predicted splice site variant. *In vitro* it comes to an in-frame exon skipping as shown by the minigene assay (Appendix Figure S2) that is predicted to result in a loss of only 4% of the protein. We assume that this does not result in a complete loss but only in an impaired protein function enabling significantly reduced progression of spermatogenesis up to the round spermatid stage in few cells (l. 354-360). We addressed this in more detail in the results section (l. 145ff and l. 189ff) and in the Appendix Figure S2 legend. Accordingly, *SHOC1* variant c.1939+2T>C is not a LoF variant and we excluded it from the quantification of subsequent analyses.

In addition, the recurrent *M1AP* c.676dup was functionally analysed in our previous work (Wyrwoll et al., 2020, PMID: 32673564). We detected *M1AP* mRNA in a testicular biopsy from one patient showing that this variant leads not to degradation of the mRNA. Furthermore heterologous expression of the mutant *M1AP* cDNA in HEK293T cells led to the production of a truncated protein that presumably leads to loss of protein function. We added this information in l. 136. Furthermore, our preliminary experiments of co-immunoprecipitation of truncated M1AP with TEX11 hint to an abolished protein-protein interaction caused by *M1AP* c.676dup and thus a loss of protein function.

As requested by Reviewer #2 we already:

- Termed meiotic sub-stages such as "zygotene-like"
- Addressed the major comment regarding the evidence for the birth of one healthy child.
- Termed M1AP as a "functional enhancer" within the meiotic recombination pathway.
- Addressed all minor comments.

Detailed Response:

Reviewer #2 comment

That said, and with particular reference to Fig. 3A, it is difficult to sub-stage meiotic prophase by immunocytochemistry, even in optimal samples, with only one marker (in this case gH2AX). The staging here is also at odds with the statement in the subsequent section (and Fig. 4B) on absence of pachytene cells in men with mutation of SHOC1, TEX11, or SPO16. Because precise stages of arrest probably cannot be determined in these samples, the authors would be wiser to use phrases such as "zygotene-like"

Response: We agree with the reviewer that it is indeed difficult to sub-stage meiotic prophase based on IHC for one marker (see our Response to *Guidelines of human sub-staging* in chapter 4. *Description of analyses that authors prefer not to carry out*). Thus, we followed the reviewer's suggestion and have modified the respective phrases throughout the text, e.g. to 'zygotene-like'.

Revision Plan

Reviewer #2 comment:

Evidence for the birth of a (healthy) child from one individual with M1AP mutation verges on the anecdotal (N=1). It is interesting but raises multiple questions and concerns about both the frequency of chromosomal abnormalities in such individuals and the transmission of the mutant alleles.

Response: We understand very well, that the evidence based on N=1 seems to be sparse. Nevertheless, if it is in principle possible for a man affected by bi-allelic *M1AP* LoF variants to conceive a child by ICSI then it could be also possible for other couples with a similar genetic condition (*M1AP* LoF), and thus providing a proof-of-principle (l. 417f). Reviewer 2 is completely right with the concerns regarding chromosomal aberrations and the transmission of the mutant allele. Thus, it is essential for clinicians/geneticists to counsel the affected couple carefully about the small but existed chance to have a biological own child and the accompanied potential but so far unexplored risks as outlined in l. 435ff.

Reviewer #2 comment:

The authors conclude that the M1AP protein is an essential "catalyzer" in the meiotic recombination pathway. However, it is not clear from the data presented that M1AP in fact has enzymatic catalysis activity or exactly when and how it participates. Because the word "catalyzer" is not buttressed with hard or convincing evidence, the authors should consider other ways to describe the proposed role of M1AP, perhaps as a "putative component" and/or "modifier" of the recombination pathway.

Response: We thank the reviewer for this advice and changed the wording to "functional enhancer".

Reviewer #2 comment:

Minor comments:

Fig. 1A - these are nice illustrations, but overly simplified with respect to timing (synapsis is not completed in zygonema)

Response: We completely agree that Figure 1A is a simplified depiction that could not reflect the temporally and spatially highly complex processes of meiosis. By adding a second dotted box and describing the process in the Figure legend in more detail, we tried to reduce the simplification. Nonetheless, we believe that this simplified schematic help readers, who are less familiar with the progression of meiosis to contextualise the described processes

Reviewer #2 comment:

Fig. 1B - greater clarity in legend would be appreciated.

Response: We described Figure 1B in more detail.

Reviewer #2 comment:

Figs. 2A & 3A - colors in bar graphs are difficult to discriminate.

Response: We improved the discrimination of bar graphs accordingly.

Revision Plan

Reviewer #2 comment:

Fig. 4A - with full appreciation for the difficulty with this material, the images are of low contrast and require considerable enlargement

Response: We agree with this opinion; and we increased the contrast. In addition, we will improve the way of representation in a revised Figure 4 in the complete revision of the manuscript in accordance with the suggestions of all three Reviewers.

As requested by reviewer #3 we:

- Already tested the commercially available and published M1AP antibody (Wyrwoll et al., 2020) in addition to several further attempts, which, unfortunately, were unsuccessful.
- Addressed all minor comments (despite “FigS13: TEX11 staining should be presented with foci counting as a main figure” which is addressed under 3. Description of the planned revisions)

Detailed response:

Reviewer #3 comment:

The published M1AP antibody should be tested to investigate its perturbation in the absence of the ZZS proteins and the hierarchy of event.

Response: In the last couple of years, we spent enormous resources (personnel, time, financial) to get a functional antibody against human M1AP, including testing of different commercial (and already published) antibodies, creating three customised antibodies against different M1AP polypeptides, a nanobody raised against the complete M1AP protein (failed because of the impossibility to purify the protein), and contacting the authors of previously published customised M1AP antibodies (Arango et al., 2013/PMID 23269666 and Li et al. 2023/PMID 36440627). Figure x recapitulates some of our attempts. Moreover, we published the initial attempts of establishing an M1AP antibody in Wyrwoll et al., 2020/PMID 32673564. Unfortunately, no human M1AP-specific antibody is available.

Additionally, we tested different TEX11, SHOC1 and SPO16 antibodies in immunohistochemistry and SHOC1 and SPO16 antibodies in immunofluorescence of spermatocyte spreads, which did not result in a specific staining (Figure R3). Due to the lack of a human specific antibody against M1AP as well as antibodies against SHOC1 and SPO16, we are not able to localise these proteins in patient testicular sections to address this highly interesting research question that remains of great interest within our work on M1AP.

In conclusion, the suggestion of testing the M1AP (+ SHOC1/TEX11/SPO16) antibody was unsuccessful. To investigate the research question of the hierarchy of interactions, we plan to perform co-immunoprecipitation for truncated M1AP constructs with TEX11 (see Chapter 1. *Description of the planned revisions*).

Figure R2. Attempts to locate M1AP in the human testis. Previous attempts to identify a commercially available antibody that reliably detects M1AP in the human testis have not been successful (Wyrwoll et al., 2020/ PMID 32673564). Accordingly, we tried to produce a human-specific antibody in cooperation with companies specified in antibody customisation (Eurogentec, Biotem). The last attempt, conducted with Biotem, is exemplarily shown in this figure. A. Human M1AP protein sequence (NP_620159.2) highlighting the antibody epitopes (orange) that were selected so that in men carrying the *M1AP* LoF variant c.676dup p.Trp226Leufs*4 in a homozygous state, the respective antibody should not be able to bind due to the protein truncation. For rabbit immunisation, both epitopes were pooled. B. HEK293T cells were transfected with DYK-tagged M1AP plasmids, either expressing the wildtype (WT) or the truncated protein (W226L). Sera of day (D) 28 and 42 of the immunised rabbit as well as the purified antibody product, a commercially available anti-M1AP antibody (HPA), and anti-DYK control antibody specificity was confirmed by Western blotting. C. Customised anti-M1AP antibody validation in human testicular control and D. M1AP-deficient tissue did not yield in a reliable staining. Various protocol optimisations were tested (different antigen retrieval, adapted blocking and antibody dilution solution, various primary and secondary antibody concentrations). Data shown represents the best result, respectively. The application of both sera and the purified antibody for spermatocyte spreading was tested in parallel and has not been successful either (data not shown). SC: Sertoli cells, SPC: spermatocytes, M-I: metaphase I cells, RS: round spermatids, ES: elongated spermatids. The scale bar represents 100 μ m and 10 μ m.

Revision Plan

Figure R3. Efforts to identify human-specific antibodies for ZZS localisation. A. Commercially available antibodies for ZZS were tested via Western blotting, aiming to reliably detect SHOC1, SPO16, and TEX11 in human testicular biopsies. HA-tagged wildtype plasmid DNA (WT) was transfected in HEK293T cells and the anti-HA antibody was used as a positive control. Only one antibody detected TEX11 reliably in the purified lysates (anti-TEX11: HPA002950). B.-D. Immunohistochemical staining was performed with all antibodies on human testicular and is representatively shown for anti-SHOC1: #BS15344-R, anti-SPO16: #BS15024-R, and anti-TEX11: HPA002950. Only the anti-TEX11

Revision Plan

(#HPA002950) was found to be specific. However, presumably due to the fixation with Bouin's solution, staining could not reliably be repeated in all samples and was not implied in this study. Various protocol optimisations were tested (different antigen retrieval, adapted blocking and antibody dilution solution, various primary and secondary antibody concentrations). Data shown represents the best result, respectively. The application of all antibodies for spermatocyte spreading was tested in parallel and have not been successful (data not shown), except for anti-TEX11 (#HPA002950, Appendix Figure S13). SC: Sertoli cells, SPC: spermatocytes, RS: round spermatids, ES: elongated spermatids. The scale bar represents 100 μ m and 10 μ m.

Reviewer #3 comment:

Minor comments

As written, the title is misleading, the paper does not investigate the impact of M1AP in ZSS recombination. Such study implies to study genetic interactions or the genetic dependency between the different proteins, which is not the case here.

Response: We agree with this comment and changed the title to "Genotype-specific differences in infertile men due to loss-of-function variants in M1AP or ZZS genes"

Reviewer #3 comment:

Labelled on histological images is not clear. The authors should clearly explain to what marker each staining correspond.

Response: We changed the labelling accordingly.

Reviewer #3 comment:

L67 to 72: the authors should update and use more accurate citations for meiotic recombination.

Response: Thanks for this suggestion. In this section, we have described the fundamental processes of meiosis, which have been repeatedly reviewed by renowned scientists. We have therefore chosen four well-cited expert reviews from different groups as references (PMID: 29385397, 24050176, 27648641, 35613017).

Reviewer #3 comment:

L76: the ZMM are specifically involved in the resolution of class I crossover. Please rephrase.

Response: We rephrased the sentence and changed it throughout the manuscript.

Reviewer #3 comment:

L94: Strictly, the author identified an interaction, they didn't establish how the interaction takes place.

Response: We rephrased the sentence.

Reviewer #3 comment:

L255: MLH1 does not quantify homologous recombination but, class I crossovers.

Response: We rephrased the sentence.

Reviewer #3 comment:

L352: The sentence is hard to understand, rephrase please.

Response: We rephrased the sentence.

4. Description of analyses that authors prefer not to carry out

Please include a point-by-point response explaining why some of the requested data or additional analyses might not be necessary or cannot be provided within the scope of a revision. This can be due to time or resource limitations or in case of disagreement about the necessity of such additional data given the scope of the study. Please leave empty if not applicable.

Unfortunately, we cannot perform any experiments, which need the use of human specific M1AP, SHOC1, or SPO16 antibodies. In Chapter 3. *Description of the revisions that have already been incorporated in the transferred manuscript* we outlined in detail our attempts to establish antibodies against human M1AP, SHOC1, and SPO16.

Reviewer #2 requested that we (underlined is what was addressed in section “2. Description of the planned revisions”):

- Give few more guidelines to criteria for staging human tubules

Detailed response:

Reviewer #2 comment:

The authors should also clarify how it was confirmed that the metaphase-like cells were spermatocytes and not spermatogonia (given that gH2AX signal is weak or unclear in some such nuclei). Readers with a focus on the more regularly staged mouse or rat tubules would appreciate a few more guidelines to criteria for staging human tubules.

A precise sub-staging of the meiotic prophase would require identifying the stage of the seminiferous epithelium. The cycle of the human seminiferous epithelium has been subdivided into 12 stages based on the acrosomal development made visible by immunohistochemistry for acrosin. However, in order to properly evaluate the human germ cell associations, only seminiferous tubules showing a well-preserved seminiferous epithelium with no apparent damage to the epithelium and the peritubular wall can be considered. In addition, all the different generations of germ cells have to be present as well as at least six spermatids (Muciaccia et al., 2013, PMID: 23946533). As these requirements cannot be fulfilled in the testicular tissue of men with a meiotic arrest due to LoF variants in *M1AP* or the *ZZS* genes, we did not perform precise staging for human tubules. Accordingly, we followed the reviewer’s suggestion and have modified the respective phrases throughout the text, e.g. to ‘zygotene-like’.

Revision Plan

Reviewer #3 requested that we (underlined is what was addressed in section "2. Description of the planned revisions):

- Show different stages of spreaded spermatocytes.
- Optionally comment or calculate crossover interference.

Detailed response:

Reviewer #3 comment:

In the same line, TEX11 staining in M1AP mutant should be more documented and in particular, the different stages shown, as well as the foci counting, to have a quantitative result, that can be compared to MLH1.

Response: Due to the limited resources of cryopreserved material, we cannot repeat the TEX11 staining in the patients with *M1AP* LoF variant to document different stages. Slides that have already been stained are unfortunately bleached and cannot be re-analysed.

Reviewer #3 comment:

OPTIONAL: the obligatory crossover was measured, a comment or calculation of interference would be very interesting, and it seems doable using the MLH1 counting, to test whether thses mutants have an effect on this process.

Response: We thank Reviewer #3 for the suggestion of this interesting question that was not within our focus so far. Due to the limited material and the small number of cells from which we digitally separated the chromosomes, we believe that the sample size is insufficient to obtain a statistically significant result.

15th Jul 2024

Dear Dr. Friedrich,

Thank you for the submission of your research manuscript to our editorial offices. I have now had the opportunity to read the manuscript, the referee reports and your point-by-point response and to discuss it with the other members of our editorial team. We all agreed that the manuscript fits the scope of EMBO Molecular Medicine but also that the referees raised important concerns that should be addressed in major revision. In our opinion your revision plan addresses the referees' criticism adequately and therefore we would like to invite submission of the revised manuscript to our journal.

We would welcome the submission of a revised version within three months for further consideration. Please let us know if you require longer to complete the revision.

I look forward to receiving your revised manuscript.

Yours sincerely,

Zeljko Durdevic

We require:

- 1) A .docx formatted version of the manuscript text (including legends for main figures, EV figures and tables). Please make sure that the changes are highlighted to be clearly visible.
- 2) Individual production quality figure files as .eps, .tif, .jpg (one file per figure). For guidance, download the 'Figure Guide PDF': (<https://www.embopress.org/page/journal/17574684/authorguide#figureformat>).
- 3) A .docx formatted letter INCLUDING the reviewers' reports and your detailed point-by-point responses to their comments. As part of the EMBO Press transparent editorial process, the point-by-point response is part of the Review Process File (RPF), which will be published alongside your paper.
- 4) A complete author checklist, which you can download from our author guidelines (<https://www.embopress.org/page/journal/17574684/authorguide#submissionofrevisions>). Please insert information in the checklist that is also reflected in the manuscript. The completed author checklist will also be part of the RPF.
- 5) Please note that all corresponding authors are required to supply an ORCID ID for their name upon submission of a revised manuscript.
- 6) It is mandatory to include a 'Data Availability' section after the Materials and Methods. Before submitting your revision, primary

datasets produced in this study need to be deposited in an appropriate public database, and the accession numbers and database listed under 'Data Availability'. Please remember to provide a reviewer password if the datasets are not yet public (see <https://www.embopress.org/page/journal/17574684/authorguide#dataavailability>).

13) Author contributions: the contribution of every author must be detailed in a separate section (before the acknowledgments).

14) A Conflict of Interest statement should be provided in the main text.

15) Every published paper now includes a 'Synopsis' to further enhance discoverability. Synopses are displayed on the journal

webpage and are freely accessible to all readers. They include a short stand first (maximum of 300 characters, including space) as well as 2-5 one-sentences bullet points that summarizes the paper. Please write the bullet points to summarize the key NEW findings. They should be designed to be complementary to the abstract - i.e. not repeat the same text. We encourage inclusion of key acronyms and quantitative information (maximum of 30 words / bullet point). Please use the passive voice. Please attach these in a separate file or send them by email, we will incorporate them accordingly.

16) Include a Reagents and Tools Table as part of the Methods section, which can be downloaded from our author guidelines (<https://www.embopress.org/page/journal/17574684/authorguide#structuredmethods>)

Rev_Com_number: RC-2024-02439

New_manu_number: EMM-2024-20278-T

Corr_author: Friedrich

Title: Genotype-specific differences in infertile men due to loss-of-function variants in M1AP or ZZS genes

Dear Editor and Reviewers,

we thank you for your constructive criticism and overall positive evaluation of our work. Based on the reviewers' comments we have addressed the vast majority of concerns.

We are convinced that our manuscript has significantly benefitted from the suggestions raised by all three reviewers.

In our point-by-point response that you can find below, we described in detail, which suggestions have been addressed, and which were unfortunately not realisable.

In the revised version of our manuscript all changes are highlighted in the track modus. In this point-by-point response the font colour of the original reviewers' comments is black while our response is written in blue.

Reviewer #1 (Evidence, reproducibility and clarity (Required)):

This manuscript compiles LoF variants of M1AP and ZZS proteins (i.e., SHOC1, TEX11 and SPO16) that almost certainly underlie infertility and reports the first case of an infertile man homozygous for a variant in SPO16. The authors validated interactions between human M1AP and ZZS that were found in mice. Analyzing testicular samples from infertile men revealed that those with deficiencies in SHOC1, TEX11 or SPO16 exhibited early meiotic arrest without haploid germ cells, whereas those with M1AP variants displayed a predominant metaphase I arrest with rare haploid germ cells. Further investigations showed that disrupted SHOC1, TEX11 or SPO16 led to defective synapsis and pairing of homologous chromosomes and unpaired DNA DSBs, while M1AP mutations reduced CO events. Importantly, men with LoF variants in M1AP can father healthy children by medically assisted reproduction. Overall, the results are clear and convincing in defining likely causative variants in infertility patients.

Response: We thank reviewer #1 for the appreciation of our work. We addressed all suggestions raised by reviewer# 1 to improve our manuscript.

I have a few minor comments for improving the manuscript:

- No statistical analyses were performed. The meaning of error bars was not mentioned. It is essential to specify the minimum number of seminiferous tubules counted for each patient.

Response: We added the statistical analysis in the Methods and Protocol section (l. 813ff) and in respective figure legends. We described now in more clarity that all round tubules in a patient's testicular section were counted (l. 801ff).

- Allele frequencies of variants are not provided.

Response: We added the allele frequencies from gnomAD v4.1.0 (SNVs) and gnomAD SVs v2.1 (CNVs) in Table 1.

- Figure 4 should clearly label the representations of each color channel.

Response: Thanks for this suggestion. We labelled each color channel accordingly.

- The authors should clearly label the bands of SPO16 in the right panel of Figure 1B.

Response: We labelled the SPO16 band in Figure 1B more clearly.

- Appendix Figure S1B and S2B, what does "rat" mean in "rat Ins2 Ex3/4"?

Response: In the minigene assay, an artificial gene was constructed with exon 3 and 4 from the insulin 2 gene of the species rat (*Rattus norvegicus*). We described this in more detail in the Methods and Protocols section "Minigene splicing assay" (l. 679ff) and in the Figure legend EV2 and EV3.

Reviewer #1 (Significance (Required)):

Overall, this study significantly contributes to the understanding of some genetic causes of human infertility and offers a potential avenue to treat patients with M1AP variants/ mutants. Since no knock-in animal model was applied to mimic the subtle phenotype variations observed in patients, the functionality of truncated proteins remains unexplored. For example, it is unclear why the germ cells in patient M3260 with the SHOC1 variant can progress to round spermatids (Fig. 2C), while those in Shoc1 KO mice (10.1093/molehr/gaac015) and other patients cannot. However, this is a minor concern.

Response: Thanks for this comment. *SHOC1* variant c.1939+2T>C present in M3260 is a predicted splice site variant. *In vitro* it results in an in-frame exon skipping as shown by the minigene assay (Fig EV3) that is predicted to lead to a loss of only 4% of the protein. We assume that this does not result in a complete loss but only in an impaired protein function enabling significantly reduced progression of spermatogenesis up to the round spermatid stage in few cells (l. 354-360). We addressed this in more detail in the Results section (l. 153ff and l. 195ff), in the Fig EV3 legend, and the Discussion (l. 391ff).

Accordingly, *SHOC1* variant c.1939+2T>C is not a LoF variant and we excluded it from the quantification of subsequent analyses. Immunohistological staining of this patients was excluded from Appendix Figure S5, S6, S8, S9, and S11 and incorporated into Fig EV3.

In addition, the recurrent M1AP c.676dup was functionally analysed in our previous work (Wyrwoll et al., 2020, PMID: 32673564). We detected M1AP mRNA in a testicular biopsy from one patient showing that this variant leads not to degradation of the mRNA. Furthermore heterologous expression of the mutant M1AP cDNA in HEK293T cells led to the production of a truncated protein that presumably leads to loss of protein function. We added this information in l. 142. Furthermore, we performed new experiments of co-immunoprecipitation of truncated M1AP with wildtype TEX11 which showed an abolished protein-protein interaction caused by M1AP c.676dup and thus a loss of protein function.

Our field of expertise is gametogenesis and meiosis in mice.

Reviewer #2 (Evidence, reproducibility and clarity (Required)):

Summary:

This interesting manuscript provides evidence for the biological and clinical relevance in human males of mutations in genes encoding M1AP and other related proteins. In mice, M1AP, "meiosis 1 associated protein," is known to associate with several proteins (SHOC1, TEX11, and SPO16) in the ZZS complex that promotes DNA recombination and crossover formation during meiosis I prophase. Mutation of these proteins in model organisms disrupts the process of recombination and cause arrest of spermatocytes prior to the first meiotic division. Here the authors took advantage of their MERGE (Male Reproductive Genomics) cohort to screen for human loss-of-function (LoF) mutations in the relevant ZZS complex and M1AP genes and to associate these with human male reproductive phenotypes. They found that men with deficiency of ZZS proteins SHOC1, TEX11 or SPO16 genes were infertile, exhibiting arrest of germ cell development early in meiotic prophase, with aberrations of chromosome synapsis and failure to repair DNA double-strand breaks (DSBs). In interesting contrast, men with M1AP mutations exhibited metaphase arrest, and indeed, in some cases, produced haploid spermatids, which in medically assisted reproduction (ICSI), led to the birth of offspring. Because they demonstrate that M1AP interacts with the other proteins, the authors conclude that M1AP is a "catalyzer," but not essential, for the processes of synapsis, recombination, and formation of haploid gametes.

Major Comments:

The work is clearly presented with detailed methods that should allow adaptation in other laboratories.

Overall, this study is a tour de force with what was no doubt difficult archival samples. The histology is generally of good quality, supporting the conclusions about progress of meiotic prophase in the mutant samples. The images of H&E-stained tissue are particularly striking, especially those in supplemental figures.

Response: We thank Reviewer #2 for the appreciation of our work and the suggestions to improve our manuscript. To provide transparency of our work, we uploaded each (immuno-) histologically stained testicular section shown in the Main and Appendix Figures in the microscopy image repository OMERO/Open Microscopy Environment (<https://omero-min.uni-muenster.de/webclient/?show=project-6067>), which is accessible via "Log in as public user".

That said, and with particular reference to Fig. 3A, it is difficult to sub-stage meiotic prophase by immunocytochemistry, even in optimal samples, with only one marker (in this case gH2AX). The staging here is also at odds with the statement in the subsequent section (and Fig. 4B) on absence of pachytene cells in men with mutation of SHOC1, TEX11, or SPO16.

Because precise stages of arrest probably cannot be determined in these samples, the authors would be wiser to use phrases such as "zygotene-like"

Response: We agree with the reviewer that it is indeed difficult to sub-stage meiotic prophase based on IHC for one marker. A precise sub-staging of the meiotic prophase would require identifying the stage of the seminiferous epithelium. The cycle of the human seminiferous epithelium has been subdivided into 12 stages based on the acrosomal development made visible by immunohistochemistry for acrosin. However, in order to properly evaluate the human germ cell associations, only seminiferous tubules showing a well-preserved seminiferous epithelium with no apparent damage to the epithelium and the peritubular wall can be considered. In addition, all the different generations of germ cells have to be present as well as at least six spermatids (Muciaccia et al., 2013, PMID: 23946533). As these requirements cannot be fulfilled in the testicular tissue of men with a meiotic arrest as due to LoF variants in *M1AP* or the *ZZS* genes, we followed the reviewer's

suggestion and have modified the respective phrases throughout the text, e.g. to 'zygotene-like'.

The authors should also clarify how it was confirmed that the metaphase-like cells were spermatocytes and not spermatogonia (given that γ H2AX signal is weak or unclear in some such nuclei). Readers with a focus on the more regularly staged mouse or rat tubules would appreciate a few more guidelines to criteria for staging human tubules.

Response: We thank the reviewer for raising this point. In order to confirm that the metaphase-like cells were indeed spermatocytes we performed additional IHC staining for γ H2AX, H3S10 and MAGEA4 on sequential testis sections (distance 3 μ m) on representative samples of the M1AP patient cohort as well as controls. For a few more guidelines on the criteria for staging human tubules, please refer to the response to the previous point.

Evidence for the birth of a (healthy) child from one individual with M1AP mutation verges on the anecdotal (N=1). It is interesting but raises multiple questions and concerns about both the frequency of chromosomal abnormalities in such individuals and the transmission of the mutant alleles.

Response: We understand very well, that the evidence based on N=1 seems to be sparse. Nevertheless, if it is in principle possible for a man affected by bi-allelic *M1AP* LoF variants to conceive a child by ICSI then it could be also possible for other couples with a similar genetic condition (*M1AP* LoF), and thus providing a proof-of-principle (l. 456f). Reviewer #2 is completely right with the concerns regarding chromosomal aberrations and the transmission of the mutant allele. Thus, it is essential for clinicians/geneticists to counsel the affected couple carefully about the small but existed chance to have a biological own child and the accompanied potential but so far unexplored risks as outlined in l. 475ff. Our future research project will address this open and highly relevant question.

The authors conclude that the M1AP protein is an essential "catalyzer" in the meiotic recombination pathway. However, it is not clear from the data presented that M1AP in fact has enzymatic catalysis activity or exactly when and how it participates. Because the word "catalyzer" is not buttressed with hard or convincing evidence, the authors should consider other ways to describe the proposed role of M1AP, perhaps as a "putative component" and/or "modifier" of the recombination pathway.

Response: We appreciate the reviewer's advice, and changed the wording to "functional enhancer" because we wanted to emphasise that the function fulfilled by the ZZS complex is increased/enhanced by M1AP.

Minor comments:

Fig. 1A - these are nice illustrations, but overly simplified with respect to timing (synapsis is not completed in zygonema)

Response: We completely agree that Figure 1A is a simplified depiction that could not reflect the temporally and spatially highly complex processes of meiosis. By adding a second dotted box and describing the process in the Figure legend in more detail, we tried to reduce the simplification. Nonetheless, we believe that this simplified schematic help readers, who are less familiar with the progression of meiosis to contextualise the described processes.

Fig. 1B - greater clarity in legend would be appreciated

Response: We described Figure 1B in more detail.

Figs. 2A & 3A - colors in bar graphs are difficult to discriminate

Response: We improved the discrimination of bar graphs accordingly.

Fig. 4A - with full appreciation for the difficulty with this material, the images are of low contrast and require considerable enlargement

Response: We agree with this opinion; we increased the contrast and improved the magnification. In addition, we uploaded the microscopic images in the microscopy image repository OMERO/Open Microscopy Environment (<https://omero-min.uni-muenster.de/webclient/?show=project-6067>), which is accessible via "Log in as public user" and which facilitates to zoom into the details. The original Figure 4 has become Figure 5 in the revised version of the manuscript.

Reviewer #2 (Significance (Required)):

This is a very interesting paper, which I evaluated from the perspective of a reproductive geneticist with expertise in meiosis and interest in infertility. I think this report will be of interest to clinicians because it identifies a gene possibly linked to marginal fertility and establishes human protein interactions similar to those previously identified in mice. It reinforces the importance of ZZS genes in humans. The contributions of this report to the field of meiosis confirm previous evidence on M1AP, including mutant phenotypes and protein interactions, extending them to humans. We can thus appreciate the conserved function of the mammalian M1AP protein, but as yet the molecular mechanisms of M1AP are not clarified.

Response: We gratefully thank Reviewer #2 for the thorough evaluation of our work and appreciate the recognition of the significance. Indeed, it was not possible to clarify the molecular mechanisms of M1AP but by performing further meiotic spread staining of different meiosis specific proteins and under consideration of published mouse models we substantiate the evidence that M1AP is functionally linked to the ZZS interplay and thereby affecting the processes of class I crossover (l. 428ff). Unfortunately, there is no M1AP specific antibody available that is functional in the required applications; please see our attempts to establish M1AP and ZZS specific antibodies in our response to Reviewer #3. Clarifying the underlying molecular mechanism is not only one of our highest interest but will also be important for the scientific community.

Reviewer #3 (Evidence, reproducibility and clarity (Required)):

In this manuscript, Rotte et al. investigate the meiotic molecular function in human of the M1AP protein and of the ZSS complex (SCHOC1, TEX11 and SPO16 proteins). The ZSS complex is a key player of meiotic recombination. It is a sub-complex of the conserved family of the ZMM proteins, essential for the formation of class I crossovers, a proper chromosomes segregation and fertility. Understanding its mode of action, regulation and conservation in human is thus a crucial issue in the fields of meiosis and human reproduction, with potential implications for patients. In that context, the recent identification of the protein M1AP as a partner of the ZSS proteins raise the question of its role, function and conservation. The aim of this study is thus of primary importance.

To perform this molecular characterization, the authors made a cohort (24 total) of men carrying LoF variants in M1AP and ZSS genes. They performed a molecular biology analysis to assess the physical interaction between the human M1AP protein and the three components of the ZSS complex. Their results confirm a previous work performed in mice, mentioned by the authors.

Then, they took advantage of available biopsies from different mutant men to perform a histological and cytological analysis of the impact of the different mutations on meiosis. The main conclusions are that in human, similarly to what is known in different organisms (ranging from yeast to mice), the ZSS complex is essential for crossover formation, synapsis and spermatogenesis, and that defect in the genes is associated with a premature prophase I arrest and no sperm formation. The authors also showed that M1AP protein plays a role in meiotic progression, but to a lesser extent compare to the ZSS proteins, with a metaphase I arrest, an undetectable recombination phenotype, apart of a reduced crossover number and, spermatozoa can form in its absence.

Major points:

The authors investigate the physical interaction between M1AP and the ZSS members through a single approach: Co-IP of tagged proteins after expression in human HEK293T cells. This approach is informative, but to reinforce the conclusions the authors should provide data from independent approaches: yeast two hybrid, expression of recombinant proteins followed by pull down, co-immunostaining (TEX11 antibodies were used in the study and M1AP antibody is present in the literature) are possible non-exclusive approaches to decipher, more in details, the interaction. Moreover, understanding the hierarchy of interactions appears important to understand its rational, regulation and function. What is the meaning of a M1AP interaction with all the members of the complex? Remains an open question.

Response: We thank Reviewer #3 for this comment. In an independent approach we specified the interaction of M1AP to the ZSS proteins. Thus, we cloned truncated versions of *M1AP* to refine the binding site of M1AP to the ZSS proteins (Figure R1). We co-transfected full-length as well as the truncated forms of *M1AP* with *TEX11* and showed via Co-IP that the interaction is only possible with the full-length M1AP and that the deletion of 300 bp is sufficient to abolish the binding between M1AP and TEX11 (l. 118ff, 355ff).

Furthermore, based on the results of the new meiotic spread staining with RAD51 and MSH5 antibodies and published data on mouse models, we substantiate the evidence that M1AP is functionally linked to the ZSS interplay and thereby affecting the processes of class I crossover (l. 428ff).

Figure R1 Tolerance landscape of *M1AP* NM_001321739.2 illustrating the respective regions selected for mutagenesis of truncated *M1AP* constructs. Adapted from MetaDome.

Moreover, in the last couple of years, we spent enormous resources (personnel, time, financial) to get a functional antibody against human M1AP, including testing of different commercial (and already published) antibodies, creating three customised antibodies against different M1AP polypeptides, a nanobody raised against the complete M1AP protein (failed because of the impossibility to purify M1AP protein), and contacting the authors of previously published customised M1AP antibodies (Arango et al., 2013/PMID 23269666 and Li et al. 2023/PMID 36440627). Figure R2 recapitulates some of our attempts. Moreover, we published the initial attempts of establishing an M1AP antibody in Wyrwoll et al., 2020/PMID 32673564. Unfortunately, no human M1AP-specific antibody was available.

Additionally, we tested different TEX11, SHOC1 and SPO16 antibodies in immunohistochemistry and SHOC1 and SPO16 antibodies in immunofluorescence of spermatocyte spreads, which did not result in a specific staining (Figure R3). Due to the lack of a human specific antibody against M1AP as well as antibodies against SHOC1 and SPO16, we are not able to localise these proteins in patient testicular sections to address this highly interesting research question that remains of great interest within our work on M1AP.

Figure R2. Attempts to locate M1AP in the human testis. Previous attempts to identify a commercially available antibody that reliably detects M1AP in the human testis have not been successful (Wyrwoll et al., 2020/ PMID 32673564). Accordingly, we tried to produce a human-specific antibody in cooperation with companies specified in antibody customisation (Eurogentec, Biotem). The last attempt, conducted with Biotem, is exemplarily shown in this figure. A. Human M1AP protein sequence (NP_620159.2) highlighting the antibody epitopes (orange) that were selected so that in men carrying the M1AP LoF variant c.676dup p.Trp226Leufs*4 in a homozygous state, the respective antibody should not be able to bind due to the protein truncation. For rabbit immunisation, both epitopes were pooled. B. HEK293T cells were transfected with DYK-tagged M1AP plasmids, either expressing the wildtype (WT) or the truncated protein (W226L). Sera of day (D) 28 and 42 of the immunised rabbit as well as the purified antibody product, a commercially available anti-M1AP antibody (HPA), and anti-DYK control antibody specificity was confirmed by Western blotting. C. Customised anti-M1AP antibody validation in human testicular control and D. M1AP-deficient tissue did not yield in a reliable staining. Various protocol optimisations were tested (different antigen retrieval, adapted blocking and antibody dilution solution, various primary and secondary antibody concentrations). Date shown represents the best result, respectively. The application of both sera and the purified antibody for spermatocyte spreading was tested in parallel and has not been successful either (data not shown). SC: Sertoli cells, SPC: spermatocytes, M-I: metaphase I cells, RS: round spermatids, ES: elongated spermatids. The scale bar represents 100 μ m and 10 μ m.

Figure R3. Efforts to identify human-specific antibodies for ZZS localisation. A. Commercially available antibodies for ZZS were tested via Western blotting, aiming to reliably detect SHOC1, SPO16, and TEX11 in human testicular biopsies. HA-tagged wildtype plasmid DNA (WT) was transfected in HEK293T cells and the anti-HA antibody was used as a positive control. Only one antibody detected TEX11 reliably in the purified lysates (anti-TEX11: HPA002950). B.-D. Immunohistochemical staining was performed with all antibodies on human testicular and is representatively shown for anti-SHOC1: #BS155344-R, anti-SPO16: #BS15024-R, and anti-TEX11: HPA002950. Only the anti-TEX11 (#HPA002950) was found to be specific. However, presumably due to the fixation with Bouin's solution, staining could not reliably be repeated in all samples and was not implied in this study. Various protocol optimisations were tested (different antigen retrieval, adapted blocking and antibody dilution solution, various primary and secondary antibody concentrations). Date shown represents the best result, respectively. The application of all antibodies for spermatocyte spreading was tested in parallel and have not been successful (data not shown), except for anti-TEX11 (#HPA002950, Appendix Figure S13). SC: Sertoli cells, SPC: spermatocytes, RS: round spermatids, ES: elongated spermatids. The scale bar represents 100 μ m and 10 μ m.

The ZZS mutants have a defect in γ H2AX pattern, a defect in synapsis and no MLH1 foci, associated to apoptosis and prophase I arrest. M1AP mutation has a minor impact. The characterization of the effect of the different mutations (in particular M1AP) on the recombination process should be addressed further, by cytological means. For example, effect on strand invasion and ssDNA production should be monitored using RPA, DMC1 and RAD51 antibodies. The impact on alternative resolution pathway (e.g. BLOOM dependent) should be tested as well as the effect on other ZMM proteins, in particular MSH4-5, should be investigated. These experiments are essential to characterize, at the molecular level, the function of the different proteins during recombination.

Response: We thank the reviewer for this suggestion and highly appreciated to investigate the different pathways in more depth. We performed additional immunofluorescence staining of DMC1, RPA2, RAD51, MSH4 and MSH5 of spermatocyte spreads of identified patients compared to the control. Among these, we were able to establish the RAD51 (II. 284ff) and MSH5 (298ff) staining to investigate the effect of the LoF variants in *M1AP* or the ZZS genes on strand invasion and on other ZMM proteins, respectively. We detected a persistent RAD51 expression in all cases examined, and a reduced (M1AP) or abolished (ZZS) MSH5 expression, which resembled published mouse models (I. 374f, I. 416ff).

We would like to take the opportunity to refer to the extremely limited access to cryopreserved testicular material of the patients presented in this manuscript: for each gene (*M1AP*, *SHOC1*, *TEX11*, *SPO16*) we were lucky to get one testicular biopsy specimen from one man for only one preparation of spermatocyte spreads. In addition, the slides of the spermatocyte spreads were a few months old and, furthermore, we didn't have access to the high-resolution Elyra microscope. Thus, the resolution of new microscopic images therefore differs from previous data. We hope for the Reviewer's understanding that we cannot address each requested staining albeit this would be of highest interest.

In the same line, TEX11 staining in M1AP mutant should be more documented and in particular the different stages shown, as well as the foci counting, to have a quantitative result, that can be compared to MLH1. Moreover, co-immunostaining of different markers with TEX11: RPA, DMC1, MSH and MLH1 are also important to understand how the pathway is perturbed and the recruitment delayed/affected.

We added the TEX11 foci count using the acquired images, which were similarly reduced, comparable to the MLH1 foci (I. 296f). Unfortunately, due to the limitation of cryopreserved testicular material it was not possible to perform co-staining of TEX11 with different meiotic markers. In the same line, we could not repeat the TEX11 staining in the patients with *M1AP* LoF variant for documentation of different stages. Moreover, slides that have already been stained are unfortunately bleached and cannot be re-analysed.

The published M1AP antibody should be tested to investigate its perturbation in the absence of the ZZS proteins and the hierarchy of event.

Response: As already outlined above, we tried to get any functional M1AP antibody for several years, which was not possible (Figure R2). Thus, we unfortunately cannot address this comment via this approach albeit this research question remains of great interest within our work on M1AP.

OPTIONAL: the obligatory crossover was measured, a comment or calculation of interference would be very interesting, and it seems doable using the MLH1 counting, to test whether these mutants have an effect on this process.

Response: We thank Reviewer #3 for the suggestion of this interesting question that was not within our focus so far. Due to the limited material and the small number of cells from which we could digitally separate the chromosomes, we believe that the sample size is insufficient to obtain a statistically significant result.

Minor comments

As written, the title is misleading, the paper does not investigate the impact of M1AP in ZSS recombination. Such study implies to study genetic interactions or the genetic dependency between the different proteins, which is not the case here.

Response: Thanks for this comment. We changed the title to “Genotype-specific differences in infertile men due to loss-of-function variants in *M1AP* or *ZZS* genes”.

Labelled on histological images is not clear. The authors should clearly explain to what marker each staining correspond.

Response: We changed the labelling accordingly.

L67 to 72: the authors should update and use more accurate citations for meiotic recombination.

Response: Thanks for this suggestion. In this section, we have described the fundamental processes of meiosis, which have been repeatedly reviewed by renowned scientists. We have therefore chosen four well-cited expert reviews from different groups as references (PMID: 29385397, 24050176, 27648641, 35613017).

L76: the ZMM are specifically involved in the resolution of class I crossover. Please rephrase.

Response: We rephrased the sentence and changed it throughout the manuscript.

L94: Strictly, the author identified an interaction, they didn't establish how the interaction takes place.

Response: We rephrased the sentence.

FigS13: TEX11 staining should be presented with foci counting as a main figure.

Response: We now present the TEX11 with foci counting as main Figure 4.

L255: MLH1 does not quantify homologous recombination but, class I crossovers.

Response: We rephrased the sentence.

L352: The sentence is hard to understand, rephrase please.

Response: We rephrased the sentence.

Reviewer #3 (Significance (Required)):

In general, the paper is well written and easy to follow. However, in light of the importance of the questions for the field of meiosis, it currently seems a little superficial, in particular if the authors aim at addressing the molecular function of the different proteins. The role of the ZSS proteins and M1AP in the control of meiotic recombination, at the molecular level is very important to decipher and additional experiments might help to better address this question. In addition, the functional links between M1AP and ZSS remains unclear and to investigate further.

This study gives information for human process, and can be compared to more advanced work done with mice.

This study will be important for the community working on meiosis in mammals, but also for people interested in reproduction.

Response: We thank Reviewer #3 for the thorough evaluation and acknowledgment of the significance of our work. We appreciated the suggestion to perform additional experiments, including TEX11 foci counting, and staining of MSH5 and RAD51 in human meiotic spreads, which was comparable to the more advanced work done with mice, to gain a better and deeper understanding of the molecular pathways involved. We hope for the Reviewer's understanding that we could not address all raised comments due to the limited material and the difficulty to get human specific antibodies in a research field that primarily works with highly valuable mouse models.

13th Mar 2025

Dear Dr. Friedrich,

As you will see from their reports pasted below, while referees #1 and #2 support publication of the manuscript, referee #3 although supportive raises important concerns. Based on the initial referee reports and after an editorial discussion, we agreed that referee #3 suggestions should be implemented in the next round of the revision. Therefore, I am pleased to inform you that we will be able to accept your manuscript pending the following final amendments:

- 1) Please implement all the referee #3 suggestions and address all his/her concerns. No additional experiments are required. Particular attention should be given to state the limitations of some experiments/conclusions and to tone down some conclusions where appropriate and as suggested by the referee #3. Additionally, please change the colors of the bars in Figures 2 and 3 to primary colors as suggested by the referee #1.
- 2) Authors: E-mail correspondence to Ann-Kristin Dicke could not be delivered. Please update author's e-mail address and make sure to enter correct e-mail addresses for all authors in our submission system.
- 3) Figures: We note that some panels are reused e.g. Fig. 2C and Fig. EV3I. Please cite in the respective figure legend every reused panel.
- 4) In the main manuscript file, please do the following:
 - Please address all comments suggested by our data editors listed below:
 - o Figure legends:
 1. Please define the annotated p values ****/**/**/* as well as provide the exact p-values for the same in the legend of figure 2B, 3B, 4B, 5B as appropriate.
 2. Please note that the scale bar is missing for figure 2C.
 3. Please note that the scale bar needs to be defined for figures EV3 D-H.
 4. Please note that scale bar and its definition are missing for figures EV3 I.
 - Remove all figures and only leave their legends at the end of the file.
 - Figure callouts should be in sequential order. Currently, Fig. 5 is called out before Fig. 4B. Please correct. Also, please check and update the callouts of figures and tables in the text, e.g. currently Appendix Table S3 is called out in Methods line 677 "Antibody details are listed in Appendix Table S3", however, current Appendix Table S3 does not contain antibody information.
 - Rename "Competing interest" to "Disclosure Statement & Competing Interests". We updated our journal's competing interests policy in January 2022 and request authors to consider both actual and perceived competing interests. Please review the policy <https://www.embopress.org/competing-interests> and update your competing interests if necessary.
 - Author contributions: Please remove it from the manuscript and specify author contributions in our submission system. CRediT has replaced the traditional author contributions section because it offers a systematic machine-readable author contributions format that allows for more effective research assessment. You are encouraged to use the free text boxes beneath each contributing author's name to add specific details on the author's contribution. More information is available in our guide to authors: <https://www.embopress.org/page/journal/17574684/authorguide#authorshipguidelines>
 - Please remove Reagents and Tools Table and uploaded it as a separate file. Structured Methods section includes Reagents and Tools Table followed by a Methods and Protocols section. More information on how to adhere to this format as well as downloadable templates (.docx) for the Reagents and Tools Table can be found in our author guidelines: <https://www.embopress.org/page/journal/17574684/authorguide#structuredmethods>
 - An example of a paper with Structured Methods can be found here: <https://www.embopress.org/doi/full/10.1038/s44320-024-00037-6#sec-4>
 - In Methods, provide the statement that informed consent was obtained from all human subjects and confirm that the experiments conformed to the principles set out in the WMA Declaration of Helsinki and the Department of Health and Human Services Belmont Report.
 - In Methods, provide the antibody dilutions that were used for each antibody.
 - Indicate in legends exact n and exact p values, not a range, along with the statistical test used. To keep the figures "clear" some authors found providing an Appendix table Sx with all exact p-values preferable. You are welcome to do this if you want to.
 - Correct the reference citation in the reference list. Citations should be listed in alphabetical order. Where there are more than 10 authors on a paper, 10 will be listed, followed by "et al.". Remove DOIs and PMID/PMCID numbers. Please check "Author Guidelines" for more information. <https://www.embopress.org/page/journal/17574684/authorguide#referencesformat>
- 5) Appendix: Please add page numbers to ToC, accept all changes and submit the file in PDF format.
- 6) The Paper Explained: Please add it to the main manuscript file.
- 7) Synopsis:
 - Synopsis image: Please format the image to 550 px-wide x (300 - 600)-px high and upload it as a high-resolution JPEG file.
 - Please check your synopsis text and image before submission with your revised manuscript. Please be aware that in the proof stage minor corrections only are allowed (e.g., typos).
- 8) Source data: Please upload source data one (zipped) file per figure.
- 9) As part of the EMBO Publications transparent editorial process initiative (see our Editorial at

<http://embomolmed.embopress.org/content/2/9/329>), EMBO Molecular Medicine will publish online a Review Process File (RPF) to accompany accepted manuscripts. This file will be published in conjunction with your paper and will include the anonymous referee reports, your point-by-point response and all pertinent correspondence relating to the manuscript. Let us know whether you agree with the publication of the RPF and as here, if you want to remove or not any figures from it prior to publication. Please note that the Authors checklist will be published at the end of the RPF.

10) Please provide a point-by-point letter INCLUDING my comments as well as the reviewer's reports and your detailed responses (as Word file).

I look forward to reading a new revised version of your manuscript as soon as possible.

Yours sincerely,

Zeljko Durdevic

*** Instructions to submit your revised manuscript ***

1) a .docx formatted version of the manuscript text (including Figure legends and tables)

2) Separate figure files*

3) supplemental information as Expanded View and/or Appendix. Please carefully check the authors guidelines for formatting Expanded view and Appendix figures and tables at <https://www.embopress.org/page/journal/17574684/authorguide#expandedview>

4) a letter INCLUDING the reviewer's reports and your detailed responses to their comments (as Word file).

5) The paper explained: EMBO Molecular Medicine articles are accompanied by a summary of the articles to emphasize the major findings in the paper and their medical implications for the non-specialist reader. Please provide a draft summary of your article highlighting

6) Author contributions: the contribution of every author must be detailed in a separate section.

7) EMBO Molecular Medicine now requires a complete author checklist (<https://www.embopress.org/page/journal/17574684/authorguide>) to be submitted with all revised manuscripts. Please use the

checklist as guideline for the sort of information we need WITHIN the manuscript. The checklist should only be filled with page numbers where the information can be found. This is particularly important for animal reporting, antibody dilutions (missing) and exact values and n that should be indicated instead of a range.

8) Every published paper now includes a 'Synopsis' to further enhance discoverability. Synopses are displayed on the journal webpage and are freely accessible to all readers. They include a short stand first (maximum of 300 characters, including space) as well as 2-5 one sentence bullet points that summarise the paper. Please write the bullet points to summarise the key NEW findings. They should be designed to be complementary to the abstract - i.e. not repeat the same text. We encourage inclusion of key acronyms and quantitative information (maximum of 30 words / bullet point). Please use the passive voice. Please attach these in a separate file or send them by email, we will incorporate them accordingly.

You are also welcome to suggest a striking image or visual abstract to illustrate your article. If you do please provide a jpeg file 550 px-wide x 300-600px high.

9) A Conflict of Interest statement should be provided in the main text

10) Please note that we now mandate that all corresponding authors list an ORCID digital identifier. This takes <90 seconds to complete. We encourage all authors to supply an ORCID identifier, which will be linked to their name for unambiguous name identification.

Currently, our records indicate that the ORCID for your account is 0000-0001-8848-3366.

Link Not Available

11) Include a Reagents and Tools Table as part of the Methods section, which can be downloaded from our author guidelines (<https://www.embopress.org/page/journal/17574684/authorguide#structuredmethods>)

Photos 400-800 DPI

*Additional important information regarding figures and illustrations can be found at <https://bit.ly/EMBOPressFigurePreparationGuideline>. See also figure legend preparation guidelines: <https://www.embopress.org/page/journal/17574684/authorguide#figureformat>

***** Reviewer's comments *****

Referee #1 (Comments on Novelty/Model System for Author):

This manuscript is of good impact and clinical relevance.

Referee #1 (Remarks for Author):

The revised manuscript still has my considerable enthusiasm. The authors have done due diligence in revising the original submission and preparing the manuscript specifically for EMBO Molecular Medicine, and it is much improved. Improvement is especially apparent in that the authors have kept conclusions consistent with what the data actually show; an excellent example of this is the change in title, but similar appropriate adjustments have been made throughout the manuscript.

The authors have responded satisfactorily to my previous comments. I appreciated the figures in OMERO. (As a somewhat picky point, I still find the shades of green/gray in the bar graphs within Figs. 2 and 3 hard to discriminate and would prefer to see primary colors; however, I leave this to journal copy editors.)

I think this report will be of interest to clinicians and to basic scientists in the field of meiosis and genetics of fertility/infertility.

Referee #2 (Remarks for Author):

The authors addressed our issues well. I have no further suggestions.

Referee #3 (Comments on Novelty/Model System for Author):

The model is perfectly adequate to the question and the reviewer 3 fully understand the technical difficulty to work with mouse model.

Referee #3 (Remarks for Author):

The manuscript can be edited to be more easy to understand for non human genetics specialist, which work on meiosis, using alternative models.

Please find here more specific comments on the experiments:

Regarding the physical interaction:

The proposed experiments map the domain of M1AP required for the interaction with TEX11, but it is not an alternative/independent approach to verify the interaction. From a technical point of view, it is the same experiment.

Do the authors talk about point mutations or truncations? This has to be stated/explained clearly. Then, are the truncations up to 108, 226, 355 or 433 or do they start from these residues? It has to be clearly explained (drawing of the protein construct might be useful). Truncation should not be discussed in bp but in numbers of amino acids.

Importantly, with the proposed experiment, the site of the interaction is not addressed, the domains are. And, it gives negative results, which can always be due to a defective folding, for example (which was not tested experimentally). It would have been relevant to test the minimum domain that interact (in other words a vice versa experiment), to get a positive result from this domain mapping experiment.

Finally, only the interaction M1AP / TEX11 is addressed. Is there any reason to speculate that M1AP uses the same domains to interact with all the ZSS members? Why focusing on TEX11?

For all these reasons, with only one approach, the validation of the interaction remains weak to me, and the domain mapping has to be clearly explained and not over interpreted.

Why not including alphaFold modeling?

Regarding immunostaining of recombination protein:

I thank the authors for this information about antibody, and hopefully in the future and next studies these tools will appear.

As a comment, I would say that for reviewer shown staining is only on sections and, to assess recombination proteins function immune-cytochemistry is more informative on prophase I chromosomes spreads. But, I do understand the difficulty in getting such human samples.

Reviewer 3 thanks the authors for their effort in providing a more complete characterization of the putative defect in HR, in the different mutant conditions.

The revised manuscript gave new additional comments on recombination starting by gH2AX staining. M1AP mutant pattern is clear. In the case of the ZSS mutant, an accumulation around SYCP3 stained region is proposed. A staining of earlier stage (e.g. leptotene) would have been essential to discriminate between a DNA DSB formation defect vs delay in repair, even if mouse a defect in DSB formation was not observed in ZSS mutants. Thus, staging is important. In addition, the quantification of global signal intensity per nucleus is essential to conclude.

Regarding SYCP1 staining and SC formation in ZSS mutant, which is a very important point to assess conservation with mice, without any quantification of the signal, the authors cannot conclude of such an effect, which is antagonist and thus unexpected. As for gH2AX a clear staging has to be done (different prophase I stages), and the entire images, not zoom, of the different stages, with separate channel (as incompletely proposed in Sup 12) as to be included in the main figures.

The RAD51 staining looks conclusive, and this accumulation/delay compatibles to what is known in the mouse. However, a careful analysis required a signal quantification.

As a comment: published work using different macro are available in the literature to quantify recombination staining.

I thank the authors for this experiment with MSH5, and I agree with their conclusion and quantification.

Reviewer 3 fully understand the difficulty in obtaining humans samples, and despite the variability with previous experiment I warmly recommend to incorporate the new figures in the main text of the revised manuscript.

Regarding my previous demands, I want to stress that I took this comment of the authors under considerations and recommend some improvement which can be made based on already done image acquisition, or by additional acquisitions, but not additional

staining.

***** Reviewer's comments *****

Referee #1 (Comments on Novelty/Model System for Author):

This manuscript is of good impact and clinical relevance.

Referee #1 (Remarks for Author):

The revised manuscript still has my considerable enthusiasm. The authors have done due diligence in revising the original submission and preparing the manuscript specifically for EMBO Molecular Medicine, and it is much improved. Improvement is especially apparent in that the authors have kept conclusions consistent with what the data actually show; an excellent example of this is the change in title, but similar appropriate adjustments have been made throughout the manuscript.

The authors have responded satisfactorily to my previous comments. I appreciated the figures in OMERO. (As a somewhat picky point, I still find the shades of green/gray in the bar graphs within Figs. 2 and 3 hard to discriminate and would prefer to see primary colors; however, I leave this to journal copy editors.)

I think this report will be of interest to clinicians and to basic scientists in the field of meiosis and genetics of fertility/infertility.

We thank Reviewer #1 for the enthusiastic appreciation of our revised manuscript. Following the suggestions of the Reviewer and Editor, we have changed the colours of the bar graphs in Figs. 2 and 3 to primary colours for improved clarity. Besides, the revised colour scheme ensures accessibility for readers with colour vision deficiencies.

Referee #2 (Remarks for Author):

The authors addressed our issues well. I have no further suggestions.

We thank Reviewer #2 for confirming that all concerns have been completely addressed and for the full appreciation of our work.

Referee #3 (Comments on Novelty/Model System for Author):

The model is perfectly adequate to the question and the reviewer 3 fully understand the technical difficulty to work with mouse model.

We appreciate Reviewer #3's positive assessment of our preferred model system.

Referee #3 (Remarks for Author):

The manuscript can be edited to be more easy to understand for non human genetics specialist, which work on meiosis, using alternative models.

We thank Reviewer #3 for this suggestion and want to express our sincere gratitude to Reviewer #3 for the positive feedback in the initial review, which acknowledged the manuscript's overall quality and readability. We understand the importance of ensuring clarity for readers working on meiosis in alternative model systems and outside the human genetics field. As EMBO Molecular Medicine is dedicated to science at the interface between clinical research and basic life sciences, we have thoughtfully tailored the language of our manuscript to engage our interdisciplinary readership. The manuscript has also undergone professional language editing (see Acknowledgements). We will be happy to further clarify specific passages during the proof stage, if the editorial team finds it necessary.

Please find here more specific comments on the experiments:

Regarding the physical interaction:

The proposed experiments map the domain of M1AP required for the interaction with TEX11, but it is not an alternative/independent approach to verify the interaction. From a technical point of view, it is the same experiment.

Do the authors talk about point mutations or truncations? This has to be stated/explained clearly. Then, are the truncations up to 108, 226, 355 or 433 or do they start from these residues? It has to be clearly explained (drawing of the protein construct might be useful).

We thank Reviewer #3 for this important clarification request. We confirm that our analysis is based on point mutations in *M1AP*, which induce premature stop codons and thereby result in truncated proteins. To increase clarity, we have now included detailed information on the used truncation constructs of M1AP in the *Methods* section (l. 111ff, l.367ff, l.695ff).

Furthermore, we have implemented a schematic representation of the truncation products in Fig. EV1A to support understanding of the experimental design.

Truncation should not be discussed in bp but in numbers of amino acids.

We have revised the description of the truncation constructs to refer to amino acids rather than base pairs accordingly (l. 371, Fig EV1 legend).

Importantly, with the proposed experiment, the site of the interaction is not addressed, the domains are. And, it gives negative results, which can always be due to a defective folding, for example (which was not tested experimentally). It would have been relevant to test the minimum domain that interact (in other words a vice versa experiment), to get a positive result from this domain mapping experiment.

Finally, only the interaction M1AP / TEX11 is addressed. Is there any reason to speculate that M1AP uses the same domains to interact with all the ZSS members? Why focusing on TEX11?

For all these reasons, with only one approach, the validation of the interaction remains weak to me, and the domain mapping has to be clearly explained and not over interpreted.

We fully acknowledge that our approach does not define the precise binding site and cannot

exclude possible effects of misfolding in the truncated constructs. In line with the Editor's guidance not to perform additional experiments, we did not attempt to experimentally verify correct folding or pursue complementary "vice versa" interaction studies. To avoid over-interpretation, we have revised the manuscript's relevant sections and carefully adapted our statements on the protein-protein interaction. Regarding the focus on TEX11, we chose this protein for interaction studies because of its central role in the ZSS complex, its well-described function in humans, and its previously reported interaction with M1AP in mice. While we cannot exclude that M1AP may use different domains to interact with other ZSS components, this remains to be addressed in future studies. We hope that these clarifications adequately address the Reviewer's concerns.

Why not including alphafold modeling?

We thank Reviewer #3 for this suggestion. We explored the use of structural in silico modelling using AlphaFold predictions, along with protein-protein interaction analysis via ChimeraX. Our findings indicated that while the predicted AlphaFold models of M1AP, TEX11, and SPO16 yielded high confidence scores (pLDDT 90-70), the prediction for SHOC1 was of very low confidence (pLDDT<50) and therefore was omitted for the prediction of the interaction. Furthermore, we noted that the buried surface area between M1AP, TEX11, and SPO16, which is an indicator of stable interaction, were high. However, similarly elevated values were observed in negative control experiments. To assess the specificity of our prediction, we analysed the buried surface area between M1AP and a nuclear RNA export protein, as well as a Sertoli cell-specific glutamate dehydrogenase, neither of which are known to interact with M1AP *in vivo*. The results from these controls mirrored the high values described before. Given these limitations in both completeness and specificity, we decided not to include the structural model of the interaction of M1AP with TEX11 and SPO16 (and without SHOC1).

Regarding immunostaining of recombination protein:

I thank the authors for this information about antibody, and hopefully in the future and next studies these tools will appear.

As a comment, I would say that the for reviewer shown staining is only on sections and, to assess recombination proteins function immune-cytochemistry is more informative on prophase I chromosomes spreads. But, I do understand the difficulty in getting such human samples.

We thank Reviewer #3 for this thoughtful comment and the understanding of the limitations when working with rare human testicular samples. In addition to staining on testicular sections, we have also stained the available antibodies in human meiotic spreads.

Unfortunately, these experiments did not yield specific staining.

Reviewer 3 thanks the authors for their effort in providing a more complete characterization of the putative defect in HR, in the different mutant conditions.

The revised manuscript gave new additional comments on recombination starting by gH2AX staining. M1AP mutant pattern is clear. In the case of the ZSS mutant, an accumulation around SYCP3 stained region is proposed. A staining of earlier stage (e.g. leptotene) would have been essential to discriminate between a DNA DSB formation defect vs delay in repair, even if mouse a defect in DSB formation was not observed in ZSS mutants. Thus, staging is important. In addition, the quantification of global signal intensity per nucleus is essential to conclude.

We thank Reviewer #3 for this comment. As noted, the presented data were already included in our previous manuscript version. Our initial analysis focused on nuclei in the pachytene stage because this was the stage at which key differences in homologues recombination were apparent in men with *M1AP* loss-of-function variants compared to controls. For men with ZSS deficiency, proper pachytene cells were absent. Therefore, we emphasised qualitative differences observed in earlier stages, such as the impairment of synapsis or the absence of XY body formation/preserved DSB. We agree that a more precise staging (e.g.,

leptotene) and quantitative assessment of γ H2AX intensity per nucleus would have strengthened the conclusions. However, variability in sample processing across different centres and differences in fixation made it impossible to quantify the global signal intensity per nucleus. Furthermore, due to the extended SIM² microscopy analysis in our initial experiments, the respective slides are unfortunately bleached, and no additional images can be acquired. As noted in our previous response, novel analyses are not possible due to the limited material. In light of these limitations, we have carefully toned down the conclusions drawn from our observations to reach the qualitative nature of our analyses (l. 257f, l. 268, Fig. 4-8 legends).

Regarding SYCP1 staining and SC formation in ZSS mutant, which is a very important point to assess conservation with mice, without any quantification of the signal, the authors cannot conclude of such an effect, which is antagonist and thus unexpected. As for γ H2AX A clear staging has to be done (different prophase I stages), and the entire images, not zoom, of the different stages, with separate channel (as incompletely proposed in Sup 12) as to be included in the main figures.

We thank Reviewer #3 for this important comment and fully agree that SYCP1 staining and SC formation are key aspects for assessing the conservation of ZSS function between mice and humans. As noted, the respective data were already presented in the previous manuscript version. Our observations in human cases with ZSS deficiency are consistent with those described for corresponding mouse models, as discussed in l. 418 ff. Moreover, as written in our previous comment, the representations of different stages are not possible without performing new experiments. To avoid overinterpretation, we have revised the wording to adopt a more descriptive tone (l. 272ff, Fig. 4 and 5 legends). In response to the Reviewer's suggestion, we have now included full-field images with separated channels in the main figure panel, thanks for the improving suggestion

The RAD51 staining looks conclusive, and this accumulation/delay compatibles to what is known in the mouse. However, a careful analysis required a signal quantification.

As a comment: published work using different macro are available in the literature to quantify recombination staining.

We thank Reviewer #3 for this suggestion. We fully acknowledge that quantitative analysis would enhance the robustness of our findings. However, due to the age of the spreaded and at -80°C stored slides (over one year) the availability of suitable nuclei in the same meiotic stage was limited, and thus, it was not feasible to acquire a sufficient number of cells for reliable quantification. Consequently, we have presented representative nuclei at different stages of prophase I in Fig. 4 and 5. To further address this limitation, we have revised the manuscript and adopted a more descriptive tone in the relevant sections (l. 281ff, Fig. 4 and 5 legends). We appreciate the Reviewer's understanding of the challenges associated with acquiring new images under these circumstances.

I thank the authors for this experiment with MSH5, and I agree with their conclusion and quantification. Thanks to Reviewer #3.

Reviewer 3 fully understand the difficulty in obtaining humans samples, and despite the variability with previous experiment I warmly recommend to incorporate the new figures in the main text of the revised manuscript.

We thank Reviewer #3 for this recommendation and incorporated all figures showing meiotic spread staining in the main text of the revised manuscript.

Regarding my previous demands, I want to stress that I took this comment of the authors under considerations and recommend some improvement which can be made based on already done image acquisition, or by additional acquisitions, but not additional staining.

We thank Reviewer #3 for the thoughtful follow-up. As previously noted, the fluorescence signals have significantly diminished due to photobleaching from extended imaging sessions. This limits the acquisition of new, high-quality images. Besides, the limited availability of

suitable nuclei at specific meiotic stages across different samples comes with challenges for reliable quantification. In response to Reviewer #3's concerns, we have thoroughly revised the manuscript to adopt a more descriptive tone when discussing our observations (e.g., lines 281ff). We have also updated the legends of Fig. 4 and 5 to reflect these improvements. Moreover, we have incorporated full-field images with separate channels in our main figures to improve clarity and transparency. We hope these revisions address the reviewer's suggestions, meet all editor's requests, and contribute to the overall quality of our manuscript.

24th Apr 2025

Dear Dr. Friedrich,

We are pleased to inform you that your manuscript is accepted for publication and is now being sent to our publisher to be included in the next available issue of EMBO Molecular Medicine.

Zeljko Durdevic
Senior Editor
EMBO Molecular Medicine
